# Rethinking GNNs and Missing Features: Challenges, Evaluation and a Robust Solution

**Francesco Ferrini** [1] **Veronica Lachi** [2] **Antonio Longa** [2] **Bruno Lepri** [3] **Matono Akiyoshi** [4] **Andrea Passerini** [1] **Xin Liu** [4] **Manfred Jaeger** [5]

## Abstract

Handling missing node features is a key challenge for deploying Graph Neural Networks (GNNs) in real-world domains such as healthcare and sensor networks. Existing studies mostly address relatively benign scenarios, namely benchmark datasets with (a) high-dimensional but sparse node features and (b) incomplete data generated under *Missing Completely At Random (MCAR)* mechanisms. For (a), we theoretically prove that high sparsity substantially limits the information loss caused by missingness, making all models appear robust and preventing a meaningful comparison of their performance. To overcome this limitation, we introduce one synthetic and three real-world datasets with dense, semantically meaningful features. For (b), we move beyond MCAR and design evaluation protocols with more realistic missingness mechanisms. Moreover, we provide a theoretical background to state explicit assumptions on the missingness process and analyze their implications for different methods. Building on this analysis, we show that a simple baseline adapted to the graph domain is competitive with respect to specialized architectures across diverse datasets and missingness regimes.[1]

## 1. Introduction

Learning with missing features is a pervasive and often unavoidable challenge in many real-world machine learn-

[1]University of Trento, Trento, Italy [2]UiT, The Arctic University of Norway, Tromsø, Norway [3]Fondazione Bruno Kessler, Trento, Italy [4]AIST, Tokyo, Japan [5]Aalborg University, Aalborg, Denmark. Correspondence to: Francesco Ferrini <francesco.ferrini@unitn.it>, Veronica Lachi <veronica.lachi@uit.no>.

*Proceedings of the 43$^{rd}$ International Conference on Machine Learning*, Seoul, South Korea. PMLR 306, 2026. Copyright 2026 by the author(s).

[1]Code available at https://github.com/francescoferrini/gnnmim.

ing applications, such as healthcare (Braem et al., 2024; Mirkes et al., 2016), IoT sensor networks (Faizin et al., 2019; Okafor & Delaney, 2021; Agbo et al., 2022), and recommender systems (Marlin et al., 2007; He et al., 2017; Marlin et al., 2011). This issue naturally extends to Graph Neural Networks (GNNs), which are increasingly applied in domains where missing features are common. In this work, we focus specifically on the problem of *missing node feature data*, a setting that has received growing attention in the GNN literature (Um et al., 2023; Yun et al., 2024; Rossi et al., 2022; Guo et al., 2023; Taguchi et al., 2021; Errica & Niepert, 2024; Um et al., 2025)

A wide range of methods have been proposed, from simple mean imputation (You et al., 2020) to architectures that jointly impute and predict during training (Guo et al., 2023). These approaches are typically evaluated by synthetically removing features from widely used node classification benchmarks such as CORA, CITESEER, and PUBMED (Yang et al., 2016). However, despite the growing number of models, little attention has been paid to the validity of these evaluation protocols. We argue that two critical issues remained largely unaddressed: (i) the datasets used for evaluation, and (ii) the missingness mechanisms applied to generate incomplete features.

Regarding (i), existing evaluations rely on datasets with *extremely sparse* node features, typically bag-of-words representations where the vast majority of entries are zero. This raises a crucial question: *can robustness to missing features be meaningfully assessed when most features are already absent?* Our theoretical analysis shows that in highly sparse settings, the mutual information between features and labels is barely affected by additional missingness, except at extremely high missing rates. Empirically, we find that all the existing GNN-based methods maintain high performance across a wide range of missingness levels on these benchmarks, with performance degrading only when more than 90% of entries are removed. These results cast serious doubt on the ability of current benchmarks to meaningfully assess the robustness of the models.

To move beyond this limitation, we identify a set of datasets, one synthetic and three real-world, with dense, raw fea-

tures that are naturally low-dimensional and semantically meaningful (e.g., physical measurements). These datasets offer a more realistic setting for studying GNNs under feature missingness. This focus on dataset quality aligns with recent calls for more careful benchmark design in graph machine learning (Bechler-Speicher et al., 2025; Coupette et al., 2025).

Regarding (ii), the design of the missingness mechanisms used during evaluation is overly simplistic. Most prior works consider only *Missing Completely At Random (MCAR)* mechanisms (Rubin, 1976; Little & Rubin, 2019), where feature deletion is independent of the data. In practice, however, missingness is often related to the feature values or prediction target (Carreras et al., 2021; Hazewinkel et al., 2022; Kopra et al., 2015). For example, a patient might be less likely to report their weight if it is above a certain threshold. This corresponds to a Missing Not At Random (MNAR) mechanism (Rubin, 1976), in which the probability of missingness depends on the unobserved feature value itself. A further limitation of existing evaluation protocols is the implicit assumption that the missingness mechanism remains identical across training and test data. In practice, however, this is often not the case: for example, training data may be historical and collected with obsolete sensors prone to failures, while test data come from newer sensors with little or no missingness. To overcome this limitation of the current evaluation procedure, we design more realistic evaluation protocols. These include new, more representative instances of MCAR and MNAR mechanisms, as well as train–test distribution shifts. Such conditions more accurately capture real-world deployment challenges, where both the causes and the distributions of missing data may vary across stages.

Finally, we study a simple MIM augmentation (Van Ness et al., 2023) for GNNs: we concatenate the node features with their binary missingness mask and feed the resulting representation to an otherwise standard GNN. This yields a lightweight baseline that requires no learned imputation and does not rely on MAR assumptions, making it suitable for MNAR cases. We also evaluate it on RelBench (Robinson et al., 2024), a suite of large graph benchmarks constructed from relational-database with naturally occurring missing values and with unknown underlying missing mechanism; the MIM augmentation is competitive in this realistic setting.

**Contributions.** To summarize, our main contributions are:

1. We provide a theoretical analysis showing that the impact of missing features depends strongly on feature sparsity, and derive an information-theoretic bound on the resulting loss.

2. We introduce one synthetic and three real-world datasets with dense, informative features, and show experimentally that models appearing robust on sparse benchmarks fail on these datasets.

3. We propose realistic evaluation protocols, including new, more representative instances of MCAR and MNAR mechanisms and train–test distribution shifts, and demonstrate that existing methods are not robust to all the possible settings.

4. We show that adding a simple MIM component to existing GNN architectures is competitive with respect to existing approaches across datasets, missingness types, distribution shifts, and benchmarks with naturally occurring missingness.

The core aim of this paper is to enable more reliable evaluation of GNNs with missing node features. We show that apparent robustness is often driven by evaluation artifacts, namely sparse features and overly benign missingness mechanisms. By combining dense, semantically meaningful datasets, realistic missingness protocols, and a clear theoretical framing, we establish a foundation that enables more meaningful and reliable research directions. Within this improved evaluation setup, adding a simple MIM component to GNNs yields a lightweight baseline that avoids MAR assumptions, making it ideal under MNAR mechanisms and robust for real-world datasets where the missingness process is unknown.

## 2. Learning from Incomplete Graph Data

We consider an attributed graph $G = (V, E, \mathbf{X}, \mathbf{Y})$, where $V = \{1, \ldots, n\}$ is the set of nodes, $E \subseteq V \times V$ is the set of edges represented by the adjacency matrix $\mathbf{A} \in \{0, 1\}^{n \times n}$, $\mathbf{X} \in \mathbb{R}^{n \times d}$ is the node feature matrix with entry $X_{ij}$ denoting feature $j$ of node $i$, and $\mathbf{Y} \in \mathcal{Y}^n$ is the vector of node labels.

When data is incomplete, some entries of $\mathbf{X}$ are unobserved. Let $\mathbf{M} \in \{0, 1\}^{n \times d}$ be the missingness indicator matrix that has $M_{ij} = 1$ if $x_{ij}$ is missing and 0 otherwise. In our setting, the missingness indicator matrix $\mathbf{M}$ is directly and deterministically constructed from the observed dataset. Missing values are explicitly marked in the raw data, so the mask $\mathbf{M}$ is uniquely defined and contains no uncertainty. Let $\mathbf{X}^{obs}$ be the elements of $\mathbf{X}$ for which $M_{ij} = 0$, and $\mathbf{X}^{miss}$ the elements for which $M_{ij} = 1$. The observed data from which we learn then can be written as $\mathbf{X}^{obs}, \mathbf{Y}, \mathbf{M}$. We note that we here make the assumption that $\mathbf{Y}$ is fully observed in the (training) data, and that there is no uncertainty about the graph structure $E$. The distribution of the data then can

be parameterized as

$$P_{\boldsymbol{\theta},\boldsymbol{\gamma},\boldsymbol{\lambda}}(\mathbf{X}^{obs}, \mathbf{Y}, \mathbf{M}) = \int_{\mathbf{X}^{miss}} P_{\boldsymbol{\theta}}(\mathbf{X}) P_{\boldsymbol{\gamma}}(\mathbf{Y}|\mathbf{X}) P_{\boldsymbol{\lambda}}(\mathbf{M}|\mathbf{X}, \mathbf{Y}),$$
(1)

where $\mathbf{X} = \mathbf{X}^{obs} \cup \mathbf{X}^{miss}$, $P_{\boldsymbol{\theta}}$ is the node feature distribution, $P_{\boldsymbol{\gamma}}$ is the conditional label distribution, and $P_{\boldsymbol{\lambda}}$ represents the *missingness mechanism*. Though not explicitly reflected in the notation, all these distributions will usually depend on the underlying graph structure, which will typically induce dependencies among the rows of $\mathbf{X}$, and among the elements of $\mathbf{Y}$.

A GNN for node classification with complete feature data is a model $P_{\boldsymbol{\gamma}}(\mathbf{Y}|\mathbf{X})$ with $\boldsymbol{\gamma}$ the weights of the GNN. For classification with incomplete data we need to learn the conditional model

$$P_{\boldsymbol{\theta},\boldsymbol{\gamma},\boldsymbol{\lambda}}(\mathbf{Y} \mid \mathbf{X}^{obs}, \mathbf{M}) = \int_{\mathbf{X}^{miss}} P_{\boldsymbol{\theta},\boldsymbol{\gamma},\boldsymbol{\lambda}}(\mathbf{Y} \mid \mathbf{X}, \mathbf{M})$$
$$P_{\boldsymbol{\theta},\boldsymbol{\gamma},\boldsymbol{\lambda}}(\mathbf{X}^{miss} \mid \mathbf{X}^{obs}, \mathbf{M}). \quad (2)$$

The classical *missing (completely) at random (M(C)AR)* assumptions (Rubin, 1976) simplify this problem. The original M(C)AR assumptions have been formulated in the context of estimating the parameter of a generative distribution. It has been observed that more specialized variations of the original definitions can be more pertinent in the context of classification (Ding & Simonoff, 2010; Ghorbani & Zou, 2018). In the following we give formulations of M(C)AR for classification that provide the foundations for our theoretical analysis.

**Definition 2.1.** The joint distribution $P_{\boldsymbol{\theta},\boldsymbol{\gamma},\boldsymbol{\lambda}}$ is *feature-MAR*, if

$$P_{\boldsymbol{\gamma},\boldsymbol{\lambda}}(\mathbf{M}|\mathbf{X}^{miss}, \mathbf{X}^{obs}) = P_{\boldsymbol{\theta},\boldsymbol{\gamma},\boldsymbol{\lambda}}(\mathbf{M}|\mathbf{X}^{obs}). \quad (3)$$

It is *label-MAR* if

$$P_{\boldsymbol{\lambda}}(\mathbf{M}|\boldsymbol{X}, \boldsymbol{Y}) = P_{\boldsymbol{\gamma},\boldsymbol{\lambda}}(\mathbf{M}|\boldsymbol{X}). \quad (4)$$

The distribution is MCAR, if

$$P_{\boldsymbol{\lambda}}(\mathbf{M}|\boldsymbol{X}, \boldsymbol{Y}) = P_{\boldsymbol{\theta},\boldsymbol{\gamma},\boldsymbol{\lambda}}(\mathbf{M}). \quad (5)$$

In (3)-(5) all probability functions are indexed with the parameters they actually depend on. Note, for example, that the conditional of $\mathbf{M}$ given $\mathbf{X}$ requires marginalization over $\mathbf{Y}$, and thereby also depends on the parameter $\boldsymbol{\gamma}$. MCAR implies both feature- and label-MAR.

The simplest realization of an MCAR mechanism is *uniform missingness (U-MCAR)* in which entries of $\mathbf{X}$ are independently missing with a fixed missingness probability $\mu$. This can be generalized by defining a missingness probability

matrix $\boldsymbol{\mu} \in [0,1]^{n \times d}$ specifying potentially different missingness probabilities for different entries of $\mathbf{X}$.

MAR assumptions allow us to eliminate the missingness model $P_{\boldsymbol{\lambda}}$ from (2). The following proposition states this classical *ignorability* result in a version most suitable in our context.

**Theorem 2.2.** *If $P_{\boldsymbol{\theta},\boldsymbol{\gamma},\boldsymbol{\lambda}}$ is feature-MAR and label-MAR, then (2) simplifies to*

$$\int_{\mathbf{X}^{miss}} P_{\boldsymbol{\gamma}}(\boldsymbol{Y}|\boldsymbol{X}) P_{\boldsymbol{\theta}}(\mathbf{X}^{miss}|\mathbf{X}^{obs}). \quad (6)$$

**Intuition.** Under feature-MAR and label-MAR, the missingness pattern carries no predictive information. The learning problem reduces to the usual classification task with imputed features, meaning that methods explicitly modeling the missingness mask do not gain theoretical advantage in this regime.

The proof is straightforward by rewriting the two factors on the right of (2) using Bayes's rule, and plugging in (3) and (4). Formulation (6) still poses two major challenges: it requires a feature distribution model $P_{\boldsymbol{\theta}}$ when in reality we only are interested in the conditional model $P_{\boldsymbol{\gamma}}$, and the integration over $\mathbf{X}^{miss}$ is usually intractable (Ipsen et al., 2022). The simplest approach to address these problems is to approximate the integral (6) by evaluating $P_{\boldsymbol{\gamma}}(\boldsymbol{Y}|\boldsymbol{X})$ at a single imputed value $\mathbf{X} = impute(\mathbf{X}^{miss})$ (Rubin, 1988). This does not require an explicit model for $P_{\boldsymbol{\theta}}$, but relies on the implicit assumption that the imputed value $impute(\mathbf{X}^{miss})$ has high probability under $P_{\boldsymbol{\theta}}$. A simple example is *mean-imputation*, in which missing values of a given feature are filled with the mean of that feature; we will refer to this approach combined with a standard GNN as `GNNmi` (You et al., 2020). In addition, we also consider *zero-imputation*, where missing entries are replaced with zeros (`GNNzero`), and *median-imputation*, where they are filled with the feature median (`GNNmedian`). Similarly, `PCFI` (Um et al., 2023) does not require an explicit model for $P_{\boldsymbol{\theta}}$; it introduces a confidence-guided imputation scheme where pseudo-confidence is derived from the shortest-path distance to observed features, and combines channel-wise diffusion with inter-channel propagation to recover a single estimate of $\mathbf{X}$. `GOODIE` (Yun et al., 2024) approximates the integral in (6) using a combination of label propagation, while `FP` (Rossi et al., 2022) propagates features by minimizing a Dirichlet energy function, whereas `FairAC` (Guo et al., 2023) does so by aggregating, via an attention mechanism, the representations from neighbors of nodes with missing features.

Other methods explicitly model $P_{\boldsymbol{\theta}}$. The `GCNmf` approach of Taguchi et al. (2021) introduces a model of $P_{\boldsymbol{\theta}}$ in the form of a mixture of Gaussians, and approximates (6) by $P_{\boldsymbol{\gamma}}(\boldsymbol{Y}, |, \mathbb{E}_{\theta}[\mathbf{L}1 \mid \mathbf{X}^{obs}])$, where $\mathbb{E}_{\theta}[\mathbf{L}_1 \mid \mathbf{X}^{obs}]$ is the ex-

pected activation at the first layer of the GNN defining $P_\gamma$. Finally, GSPN (Errica & Niepert, 2024) explicitly models $P_\theta$ with graph-induced sum–product networks, so missing features are handled by exact marginalization.

An alternative to all these approaches that work entirely with models $P_\theta$, $P_\gamma$ for the (complete) data distribution is to include the missingness mechanism explicitly in a model $P_{\gamma^+}(\boldsymbol{Y}|\mathbf{X}^{obs}, \boldsymbol{M})$, that directly captures the left side of (2). We here write $\gamma^+$ for the parameters of the model to emphasize that it can be structurally similar to a model $P_\gamma(\boldsymbol{Y}|\boldsymbol{X})$, but different in that it has the missingness matrix $\boldsymbol{M}$ as an explicit extra input. This modeling strategy, often referred to as the Missing Indicator Method (MIM), has been studied in the context of supervised learning with missing features (Van Ness et al., 2023), and applied in other domains such as multivariate time series imputation (Cao et al., 2018; Cini et al., 2022), where however the focus is on reconstructing missing values under stationary MAR-style mechanisms. To the best of our knowledge, MIM has not been explored in the context of graph machine learning for node level tasks under arbitrary missingness mechanisms, including MNAR and train–test distribution shifts. In this work, we propose a GNN-based instantiation of the MIM framework, which we call GNNmim. In GNNmim, we implement $P_{\gamma^+}$ as a GNN, we construct the matrix *zero-fill*($\mathbf{X}^{obs}$) in which missing values are filled in by zeros, and use the concatenation *zero-fill*($\mathbf{X}^{obs})_{i,:}||\boldsymbol{M}_{i,:}$ as the feature vector for node $i$ in an otherwise standard GNN architecture[2]. GNNmim does not rely on any MAR assumptions, and thereby can be expected to perform more robustly than other approaches under different missingness mechanisms. As our experiments in Section 5 show, this simple yet principled strategy yields robust performance across a wide variety of missingness scenarios. In Appendix J we provide additional analyses where the missing-feature mask is applied not only to zero imputation but also to the existing models presented in this section, and in Appendix L we further compare GNNmim against classical iterative imputation MICE (Van Buuren & Groothuis-Oudshoorn, 2011) and non-graph baselines, namely MLP+MIM and XGBoost (Chen & Guestrin, 2016).

## 3. Are we evaluating GNNs for missing features on the right data?

A rigorous evaluation of GNNs under feature missingness requires not only well-designed models, but also datasets that are suitable for the problem at hand. Recent work in the

[2]We deliberately here say "zero-filling" rather than "zero-imputation". The latter would imply that we view the zeros as somehow reasonable stand-ins for the true unobserved values. We view the zeros as arbitrary placeholders. Ideally, the trained model will learn to ignore these values when the corresponding missingness indicator is 1.

*Table 1.* Feature sparsity across benchmarks and custom datasets.

| Dataset | #Nodes | #Features | Sparsity ↓ | Type of features |
|---|---|---|---|---|
| CORA | 2708 | 1433 | 0.9873 | BoW (binary) |
| CITESEER | 3327 | 3703 | 0.9915 | BoW (binary) |
| PUBMED | 19717 | 500 | 0.8998 | BoW (binary) |
| SYNTHETIC | 1000 | 5 | 0.0000 | Gaussian |
| AIR | 430 | 7 | 0.1615 | Raw |
| ELECTRIC | 2000 | 5 | 0.2000 | Raw |
| TADPOLE | 555 | 15 | 0.0000 | Raw |

graph learning community has emphasized the importance of dataset suitability in benchmarking (Bechler-Speicher et al., 2025; Coupette et al., 2025). In the context of learning with missing node features, dataset suitability is even more critical. Models designed to handle missingness should be tested on datasets where the presence of missing features meaningfully affects model performance and where reasoning under missingness is necessary and non-trivial.

The current standard practice in the literature is to evaluate state-of-the-art methods on a set of widely-used benchmarks for node-level tasks, namely, CORA, CITESEER, PUBMED, AMAZONCOMPUTERS, and AMAZONPHOTO. In these datasets, node features are constructed as follows: CORA, CITESEER and PUBMED use binary bag-of-words features, while AMAZONCOMPUTERS and AMAZONPHOTO use TF-IDF vectors (Aizawa, 2003). These feature matrices are typically very sparse, which we quantify using the notion of *feature sparsity*, formally defined as below:

**Definition 3.1** (Feature Sparsity). Given a node feature matrix $\mathbf{X} \in \mathbb{R}^{n \times d}$, the *feature sparsity* is defined as the proportion of zero entries: $s(\mathbf{X}) = \frac{1}{nd} \sum_{i=1}^{n} \sum_{j=1}^{d} \mathbf{1}[X_{ij} = 0]$, where $\mathbf{1}[\cdot]$ denotes the indicator function.

The sparsity values of the benchmark datasets are reported in Table 1 (first three rows). All datasets exhibit substantial sparsity, with more than 50% of features being zero across all the datasets, with Citeseer reaching an extreme sparsity level of approximately 99%. This raises a crucial question: does it make sense to evaluate models designed to handle missing features on datasets where the feature representations are already extremely sparse? In such sparse settings, a high probability of missingness is needed to induce a meaningful information loss. Otherwise, the observed model performance under missingness may reflect artifacts of the dataset rather than the robustness of the method. We formalize this observation in the following theorem.

**Theorem 3.2.** *Let $\mathbf{X} \in \mathbb{R}^{n \times d}$ and $\mathbf{Y} \in \mathcal{Y}^n$ be random variables, $\mathbf{M} \in \{0,1\}^{n \times d}$ be a missingness mask and $\mathbf{X}^{\text{obs}}$ denotes the observed (incomplete) data. We encode the pair*

$(\mathbf{X}^{\text{obs}}, \mathbf{M})$ *with the random variable* $\tilde{\mathbf{X}}$ *with*

$$\tilde{X}_{ij} = \begin{cases} X_{ij}, & M_{ij} = 0, \\ ?, & M_{ij} = 1. \end{cases}$$

*Let the change in the information be defined as* $\Delta := I(\mathbf{Y}; \tilde{\mathbf{X}}) - I(\mathbf{Y}; \mathbf{X})$, *where* $I(\cdot; \cdot)$ *denotes the mutual information. Then,*

1. *If the missingness is label-MAR, then* $\Delta \leq 0$.

2. *If* $\mathbf{X} \in \{0, 1\}^{n \times d}$ *and the missingness is U-MCAR with missingness probability* $\mu$, *and* $s(\mathbf{X})$ *is the sample sparsity as in Definition 3.1, then*

$$- nd\,\mu\,h_2\big(\mathbb{E}[s(\mathbf{X})]\big) \leq \Delta \leq 0,$$

*where* $h_2(u) = -u \log u - (1-u) \log(1-u)$.

**Intuition.** When node features are extremely sparse (e.g., BoW/TF-IDF), the information loss induced by missingness is provably bounded by a quantity that vanishes with sparsity, unless missingness is extremely high. As a result, existing sparse benchmarks inherently make all methods appear robust, preventing meaningful comparison.

The proof can be found in Appendix A. Theorem 3.2 demonstrates that when feature sparsity is high, a very large amount of missingness is required to produce a meaningful loss of information. This confirms that such benchmarks do not meaningfully differentiate between approaches, casting doubt on their suitability for evaluating GNNs under feature missingness. As a consequence, we argue for the use of datasets where missingness poses a real challenge. In particular, we introduce a set of four alternative datasets, one new synthetic and three real-world. More details about the datasets are reported in Appendix C.

**(1) A synthetic dataset tailored to controlled missingness.** We construct a dataset based on a Barabási–Albert graph topology, where node features are sampled from a Gaussian distribution. Node labels are assigned using a fixed two-layer GCN applied to the full, complete features, ensuring that a GNN model has the capacity to achieve high classification accuracy in the absence of missingness. This controlled setting provides a testbed for isolating the effects of missingness under varying sparsity, while maintaining a well-defined ground truth.

**(2) Real-world datasets with semantically meaningful features.** We also advocate for the use of real datasets in which node features correspond to raw, observable properties: 1) **AIR** (Zheng et al., 2015), a sensor network dataset from IoT applications, where node features correspond to environmental measurements and node labels indicate sensor status categories; 2) **ELECTRIC** (Birchfield et al., 2016;

*Table 2.* Evaluation of P1 (feature-structure separability) and P2 (feature-structure complementarity) on our custom datasets. Each cell reports the KS statistic, with all $p$-values ranging from $1.93e-14$ to $8.80e-62$, for separability under six perturbation settings. $\gamma_{1,1}$ indicates the feature-structure complementarity. Datasets satisfying each property (as per Coupette et al. (2025)) are marked with $\checkmark$.

| Setting | SYNTHETIC | AIR | ELECTRIC | TADPOLE |
|---|---|---|---|---|
| Empty Feat. | 1.00 | 1.00 | 1.00 | 1.00 |
| Random Feat. | 1.00 | 1.00 | 1.00 | 0.90 |
| Complete Feat. | 1.00 | 1.00 | 1.00 | 0.61 |
| Empty Graph | 1.00 | 0.67 | 0.98 | 0.77 |
| Random Graph | 1.00 | 1.00 | 1.00 | 1.00 |
| Complete Graph | 1.00 | 1.00 | 1.00 | 1.00 |
| $\gamma_{1,1}$ | 0.62 | 0.68 | 0.69 | 0.64 |
| **P1** | $\checkmark$ | $\checkmark$ | $\checkmark$ | $\checkmark$ |
| **P2** | $\checkmark$ | $\checkmark$ | $\checkmark$ | $\checkmark$ |

Baek & Birchfield, 2023), a dataset of interconnected electrical sensors, with real-valued measurements as features and operational condition classification as the target task; 3) **TADPOLE** (Zhu et al., 2019), a medical graph dataset derived from the TADPOLE challenge, where each node represents a patient, node features include clinical and imaging biomarkers, and the goal is to predict diagnostic labels.

Both the synthetic and real-world datasets exhibit low feature sparsity (Table 1), a necessary condition for studying missingness. However, sparsity alone is not sufficient: suitable datasets must also ensure that both features and structure are task-informative and interact non-trivially. We assess this using the RINGS framework (Coupette et al., 2025), which measures performance separability through KS statistics under perturbations (e.g., removing all edges or replacing features with noise) and complementarity of the topology of features through the normalized Gromov–Wasserstein distance $\gamma_{1,1}$ between the structural and feature-induced metric spaces (values greater than 0.5 are considered satisfactory). As shown in Table 2, all proposed datasets satisfy both mode complementarity and performance separability. Combined with their low feature sparsity, these properties make the datasets more suitable than traditional benchmarks for evaluating robustness to incomplete node attributes.

While the real-world datasets we introduce have moderate numbers of nodes and features, they satisfy the key requirements for evaluating robustness to missing node features. In contrast, most commonly used node classification benchmarks, especially larger-scale ones, rely on either extremely sparse feature representations or on learned embeddings, both of which are ill-suited for evaluating robustness to missing node features. Extreme sparsity is problematic in this setting, as shown by Theorem 3.2. Learned embeddings are likewise unsuitable for two reasons. First, missing en-

tries are unrealistic in artificially constructed representations: embeddings are deterministic outputs of an algorithm, and therefore do not naturally admit feature-level missingness. Second, the information encoded by embeddings is typically distributed across many dimensions in an overparameterized manner (Arora et al., 2016a;b), so that introducing missing features does not meaningfully expose the effects of missingness, making them hard to interpret, as confirmed by the experiment in Appendix K. To the best of our knowledge, all existing large-scale graph datasets for node classification rely either on extremely sparse feature representations or on learned embeddings, making them ill-suited for studying robustness to missing node features; in Appendix D we provide a systematic analysis of existing benchmarks, including large-scale ones, highlighting their limitations in this respect. Importantly, while our real-world datasets are of moderate size, this does not constitute a limitation for studying the effects of feature missingness. We complement them with experiments on the large-scale datasets with real missingness from RelBench (Robinson et al., 2024) (Section 5), and further show that dataset scale does not hinder meaningful evaluation (Appendix F).

## 4. Beyond Uniform Missingness

Dataset suitability is only one dimension of the evaluation problem. A second, equally important factor is the choice of the missingness mechanism under which models are tested. In the literature, nearly all prior works adopt a masking scheme based on *U-MCAR* mechanism. In other works (Taguchi et al., 2021; Um et al., 2023), a different variant is used where entire feature vectors of randomly selected nodes are masked. We denote this as ***Structural MCAR (S-MCAR)***. These two settings have become the default evaluation standards in the context of graph learning. We argue that more challenging and realistic missing data patterns need to be considered for a more informative evaluation of different methods' capabilities. We first introduce a more challenging MCAR mechanism:

***Label–Dependent MCAR (LD-MCAR).*** Missingness here is applied at the feature (column) level, assigning higher missingness probability to features $X_{:,j}$ that are more informative for the label, as measured by the mutual information $I(X_{:,j}; Y)$. Then, each entry $X_{ij}$ is masked independently with probability $P(M_{ij} = 1) = \rho \cdot I(X_{:,j}; Y)$, where $\rho \in [0, 1]$ is a scaling factor selected to achieve the overall desired expected missingness rate across the dataset. Importantly, this mechanism is still MCAR: the probability that a specific entry is missing does not depend on the actual value of the feature or the label, but only on the mutual information of the feature column and the label.

In many practical scenarios, missing features are related to their values or to the prediction target (Ghorbani & Zou, 2018; Mohan & Pearl, 2021; Jaeger, 2022; Van Ness et al., 2023). For instance, a patient might be less likely to report their weight if it is above a certain threshold. This corresponds to a MNAR mechanism. Testing GNN models exclusively under MCAR conditions fails to capture the challenge of more realistic settings. We therefore propose two different MNAR scenarios:

***Feature-Dependent MNAR (FD-MNAR).*** In this mechanism the probability of missingness depends on the value of the feature itself. In particular, we assume that extreme feature values, e.g., high quantiles, are more likely to be missing, as often observed in real-world settings such as healthcare, where abnormal values may be withheld. Formally, for each feature column $j$, let $q_j^{(\tau)}$ denote the $\tau$-quantile of the observed values. We define the missingness probability for entry $X_{ij}$ as:

$$P(M_{ij} = 1) = \begin{cases} \mu^{\mathrm{hi}} & \text{if } X_{ij} \geq q_j^{(\tau)}, \\ \mu^{\mathrm{lo}} & \text{otherwise,} \end{cases}$$

with $\mu^{\mathrm{hi}} > \mu^{\mathrm{lo}}$ and both selected to match a desired overall missingness rate.

***Class–Dependent MNAR (CD-MNAR).*** In this mechanism, features whose values are informative for the label are more likely to be omitted. For example, in medical datasets, patients may be less likely to disclose whether they smoke, a feature strongly associated with the label indicating a history of heart attack. To identify such dependencies, we train a decision tree classifier in a one-vs-rest setting, using the observed features to predict class membership. For each class $c \in \{1, \ldots, C\}$, we extract decision paths that lead to leaf nodes predicting $c$. These paths define a set of feature-value conditions that contribute to the prediction of class $c$, which we denote as $\mathcal{R}_c$. Let $\mathrm{Cond}_c(j, X_{ij})$ be a predicate that evaluates to true if the value of feature $j$ for node $i$ satisfies at least one condition in $\mathcal{R}_c$. Then, the missingness probability is defined as:

$$P(M_{ij} = 1 \mid Y_i = c) = \begin{cases} \mu^{\mathrm{hi}} & \text{if } \mathrm{Cond}_c(j, X_{ij}) = \text{true}, \\ \mu^{\mathrm{lo}} & \text{otherwise,} \end{cases}$$

where $\mu^{\mathrm{hi}} > \mu^{\mathrm{lo}}$, and both are selected to meet a target overall missingness rate.

In almost all existing experimental studies the missingness mechanism is the same in training and test data. An exception is (Ding & Simonoff, 2010), where two types of test data are considered: data that underlies the same missingness as the training data, and complete data. We consider a possible distribution shift in $P_{\boldsymbol{\lambda}}(\boldsymbol{M} \mid \boldsymbol{X}, \boldsymbol{Y})$ to be an important concern for two reasons: first, it represents a realistic

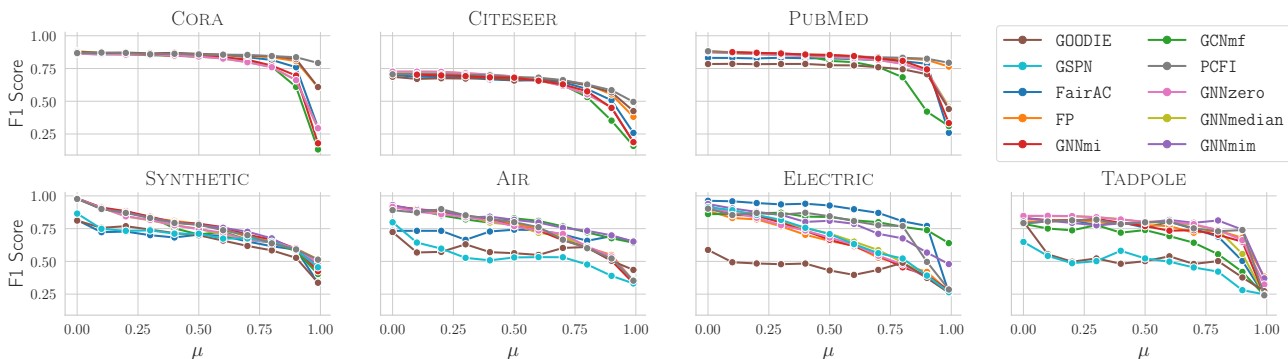

*Figure 1.* Mean F1-score across 5 runs as a function of the missingness probability $\mu$ on the proposed datasets and established benchmarks. Each panel reports the performance of all models on a specific dataset under the **S-MCAR** setting. The complete tables for all missingness mechanisms are provided in Appendix B.

|  | SYNTHETIC | | | | | AIR | | | | | ELECTRIC | | | | | TADPOLE | | | | |
|---|---|---|---|---|---|---|---|---|---|---|---|---|---|---|---|---|---|---|---|---|
| GOODIE | 0.77 | 0.67 | 0.76 | 0.61 | 0.59 | 0.68 | 0.58 | 0.71 | 0.60 | 0.58 | 0.59 | 0.45 | 0.59 | 0.44 | 0.43 | 0.75 | 0.50 | 0.73 | 0.53 | 0.47 |
| GSPN | 0.70 | 0.69 | 0.64 | 0.66 | 0.68 | 0.55 | 0.53 | 0.59 | 0.66 | 0.64 | 0.62 | 0.68 | 0.47 | 0.71 | 0.72 | 0.58 | 0.48 | 0.51 | 0.61 | 0.58 |
| FairAC | 0.69 | 0.67 | 0.62 | 0.64 | 0.63 | 0.69 | 0.71 | 0.53 | 0.70 | 0.69 | 0.86 | 0.87 | 0.58 | 0.82 | 0.85 | 0.72 | 0.72 | 0.66 | 0.69 | 0.67 |
| FP | 0.75 | 0.77 | 0.69 | 0.74 | 0.73 | 0.77 | 0.75 | 0.64 | 0.80 | 0.79 | 0.65 | 0.64 | 0.46 | 0.63 | 0.69 | 0.76 | 0.75 | 0.65 | 0.80 | 0.77 |
| GNNmi | 0.75 | 0.77 | 0.69 | 0.75 | 0.73 | 0.76 | 0.74 | 0.64 | 0.80 | 0.75 | 0.66 | 0.65 | 0.47 | 0.67 | 0.67 | 0.74 | 0.74 | 0.67 | 0.75 | 0.77 |
| GCNmf | 0.72 | 0.74 | 0.69 | 0.71 | 0.70 | 0.71 | 0.80 | 0.68 | 0.72 | 0.71 | 0.84 | 0.82 | 0.72 | 0.81 | 0.81 | 0.73 | 0.66 | 0.68 | 0.77 | 0.74 |
| PCFI | 0.75 | 0.76 | 0.69 | 0.74 | 0.71 | 0.76 | 0.74 | 0.63 | 0.77 | 0.78 | 0.79 | 0.78 | 0.52 | 0.76 | 0.78 | 0.77 | 0.76 | 0.66 | 0.79 | 0.78 |
| GNNzero | 0.74 | 0.75 | 0.64 | 0.75 | 0.74 | 0.78 | 0.74 | 0.65 | 0.79 | 0.79 | 0.64 | 0.65 | 0.45 | 0.62 | 0.66 | 0.79 | 0.78 | 0.73 | 0.80 | 0.81 |
| GNNmedian | 0.74 | 0.74 | 0.64 | 0.75 | 0.71 | 0.79 | 0.74 | 0.65 | 0.80 | 0.76 | 0.67 | 0.67 | 0.45 | 0.61 | 0.64 | 0.76 | 0.76 | 0.72 | 0.76 | 0.77 |
| GNNmim | 0.75 | 0.77 | 0.72 | 0.76 | 0.76 | 0.79 | 0.81 | 0.76 | 0.82 | 0.81 | 0.80 | 0.77 | 0.66 | 0.81 | 0.82 | 0.79 | 0.78 | 0.71 | 0.80 | 0.81 |
|  | U
MCAR | S
MCAR | LD
MCAR | FD
MNAR | CD
MNAR | U
MCAR | S
MCAR | LD
MCAR | FD
MNAR | CD
MNAR | U
MCAR | S
MCAR | LD
MCAR | FD
MNAR | CD
MNAR | U
MCAR | S
MCAR | LD
MCAR | FD
MNAR | CD
MNAR |

*Figure 2.* Column-normalized heatmaps showing the AUC (area under the F1 vs. missingness rate $\mu$ curve) for each model, dataset, and missingness mechanism. Higher values (lighter colors) indicate better overall robustness across increasing levels of missingness.

scenario in practical applications. For instance, training data may consist of historical patient records where some features are self-reported and selectively omitted for personal reasons, inducing an MNAR missingness mechanism. At test time, data collection may be fully automated, so that features are either always observed or missing independently of their values (MCAR). This results in a shift from MNAR missingness during training to complete or MCAR data at test time.

The second reason for considering distribution shifts in $P_\lambda$ is to assess a possible weakness of GNNmim: as a model of the form $P_{\gamma^+}(Y|\mathbf{X}^{obs}, M)$ it explicitly incorporates a model of the missingness mechanism, and thereby could be expected to be less robust under missingness distribution shifts than models that are based on MAR assumptions and (6) (which would be expected to be robust as long as the mechanism is feature and label MAR in both training and test data). We therefore define two evaluation regimes (R1 and R2) with and without a shift in the missingness process. Let $\mu_{\mathrm{tr}}(\mathbf{M} \mid \mathbf{X}, \mathbf{Y})$ and $\mu_{\mathrm{te}}(\mathbf{M} \mid \mathbf{X}, \mathbf{Y})$ denote the missingness distributions in training and testing, respectively.

**R1: *i.i.d. missingness* (no shift).** The same missingness mechanism (*U-MCAR, S-MCAR, LD-MCAR, FD-MNAR, CD-MNAR*) and rate are applied to training and test data, i.e., $\mu_{\mathrm{tr}} = \mu_{\mathrm{te}}$.

**R2: *missingness distribution shift* (train $\neq$ test).** In this setting, we evaluate combinations of a training missingness mechanism $M_{\mathrm{tr}} \in \{FD\text{-}MNAR, CD\text{-}MNAR\}$ with missingness probability $\mu_{\mathrm{tr}} = 50\%$, and a test missingness mechanism $M_{\mathrm{te}} = U\text{-}MCAR$ with missingness probability $\mu_{\mathrm{te}} \in \{0\%, 25\%, 50\%\}$.

While many different types of shifts in the missingness mechanism are possible, we focus on this setting as it captures a simple yet realistic scenario: training data affected by data-dependent missingness, followed by test data collected through automated processes, where missingness is either absent or independent of the data.

## 5. Experimental Results

We conduct experiments on node-level tasks using the datasets introduced in Section 3 and the more realistic missingness protocols described in Section 4. We compare the GNN-based models designed to handle missing features described in Section 2, namely GNNzero, GNNmedian, GNNmi, GCNmf, GOODIE, GSPN, PCFI, FP, and FairAC as well as our proposed method, GNNmim. Following the evaluation protocol adopted by these competitors, we perform all main experiments in a transductive setting. However, we note that GNNmim can also be applied in an inductive scenario; for completeness, in Appendix I we report

additional experiments conducted under an inductive setting. For all the experiments, we decide to treat the specific GNN layer type in `GNNmedian`, `GNNzero`, `GNNmi` and `GNNmim` as a hyperparameter optimized on a validation set. Full implementation details and hyperparameter settings are provided in Appendix E. The code is provided in the supplementary material. The experiments are designed to answer the following research questions:

- **Q1**: To what extent does dataset choice influence conclusions about GNN robustness to missing features?

- **Q2:** How robust are different models for handling incomplete features under different types of missingness?

- **Q3**: Do different models maintain their performance under distribution shifts in missingness between training and test sets?

- **Q4**: Do our findings on `GNNmim` generalize to large-scale benchmarks with naturally occurring missingness?

**Q1:** To assess the impact of the dataset on evaluating robustness under different missingness rates, we compute the F1 score for each model as a function of the missingness rate $\mu$. Figure 1 reports these curves under *Structural MCAR (S-MCAR)* and R1 regimes (see Section 4) for both the standard benchmarks (CORA, CITESEER, PUBMED) and the datasets we propose (ELECTRIC, AIR, TADPOLE, and SYNTHETIC). Results for other missingness mechanisms lead to equal conclusions and are included in Appendix B.

On CORA, CITESEER, PUBMED, all models appear robust, as their F1 score remains high across a wide range of $\mu$, and only drops at very high missingness rates (85-90%). In contrast, on our proposed datasets, performance drops much earlier, often already at low missingness rates. On TADPOLE, the degradation is less pronounced at low $\mu$ overall; however, two models, GOODIE and GSPN, notably diverge from the rest, showing much weaker performance even with limited missingness. These results indicate that evaluating robustness solely on traditional benchmarks can lead to overly optimistic conclusions. To properly assess GNN behavior under different missing rates, it is essential to rely on more challenging datasets. Importantly, these trends are not an artifact of dataset scale, as we observe identical performance degradation patterns on larger variants of the proposed synthetic dataset, as reported in Appendix F.

**Q2:** To assess robustness across mechanisms, we compute the area under the F1–missingness curve (AUC) for each dataset, model, and missingness mechanism under R1 regimes (complete F1 results by model, dataset, missingness rate, and mechanism are reported in Appendix G). Figure 2 shows AUC heatmaps (lighter colors indicate better performance) for each mechanism and dataset. Many methods are highly sensitive to the missingness type. For instance, `FairAC` performs well under *S-MCAR* on ELEC-

TRIC (0.870 AUC, best overall) but degrades under *FD-MNAR* on SYNTHETIC (0.641, second-last). Similarly, GOODIE ranks first on SYNTHETIC with uniform missingness (0.771) but drops to 0.587 under *CD-MNAR*. Performance under *U-MCAR* is not predictive of robustness under realistic *FD-MNAR*, questioning evaluations based only on uniform or structure-based missingness. `GNNmim` achieves consistently high AUC across all missingness types and datasets, showing that broad robustness is attainable even with lightweight, non-MAR models. Beyond GNN-based competitors, `GNNmim` also substantially outperforms classical iterative imputation (MICE) and non-graph baselines (MLP+MIM, XGBoost) under CD-MNAR (Appendix L).

**Q3:** To evaluate model robustness under distribution shifts in missingness, we compute the F1 score (mean ± standard deviation over 5 runs) for each dataset, model, and shift configuration of the R2 regime (Section 4). Full results are in Appendix H; Figure 3 shows a representative subset of the best-performing models from Q2 (GNNmim, GNNmi, GCNmf, FP, PCFI), trained on *FD-MNAR* with $\mu_{\text{tr}} = 50\%$ and tested on *U-MCAR* with $\mu_{\text{te}} \in \{0\%, 25\%, 50\%\}$. Similar results hold for other models and for the case where the training missing mechanisms is *CD-MNAR* (Appendix H). Each panel shows one dataset, with F1 on the x-axis, models on the y-axis, and color encoding $\mu_{\text{te}}$ (yellow 0%, blue 25%, green 50%). Dots indicate mean F1, horizontal lines standard deviation, and the red vertical bar marks regime R1 with *FD-MNAR* on train and test and $\mu_{\text{tr}} = \mu_{\text{te}} = 50\%$. Two findings emerge:

1. Distribution-shift generalization is challenging: in most cases, performance under R2 test conditions (*U-MCAR*, 25%) is lower than in the i.i.d. R1 setting, despite less severe test missingness. This occurs when the blue dot ($\mu_{\text{te}} = 25\%$) lies left of the red bar ($\mu_{\text{tr}} = \mu_{\text{te}} = 50\%$), showing that shifts in missingness create a harder generalization problem not explained by severity alone. The effect is dataset-dependent, reinforcing the need to evaluate robustness across shifts and datasets.

2. `GNNmim` is competitive under R2 conditions: across datasets and test missingness levels, it achieves the highest F1 scores (yellow, blue, and green dots farther right), maintaining an advantage over alternative methods.

**Q4:** So far, our evaluation has isolated the effect of different missingness mechanisms by injecting controlled missingness into datasets. We now ask whether the same findings extend to a more realistic setting where missing values occur naturally and the underlying mechanism is unknown. To this end, we evaluate on RelBench (Robinson et al., 2024), a collection of large-scale temporal relational graphs constructed from relational databases (Ferrini et al., 2024; 2025). We select tasks from databases with the highest observed missingness and include REL-ARXIV, which is nearly complete,

*Figure 3.* F1 scores (mean ± std over 5 runs) under distribution shifts in missingness between training and test data. All models are trained with *FD-MNAR* missingness at 50%. Each panel corresponds to a dataset; each row to a model. Colored dots represent test-time F1 under *U-MCAR* with varying missingness rates: yellow = 0%, blue = 25%, green = 50%. Vertical red lines indicate the F1 achieved in the i.i.d. setting (*FD-MNAR* 50% at both train and test).

*Table 3.* Results on RelBench datasets with naturally occurring missingness. #Nodes denotes the total number of rows across all relational tables. Lower is better for MAE; higher is better for ROC-AUC. Runtime measures end-to-end wall-clock time per seed (preprocessing, text-embedding computation, 10 training epochs, validation, and test inference) on a single GPU; mean ± std across 3 seeds. Both methods use the same number of training epochs (no early stopping) and identical loaders, so the comparison isolates the cost of the MIM augmentation.

| Dataset | Missingness | #Nodes | Task | Metric | RDL | RDLmim | RDL Runtime (s) | RDLmim Runtime (s) |
|---|---|---|---|---|---|---|---|---|
| **Node regression** | | | | | | | | |
| REL-EVENT | 12.23% | 41,328,337 | user-attendance | MAE | $0.2628 \pm 0.0020$ | $\mathbf{0.2553 \pm 0.0018}$ | $1512.0 \pm 22.0$ | $1526.2 \pm 6.6$ |
| REL-TRIAL | 23.90% | 5,434,924 | study-adverse | MAE | $43.5251 \pm 0.3433$ | $\mathbf{43.1148 \pm 0.1554}$ | $531.7 \pm 7.4$ | $319.6 \pm 67.4$ |
| REL-TRIAL | 23.90% | 5,434,924 | site-success | MAE | $0.4235 \pm 0.0096$ | $\mathbf{0.3746 \pm 0.0170}$ | $368.1 \pm 1.1$ | $369.6 \pm 8.7$ |
| REL-ARXIV | 0.07% | 2,146,112 | author-publication | MAE | $\mathbf{0.5033 \pm 0.0115}$ | $0.5051 \pm 0.0041$ | $348.8 \pm 15.4$ | $310.7 \pm 22.6$ |
| **Node classification** | | | | | | | | |
| REL-F1 | 9.42% | 97,606 | driver-top3 | ROC-AUC | $0.755 \pm 0.006$ | $\mathbf{0.768 \pm 0.004}$ | $11.2 \pm 0.8$ | $10.5 \pm 0.1$ |

as a control case. Existing methods for missing node features considered above are designed for static graphs and cannot be directly applied to RelBench, whose graphs are temporal and multi-relational. We therefore use RDL (Fey et al., 2024), the SOTA GNN-based model on RelBench. The default RDL pipeline handles missing values through mean imputation; we compare it against the same architecture augmented with a simple MIM component, i.e., by concatenating each feature vector with its binary missingness mask (we refer to this as RDLmim). Table 3 shows that the MIM augmentation improves over the RDL baseline on all datasets with non-negligible naturally occurring missingness, while RDL is slightly better on the nearly complete control dataset. This supports our main conclusion: when the missingness mechanism is unknown, as in real relational databases, explicitly exposing the missingness pattern is a robust and assumption-free choice. The comparable end-to-end training and inference times further show that this augmentation scales to large graphs without introducing substantial overhead; full setup details are reported in Appendix E.1.

## 6. Conclusion and Future Work

We revisited the problem of learning GNNs under missing node features, highlighting fundamental limitations of current evaluation protocols, namely the reliance on inadequate benchmarks and oversimplified missingness mecha-

nisms. To address these issues, we introduced new datasets with dense, informative features and more realistic missingness patterns, and proposed GNNmim, a simple yet effective method that explicitly models missingness through the missing-indicator approach. Our experiments show that GNNmim is competitive with respect to more complex architectures across datasets, missingness types, and train–test shifts. This work calls for a shift towards more realistic evaluation settings and demonstrates that lightweight yet principled strategies can achieve robustness in challenging scenarios. Our study show the need for larger benchmarks specifically designed for missing features, aligning with recent calls for better graph datasets (Bechler-Speicher et al., 2025), and reveals that there remains room for developing models that are robust to diverse types of missingness.

**Limitations** This work focuses on node-level tasks (classification and regression) with missing node features, leaving other graph learning tasks, such as link prediction (Lachi et al., 2024; 2026) and graph classification (Errica et al., 2019), to future work. Finally we assume that the graph structure and training labels are observed, and do not address settings with missing edges, uncertain topology, or missing labels.

## Impact Statement

This paper focuses on methodological and evaluation aspects of learning with missing node features in Graph Neural Networks. Our primary contribution is to improve the reliability and realism of existing evaluation protocols by highlighting limitations of commonly used benchmarks and proposing more representative datasets and missingness mechanisms.

The techniques and analyses presented here are intended to support the development of more robust and trustworthy machine learning models in domains where missing data is unavoidable, such as sensor networks and healthcare. By explicitly accounting for realistic missingness patterns, this work may help reduce misleading performance assessments and unintended overconfidence in deployed systems.

We do not foresee immediate negative societal impacts arising directly from this work. As with most advances in machine learning methodology, potential downstream effects depend on the specific application context and data used. We encourage practitioners to consider domain-specific ethical constraints, particularly when working with sensitive data.

## Acknowledgments

Funded by the European Union. Views and opinions expressed are however those of the author(s) only and do not necessarily reflect those of the European Union or the European Health and Digital Executive Agency (HaDEA). Neither the European Union nor the granting authority can be held responsible for them. Grant Agreement no. 101120763 - TANGO and Grant Agreement no. 101120237 - ELIAS. The work was also partially supported by Ministero delle Imprese e del Made in Italy (IPCEI Cloud DM 27 giugno 2022 – IPCEI-CL-0000007) and European Union (Next Generation EU), and by the Research Council of Norway, project number 332645. Finally, the conducted work received support by JSPS Grant-in-Aid for Scientific Research (grant numbers 25K03231, 23H03451). This paper benefits from results obtained from the BRIDGE Program (R7-H05), implemented by the Cabinet Office, Government of Japan.

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

# A. Proofs

**Theorem 2.2.** *If $P_{\boldsymbol{\theta},\boldsymbol{\gamma},\boldsymbol{\lambda}}$ is feature-MAR and label-MAR, then (2) simplifies to*

$$\int_{\mathbf{X}^{\mathrm{miss}}} P_{\boldsymbol{\gamma}}(\boldsymbol{Y}|\boldsymbol{X}) P_{\boldsymbol{\theta}}(\mathbf{X}^{\mathrm{miss}}|\mathbf{X}^{\mathrm{obs}}). \tag{6}$$

*Proof.*

$$P_{\boldsymbol{\theta},\boldsymbol{\gamma},\boldsymbol{\lambda}}(\boldsymbol{Y}|\boldsymbol{X},\boldsymbol{M}) = P_{\boldsymbol{\lambda}}(\boldsymbol{M}|\boldsymbol{X},\boldsymbol{Y})\frac{P_{\boldsymbol{\gamma}}(\boldsymbol{Y}|\boldsymbol{X})}{P_{\boldsymbol{\gamma},\boldsymbol{\lambda}}(\boldsymbol{M}|\boldsymbol{X})} \stackrel{(4)}{=} P_{\boldsymbol{\gamma}}(\boldsymbol{Y}|\boldsymbol{X})$$

$$P_{\boldsymbol{\theta},\boldsymbol{\gamma},\boldsymbol{\lambda}}(\mathbf{X}^{miss}|\mathbf{X}^{obs},\boldsymbol{M}) = P_{\boldsymbol{\gamma},\boldsymbol{\lambda}}(\boldsymbol{M}|\mathbf{X}^{obs},\mathbf{X}^{miss})\frac{P_{\boldsymbol{\theta}}(\mathbf{X}^{miss}|\mathbf{X}^{obs})}{P_{\boldsymbol{\theta},\boldsymbol{\gamma},\boldsymbol{\lambda}}(\boldsymbol{M}|\mathbf{X}^{obs})} \stackrel{(3)}{=} P_{\boldsymbol{\theta}}(\mathbf{X}^{miss}|\mathbf{X}^{obs})$$

$\square$

**Theorem 3.2.** *Let $\mathbf{X} \in \mathbb{R}^{n \times d}$ and $\mathbf{Y} \in \mathcal{Y}^n$ be random variables, $\mathbf{M} \in \{0,1\}^{n \times d}$ be a missingness mask and $\mathbf{X}^{\mathrm{obs}}$ denotes the observed (incomplete) data. We encode the pair $(\mathbf{X}^{\mathrm{obs}}, \mathbf{M})$ with the random variable $\tilde{\mathbf{X}}$ with*

$$\tilde{X}_{ij} = \begin{cases} X_{ij}, & M_{ij} = 0, \\ ?, & M_{ij} = 1. \end{cases}$$

*Let the change in the information be defined as $\Delta := I(\mathbf{Y}; \tilde{\mathbf{X}}) - I(\mathbf{Y}; \mathbf{X})$, where $I(\cdot; \cdot)$ denotes the mutual information. Then,*

1. *If the missingness is label-MAR, then $\Delta \leq 0$.*

2. *If $\mathbf{X} \in \{0,1\}^{n \times d}$ and the missingness is U-MCAR with missingness probability $\mu$, and $s(\mathbf{X})$ is the sample sparsity as in Definition 3.1, then*
$$-nd\,\mu\,h_2\big(\mathbb{E}[s(\mathbf{X})]\big) \leq \Delta \leq 0,$$
   *where $h_2(u) = -u\log u - (1-u)\log(1-u)$.*

*Proof.* By construction $\tilde{\mathbf{X}} = g(\mathbf{X}, \mathbf{M})$ for some measurable $g$. Thus $(\mathbf{Y}) \to (\mathbf{X}, \mathbf{M}) \to \tilde{\mathbf{X}}$ is a Markov chain, and the data–processing inequality implies

$$I(\mathbf{Y}; \tilde{\mathbf{X}}) \leq I(\mathbf{Y}; \mathbf{X}, \mathbf{M}). \tag{7}$$

Moreover, for any three random elements $(A, B, C)$ we have the chain–rule identities

$$I(A; B, C) = I(A; C) + I(A; B \mid C). \tag{8}$$

**(1) Label-MAR $\Delta \leq 0$.** Assume label-MAR: $\mathbb{P}(\mathbf{M} \mid \mathbf{X}, \mathbf{Y}) = \mathbb{P}(\mathbf{M} \mid \mathbf{X})$, which is equivalent to $\mathbf{Y} \perp \mathbf{M} \mid \mathbf{X}$. Applying (8) with $(A, B, C) = (\mathbf{Y}, \mathbf{X}, \mathbf{M})$,

$$I(\mathbf{Y}; \mathbf{X}, \mathbf{M}) = I(\mathbf{Y}; \mathbf{X}) + I(\mathbf{Y}; \mathbf{M} \mid \mathbf{X}).$$

Under label-MAR, $I(\mathbf{Y}; \mathbf{M} \mid \mathbf{X}) = 0$, hence

$$I(\mathbf{Y}; \mathbf{X}, \mathbf{M}) = I(\mathbf{Y}; \mathbf{X}). \tag{9}$$

Combining (7) and (9) yields

$$I(\mathbf{Y}; \tilde{\mathbf{X}}) \leq I(\mathbf{Y}; \mathbf{X}) \quad \Longleftrightarrow \quad \Delta = I(\mathbf{Y}; \tilde{\mathbf{X}}) - I(\mathbf{Y}; \mathbf{X}) \leq 0.$$

**(2) Two-sided bound under uniform MCAR and $\alpha$-$\beta$ sparsity.** Assume uniform MCAR: $M_{ij} \sim \mathrm{Bernoulli}(1 - \mu)$ independently of $(\mathbf{X}, \mathbf{Y})$ and i.i.d. across $(i, j)$, and that $\mathbb{P}\big(s(\mathbf{X}) \geq \alpha\big) \geq \beta$, where $s(\mathbf{X}) = \frac{1}{nd}\sum_{i,j} \mathbb{I}\{X_{ij} = 0\}$.

*Upper side.* MCAR implies label-MAR, so by part (1): $\Delta \leq 0$.

*Lower side.* We start from the chain–rule identity applied to $(A, B, C) = (\mathbf{Y}, \mathbf{X}, \tilde{\mathbf{X}})$:

$$I(\mathbf{Y}; \mathbf{X}, \tilde{\mathbf{X}}) = I(\mathbf{Y}; \tilde{\mathbf{X}}) + I(\mathbf{Y}; \mathbf{X} \mid \tilde{\mathbf{X}}) = I(\mathbf{Y}; \mathbf{X}) + I(\mathbf{Y}; \tilde{\mathbf{X}} \mid \mathbf{X}).$$

Rearranging gives

$$-\Delta = I(\mathbf{Y}; \mathbf{X}) - I(\mathbf{Y}; \tilde{\mathbf{X}}) = I(\mathbf{Y}; \mathbf{X} \mid \tilde{\mathbf{X}}) - I(\mathbf{Y}; \tilde{\mathbf{X}} \mid \mathbf{X}). \tag{10}$$

The second term on the right is nonnegative, hence

$$-\Delta \leq I(\mathbf{Y}; \mathbf{X} \mid \tilde{\mathbf{X}}). \tag{11}$$

Using the bound $I(U; V \mid W) \leq H(V \mid W)$, we get

$$-\Delta \leq H(\mathbf{X} \mid \tilde{\mathbf{X}}). \tag{12}$$

Index the matrix entries by a total order $\prec$ on pairs $(i, j)$ and apply the chain rule:

$$H(\mathbf{X} \mid \tilde{\mathbf{X}}) = \sum_{(i,j)} H\big(X_{ij} \mid \tilde{\mathbf{X}}, \{X_{kl} : (k, l) \prec (i, j)\}\big).$$

Since conditioning reduces entropy,

$$H(\mathbf{X} \mid \tilde{\mathbf{X}}) \leq \sum_{i,j} H\big(X_{ij} \mid \tilde{X}_{ij}\big). \tag{13}$$

Fix $(i, j)$ and denote $\pi_{ij} = \Pr[X_{ij} = 1]$. Under uniform MCAR,

$$\Pr[\tilde{X}_{ij} = ?] = \mu, \qquad \Pr[\tilde{X}_{ij} = x] = (1 - \mu) \Pr[X_{ij} = x], \quad x \in \{0, 1\}.$$

Hence: (i) if $\tilde{X}_{ij} \in \{0, 1\}$ then $X_{ij}$ is revealed, so $H(X_{ij} \mid \tilde{X}_{ij} \in \{0, 1\}) = 0$; (ii) if $\tilde{X}_{ij} = ?$, then $\Pr[X_{ij} = 1 \mid \tilde{X}_{ij} = ?] = \pi_{ij}$ and $H(X_{ij} \mid \tilde{X}_{ij} = ?) = h_2(\pi_{ij})$. Averaging over $\tilde{X}_{ij}$ gives

$$H(X_{ij} \mid \tilde{X}_{ij}) = \mu \, h_2(\pi_{ij}). \tag{14}$$

Combining (13) and (14):

$$H(\mathbf{X} \mid \tilde{\mathbf{X}}) \leq \sum_{i,j} \mu \, h_2(\pi_{ij}) = nd\,\mu \cdot \frac{1}{nd} \sum_{i,j} h_2(\pi_{ij}) \leq nd\,\mu \cdot h_2\left(\frac{1}{nd} \sum_{i,j} \pi_{ij}\right),$$

since $h_2$ is concave. Note that

$$\frac{1}{nd} \sum_{i,j} \pi_{ij} = \frac{1}{nd} \sum_{i,j} \Pr[X_{ij} = 1] = \mathbb{E}\left[\frac{1}{nd} \sum_{i,j} \mathbb{I}\{X_{ij} = 1\}\right] = 1 - \mathbb{E}[s(\mathbf{X})].$$

Using the symmetry $h_2(u) = h_2(1 - u)$, we conclude

$$H(\mathbf{X} \mid \tilde{\mathbf{X}}) \leq nd\,\mu \cdot h_2\big(\mathbb{E}[s(\mathbf{X})]\big).$$

Combining with $-\Delta \leq H(\mathbf{X} \mid \tilde{\mathbf{X}})$ gives

$$- nd\,\mu\, h_2\big(\mathbb{E}[s(\mathbf{X})]\big) \leq \Delta \leq 0.$$

This concludes the proof. $\qquad\square$

*Remark* A.1. While loose in general, the bound becomes tight under specific conditions that align with the proof steps:

(i) If $\mathbf{Y}$ is a deterministic, injective function of $X$, then the relaxation from (11) to (12) is exact.

(ii) If the feature entries $\mathbf{X}_{i,j}$ are independent, the decomposition in (13) holds with equality.

(iii) If all $\mathbf{X}_{i,j}$ share the same marginal distribution, the bound following (14) is also tight.

While these conditions are very strong and not fully realistic, they provide useful insights into the nature of the bound. For missing data to incur a large information loss, there has to be significant mutual information to begin with. This is captured by condition (i): at the opposite extreme, if $\mathbf{Y}$ and $\mathbf{X}$ are independent, then $\Delta = 0$, and the lower bound is very loose. Conditions (ii) and (iii) are independence and uniformity conditions that allow us to express the effect of missing feature values in terms of simple summary statistics of expected sparsity. Thus, while not incorporating all potentially relevant factors, our bound focuses on the effect of sparsity and shows that, in highly sparse settings, sparsity alone can fundamentally limit the impact of missingness, independently of the task.

## B. Additional Results on Benchmarks and Proposed Datasets

This section presents the full plots of the results under the R1 regime introduced in Section 4.

Figure 4 shows the complete set of results across all datasets, whose statistics are summarized in Table 1. The top three rows correspond to the classic benchmarks (CORA, CITESEER, PUBMED). Consistently with Proposition 3.2, models maintain nearly constant F1 scores up to extremely high missingness levels ($\sim 90\%$), confirming that these benchmarks are of limited value for evaluating robustness to missing features.

The bottom four rows correspond to our proposed datasets (SYNTHETIC, AIR, ELECTRIC, TADPOLE). In these cases, performance degrades much earlier and more severely, highlighting the higher realism and difficulty of our benchmarks.

## C. More challenging datasets

In Section 3, we introduced the synthetic and real-world datasets employed in our experiments. We now provide additional details on their construction and characteristics.

SYNTHETIC   Synthetic dataset based on a Barabási–Albert graph topology. Each node is associated with five real-valued features sampled from a Gaussian distribution. Node labels are generated deterministically by applying a fixed two-layer GCN with hard-coded weights to the complete feature matrix. This construction ensures that the ground-truth labeling function is fully expressible by a GNN, allowing models to achieve near-perfect accuracy in the absence of missingness. The resulting task is a binary node classification problem, with classes separated according to structured feature combinations defined by the fixed GCN. This controlled setup provides a principled testbed to isolate and analyze the effects of different missingness mechanisms, while preserving a well-defined ground truth.

AIR   Dataset (Zheng et al., 2015) built from a network of air quality monitoring stations deployed in an urban area. Each node corresponds to a station and is associated with a set of environmental measurements. The node features include both air pollutant concentrations (CO, $NO_2$, $PM_{10}$, $O_3$, $SO_2$) and meteorological variables (`temperature`, `humidity`, `wind speed`, `wind direction`). Edges are constructed based on the geographical distance between stations, with two nodes connected if their distance is below a given threshold. The target variable is derived from the $PM_{2.5}$ concentration, which is discretized into three balanced categories (low, medium, high) according to the distribution of observed values. This formulation allows us to frame the problem as a semi-supervised node classification task with three classes.

ELECTRIC   Dataset (Birchfield et al., 2016; Baek & Birchfield, 2023) derived from a large-scale model of the Texas power grid. Nodes correspond to buses in the electrical network, each enriched with both structural and operational attributes. The node features include identifiers (`area`, `zone`), electrical measurements (`voltage magnitude`, `voltage angle`), and a topological property (`betweenness centrality`). Edges are constructed directly from the transmission lines specified in the raw grid data, connecting pairs of buses. The classification target is the nominal voltage level of each bus (`base kV`), which we discretize into three categories: low voltage ($<100$ kV), medium voltage ($100$–$200$ kV), and high voltage ($>200$ kV). This setup results in a three-class node classification problem reflecting operational conditions across the grid.

TADPOLE   The TADPOLEdataset (Zhu et al., 2019) originates from the TADPOLE challenge, which provides longitudinal clinical and imaging data for patients at risk of developing Alzheimer's disease. In our graph formulation, each node

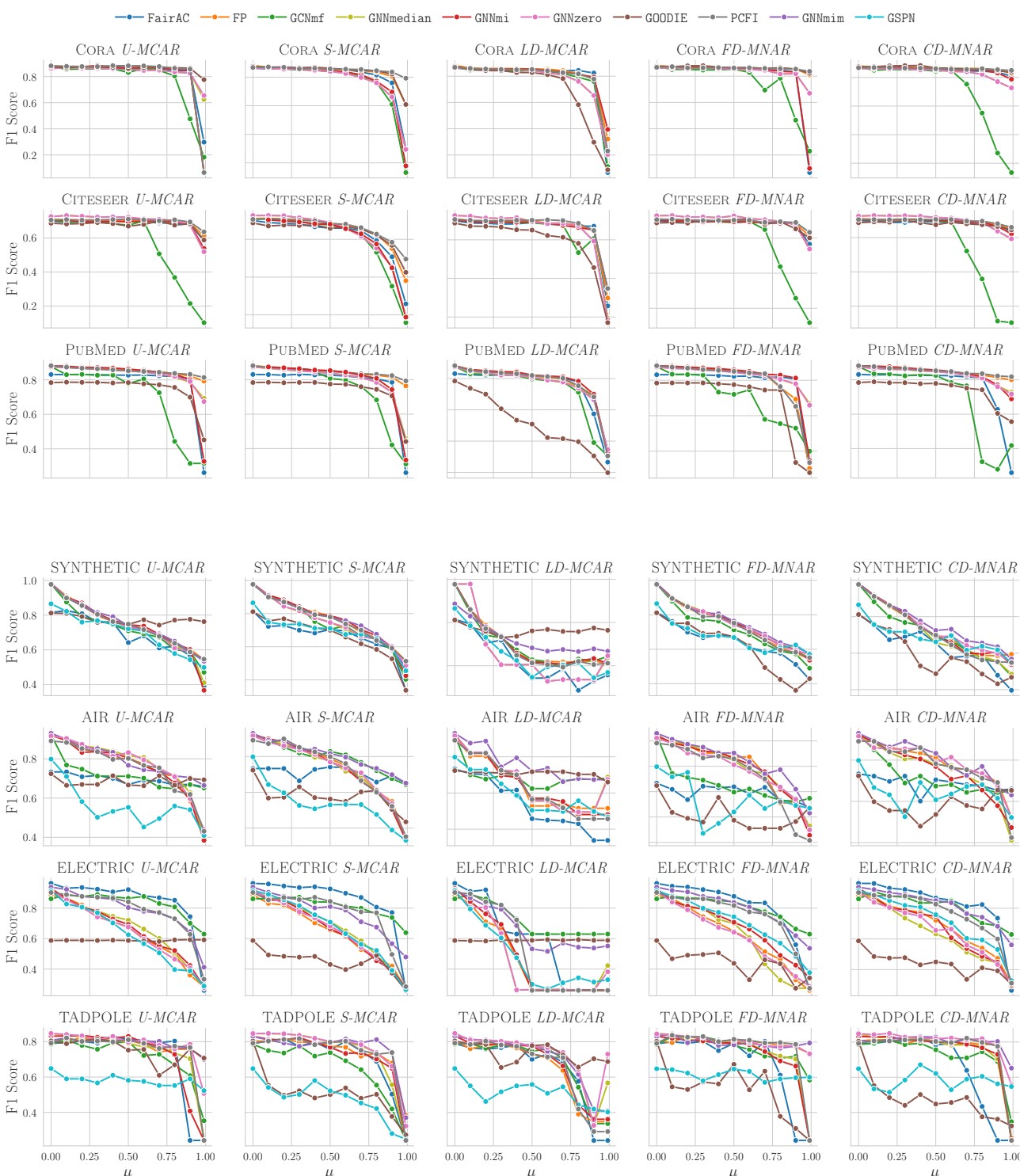

*Figure 4.* F1 score as a function of feature missingness ($\mu$) for both classic benchmarks (top three rows) and our proposed datasets (bottom four rows), under the mechanisms described in Section 4. Classic benchmarks show almost no degradation until extremely high $\mu$, while the proposed datasets reveal model weaknesses at more realistic missingness levels. Tables for numeric results are in App. G

corresponds to a patient and is associated with a set of features encompassing clinical scores, cerebrospinal fluid (CSF) biomarkers, and neuroimaging measures such as MRI- and PET-derived variables. Since the original dataset does not

provide graph connectivity, we construct edges using a $k$-nearest neighbors approach over the most informative biomarkers, so that patients with similar profiles are connected. The target variable is the diagnostic label, categorized into three classes (cognitively normal, mild cognitive impairment, Alzheimer's disease). This results in a semi-supervised node classification problem where the goal is to predict the diagnostic status of patients based on multimodal biomedical features and patient similarity structure.

Table 1 reports, for each dataset, the number of nodes, number of features, feature sparsity, and the type of features. While the number of nodes and features may seem small compared to standard benchmark graph datasets, we emphasize that using real features (as in AIR, ELECTRIC, and TADPOLE) is more realistic in the context of feature missingness. In fact, it is not meaningful to study missingness on pre-computed embeddings, since embeddings are typically high-dimensional representations mapped to wide feature spaces and are not expected to exhibit missingness in practice.

## D. Why Existing Large-Scale Node Classification Datasets Are Unsuitable for Studying Robustness to Feature Missingness

In this appendix, we provide an analysis of existing large-scale graph datasets for node classification and explain why they are not suitable for studying robustness to missing node features. Our goal is not to argue that these datasets are flawed in general, but rather that they violate key requirements that are necessary for a meaningful evaluation of GNN under feature missingness.

We focus on two fundamental prerequisites. First, node features must be dense, low-dimensional, and semantically meaningful. Feature missingness is only well-defined and realistic when features correspond to directly observed attributes whose absence has a clear interpretation. Datasets with extremely sparse representations, such as bag-of-words or TF–IDF features, already encode the absence of information by construction, making additional missingness largely inconsequential (see Section 3. Similarly, learned embeddings are ill-suited for this setting: they are artificially constructed latent representations in which semantic information is deliberately distributed across many dimensions in an overparameterized manner, so that individual features are not interpretable in isolation (Arora et al., 2016a;b). As a result, feature-level missingness in embeddings is ill-defined and does not correspond to a realistic missing-data mechanism. We argue that datasets failing to satisfy this first criterion should not be used to evaluate robustness to missing node features.

Second, suitable datasets must exhibit non-trivial predictive signal under complete information and complementary and separable contributions of node features and graph structure. If either features or structure alone are sufficient for high performance, or if one of them is largely uninformative, then performance under missingness becomes difficult to interpret, as degradation may reflect dataset artifacts rather than genuine robustness properties of the model. Moreover, a reasonable baseline performance of a standard GNN under complete information is necessary to ensure that the task itself is well-posed before introducing missingness.

Based on these criteria, we analyze a broad set of widely used benchmarks, including classical node classification datasets and more recent large-scale benchmarks proposed in the literature, such as those from the OGB collection. Table 4 evaluates whether existing large-scale node classification datasets satisfy the minimal necessary condition for studying robustness to missing node features, namely the presence of dense, low-dimensional, and semantically meaningful node attributes. We find that none of the considered benchmarks meet this requirement: node features are either highly sparse (e.g., bag-of-words or categorical encodings), embedding-based, or entirely absent. When this condition is violated, feature-level missingness is ill-defined and its effect on model performance becomes inherently uninformative, independently of dataset scale. As a result, we do not further assess separability, complementarity, or baseline performance on these datasets, as the problem of feature missingness is already ill-posed at the feature level. This observation supports our choice of moderate-scale datasets with dense, semantically meaningful features, and aligns with recent critiques of current benchmarking practices in graph learning (Bechler-Speicher et al., 2025).

## E. Experimental Details

All baseline and competitor methods are implemented using the official code released in their respective repositories, following the recommended training protocols and hyperparameter settings. For GNNmi and GNNmim, as well as for FP and PCFI, where the GNN backbone is separable from the imputation strategy, we adopt a standard GNN architecture where the convolutional layer type (Table 5), the number of layers (1-3), the learning rate ($10^{-4}$-$10^{-2}$), and the weight decay ($10^{-5}$-$10^{-3}$) are tuned via grid search on the validation set. For GCNmf, GSPN, FairAC, and GOODIE, instead, the

| | | #Features | Type of Features | Sparsity | Suitable for Miss. Analysis |
|---|---|---|---|---|---|
| | hm-categories | 35 | Cat. & Num. | 0.7 | ✗ |
| | tolokers-2 | 16 | Cat. & Num. | 0.5 | ✗ |
| GraphLand (Bazhenov et al., 2025) | Web-topics | 263 | Cat. & Num. | 0.7 | ✗ |
| | city-reviews | 37 | Cat. & Num. | 0.9 | ✗ |
| | artnet-exp | 75 | Cat. & Num. | 0.4 | ✗ |
| | pokec-regions | 56 | Cat. & Num. | 0.7 | ✗ |
| | ogbn-arxiv | 128 | Embeddings | 0.0 | ✗ |
| OGB (Hu et al., 2020) | ogbn-Products | 100 | BoW | 0.1 | ✗ |
| | ogbn-Proteins | 0 | - | - | ✗ |
| | ogbn-Mag | 128 | Embeddings | 0.0 | ✗ |
| GraphBench (Stoll et al., 2025) | Max Clique | 0 | - | - | ✗ |

*Table 4.* For each benchmark, we report the number of node features, feature type, and feature sparsity. The table assesses whether datasets satisfy the minimal necessary condition for feature-level missingness analysis, namely the presence of dense, low-dimensional, and semantically meaningful node attributes. Datasets with highly sparse representations, embedding-based features, or no node features are marked as unsuitable, as feature-level missingness is ill-defined in these settings, independently of dataset scale.

GNN backbone is an integral part of the architecture and cannot be modified without altering the method itself, so we use the backbone specified in the original implementations. All models are trained on the same data splits with early stopping to ensure a fair comparison.

*Table 5.* Best GNN encoder selected within `GNNmim`, `FP` and `PCFI` for each dataset and missingness mechanism.

| Method | Dataset | U-MCAR | S-MCAR | LD-MCAR | FD-MNAR | CD-MNAR |
|---|---|---|---|---|---|---|
| `GNNmim` | SYNTHETIC | GCN | GCN | GraphSAGE | GCN | GCN |
| | AIR | GraphSAGE | GraphSAGE | GraphSAGE | GraphSAGE | GraphSAGE |
| | ELECTRIC | GIN | GIN | GraphSAGE | GIN | GIN |
| | TADPOLE | GCN | GraphSAGE | GraphSAGE | GraphSAGE | GCN |
| `FP` | SYNTHETIC | GCN | GCN | GCN | GCN | GCN |
| | AIR | GraphSAGE | GraphSAGE | GraphSAGE | GraphSAGE | GraphSAGE |
| | ELECTRIC | GraphSAGE | GraphSAGE | GCN | GraphSAGE | GCN |
| | TADPOLE | GCN | GCN | GCN | GCN | GCN |
| `PCFI` | SYNTHETIC | GCN | GCN | GCN | GraphSAGE | GCN |
| | AIR | GraphSAGE | GraphSAGE | GraphSAGE | GraphSAGE | GraphSAGE |
| | ELECTRIC | GraphSAGE | GCN | GraphSAGE | GraphSAGE | GraphSAGE |
| | TADPOLE | GCN | GCN | GCN | GraphSAGE | GCN |

### E.1. RelBench Experimental Setup

The setup described above applies to the experiments in Section 5 (Q1–Q3). For the RelBench experiments answering Q4, we use the official `RDL` pipeline of Fey et al. (2024) as our backbone, with two graph layers, hidden dimension 128, sum aggregation, batch normalization, and the Adam optimizer with learning rate $5 \cdot 10^{-3}$. Each training run consists of 10 epochs with batch size 512 and neighbor sampling fanouts $[128, 128]$; no early stopping is applied, so both `RDL` and `GNNmim` perform exactly the same number of optimization steps per seed. At each epoch we evaluate on the validation set and retain the model with the best validation metric for test-time evaluation. Text features are encoded with a GloVe-based embedder, recomputed at each seed (no caching) to include their cost in the runtime measurement.

For the `GNNmim` variant, we augment each numeric and textual feature with a binary mask indicating missing entries before constructing the heterogeneous graph; all other components of the pipeline are identical to the baseline. Runtimes in Table 3 are measured as end-to-end wall-clock time per seed, starting from the model-specific preprocessing and ending after test inference, on a single NVIDIA GeForce RTX 4090 (24GB) GPU. We report mean and standard deviation across 3 seeds $(0, 1, 2)$. GPU memory usage is essentially identical for the two variants, as `GNNmim` only adds one binary column per

masked feature.

## F. Scaling the Synthetic Dataset

In this section, we analyze what happens when either the number of features or the number of nodes in the synthetic dataset is increased. To this end, we constructed three additional synthetic datasets (SYNTHETIC2, SYNTHETIC3, SYNTHETIC4) following the same design principles as SYNTHETIC. Table 6 reports their statistics.

As shown in Figure 5, the behavior of the models in this larger-scale setting is consistent with the one observed in our original setup. In this case, we experimented with the *uniform random missingness* mechanism, and we observe a monotonic decrease in performance for all models as the missingness rate $\mu$ increases. This confirms that dataset size does not affect the overall trend of performance degradation under feature missingness.

To further support this point, we also report the runtime and GPU memory consumption of all models on both the main synthetic dataset (SYNTHETIC) and its larger-scale counterpart (SYNTHETIC3), which features an increased number of features. As shown in Table 7, the runtime and memory requirements remain substantially stable across datasets, with negligible variations between models. This behavior confirms that our approach scales efficiently with the dataset size, as it only involves a standard GNN architecture augmented with a simple MIM mask concatenated to the input features, introducing minimal computational overhead.

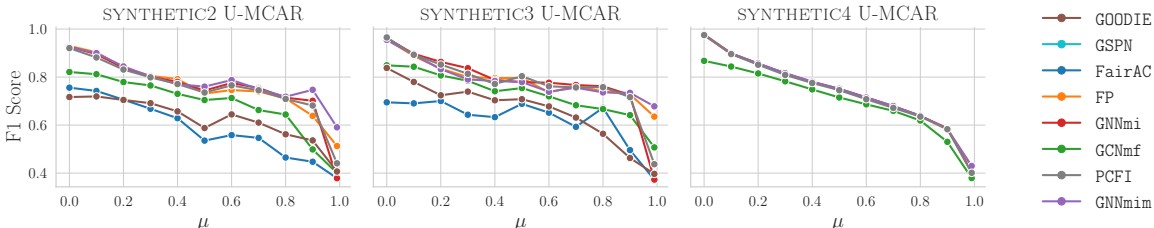

*Figure 5.* F1 score as a function of feature missingness ($\mu$) for additional synthetic datasets generated with the same procedure as SYNTHETIC, but with either an increased number of nodes or features. For SYNTHETIC4, the `FairAC` model is not reported since training exceeded the 12-hour time limit, while `GOODIE` is excluded due to out-of-memory errors.

*Table 6.* Datasets information.

| Dataset | #Nodes | #Features | Sparsity ↓ | Type of features |
|---|---|---|---|---|
| SYNTHETIC | 1000 | 5 | 0.0000 | Gaussian |
| SYNTHETIC2 | 1000 | 20 | 0.0000 | Gaussian |
| SYNTHETIC3 | 1000 | 50 | 0.0000 | Gaussian |
| SYNTHETIC4 | 50000 | 5 | 0.0000 | Gaussian |

*Table 7.* Runtime and GPU peak memory consumption for the main synthetic dataset (SYNTHETIC) and the scaled version (SYNTHETIC3). Each value corresponds to the average across all missingness levels under the UMCAR mechanism.

| Model | SYNTHETIC | | SYNTHETIC4 | |
|---|---|---|---|---|
| | Runtime [s] ↓ | GPU Mem [GB] ↓ | Runtime [s] ↓ | GPU Mem [GB] ↓ |
| GNNmi | 1.7 | 0.03 | 5.3 | 0.78 |
| GNNzero | 1.6 | 0.03 | 5.0 | 0.77 |
| GNNmedian | 1.6 | 0.03 | 5.0 | 0.77 |
| GNNmim | 1.8 | 0.03 | 6.3 | 0.77 |
| GCNmf | 4.5 | 0.02 | 28.0 | 0.53 |
| FP | 1.5 | 0.02 | 5.3 | 0.77 |
| PCFI | 1.8 | 0.02 | 5.2 | 0.77 |
| FairAC | 3.9 | 0.04 | – | – |
| GSPN | 55.0 | 0.03 | 150.0 | 0.84 |
| GOODIE | 2.3 | 0.06 | – | – |

## G. Complete Result Tables – R1 Regime

*Table 8.* F1 scores for CORA under mechanism *U-MCAR* and varying $\mu$ (GSPNis not reported as it is not designed for categorical features).

| $\mu$ | GOODIE | FairAC | FP | GNNmi | GCNmf | PCFI | GNNzero | GNNmedian |
|---|---|---|---|---|---|---|---|---|
| 0.00 | 0.875 (± 0.00) | 0.863 (± 0.01) | 0.882 (± 0.00) | 0.873 (± 0.00) | 0.875 (± 0.00) | **0.882 (± 0.00)** | 0.862 (± 0.02) | 0.862 (± 0.02) |
| 0.10 | 0.867 (± 0.00) | 0.866 (± 0.00) | 0.877 (± 0.00) | 0.876 (± 0.00) | 0.856 (± 0.00) | **0.878 (± 0.00)** | 0.868 (± 0.01) | 0.868 (± 0.01) |
| 0.20 | 0.875 (± 0.00) | 0.862 (± 0.00) | **0.878 (± 0.00)** | 0.873 (± 0.00) | 0.858 (± 0.00) | 0.877 (± 0.00) | 0.864 (± 0.02) | 0.864 (± 0.02) |
| 0.30 | 0.873 (± 0.00) | 0.865 (± 0.00) | 0.881 (± 0.00) | **0.885 (± 0.00)** | 0.860 (± 0.00) | 0.876 (± 0.00) | 0.863 (± 0.01) | 0.863 (± 0.01) |
| 0.40 | 0.869 (± 0.00) | 0.857 (± 0.00) | 0.878 (± 0.00) | 0.873 (± 0.00) | 0.860 (± 0.00) | **0.884 (± 0.00)** | 0.860 (± 0.02) | 0.860 (± 0.02) |
| 0.50 | 0.861 (± 0.00) | 0.856 (± 0.00) | **0.882 (± 0.00)** | 0.867 (± 0.00) | 0.831 (± 0.00) | 0.882 (± 0.00) | 0.856 (± 0.01) | 0.856 (± 0.01) |
| 0.60 | 0.866 (± 0.00) | 0.847 (± 0.00) | **0.882 (± 0.00)** | 0.871 (± 0.00) | 0.862 (± 0.00) | 0.881 (± 0.00) | 0.847 (± 0.01) | 0.847 (± 0.01) |
| 0.70 | 0.866 (± 0.00) | 0.858 (± 0.00) | 0.869 (± 0.00) | 0.865 (± 0.00) | 0.847 (± 0.00) | **0.877 (± 0.00)** | 0.849 (± 0.01) | 0.849 (± 0.01) |
| 0.80 | **0.868 (± 0.00)** | 0.843 (± 0.00) | 0.864 (± 0.00) | 0.854 (± 0.00) | 0.805 (± 0.00) | 0.863 (± 0.00) | 0.835 (± 0.01) | 0.835 (± 0.01) |
| 0.90 | **0.864 (± 0.00)** | 0.845 (± 0.00) | 0.860 (± 0.00) | 0.848 (± 0.00) | 0.476 (± 0.00) | 0.856 (± 0.00) | 0.826 (± 0.00) | 0.826 (± 0.00) |
| 0.99 | **0.776 (± 0.00)** | 0.298 (± 0.00) | 0.066 (± 0.00) | 0.066 (± 0.00) | 0.183 (± 0.00) | 0.065 (± 0.00) | 0.655 (± 0.03) | 0.625 (± 0.02) |

*Table 9.* F1 scores for CORA under mechanism *S-MCAR* and varying $\mu$ (GSPNis not reported as it is not designed for categorical features).

| $\mu$ | GOODIE | FairAC | FP | GNNmi | GCNmf | PCFI | GNNzero | GNNmedian |
|---|---|---|---|---|---|---|---|---|
| 0.00 | 0.875 (± 0.00) | 0.863 (± 0.01) | **0.882 (± 0.00)** | 0.872 (± 0.00) | 0.875 (± 0.00) | 0.868 (± 0.00) | 0.862 (± 0.02) | 0.862 (± 0.02) |
| 0.10 | 0.868 (± 0.00) | 0.857 (± 0.00) | 0.869 (± 0.00) | 0.862 (± 0.00) | 0.869 (± 0.00) | **0.872 (± 0.00)** | 0.862 (± 0.02) | 0.862 (± 0.02) |
| 0.20 | **0.872 (± 0.00)** | 0.860 (± 0.00) | 0.863 (± 0.00) | 0.863 (± 0.00) | 0.858 (± 0.00) | 0.869 (± 0.00) | 0.856 (± 0.02) | 0.856 (± 0.02) |
| 0.30 | **0.865 (± 0.00)** | 0.850 (± 0.00) | 0.854 (± 0.00) | 0.855 (± 0.00) | 0.852 (± 0.00) | 0.858 (± 0.00) | 0.857 (± 0.02) | 0.857 (± 0.02) |
| 0.40 | **0.870 (± 0.00)** | 0.857 (± 0.00) | 0.859 (± 0.00) | 0.848 (± 0.00) | 0.848 (± 0.00) | 0.862 (± 0.00) | 0.849 (± 0.02) | 0.849 (± 0.01) |
| 0.50 | **0.862 (± 0.00)** | 0.854 (± 0.00) | 0.854 (± 0.00) | 0.844 (± 0.00) | 0.839 (± 0.00) | 0.858 (± 0.00) | 0.841 (± 0.01) | 0.841 (± 0.01) |
| 0.60 | 0.855 (± 0.00) | 0.854 (± 0.00) | 0.853 (± 0.00) | 0.837 (± 0.00) | 0.837 (± 0.00) | **0.856 (± 0.00)** | 0.826 (± 0.01) | 0.826 (± 0.01) |
| 0.70 | 0.847 (± 0.00) | 0.836 (± 0.00) | 0.845 (± 0.00) | 0.817 (± 0.00) | 0.807 (± 0.00) | **0.854 (± 0.00)** | 0.798 (± 0.02) | 0.798 (± 0.02) |
| 0.80 | **0.845 (± 0.00)** | 0.815 (± 0.00) | 0.836 (± 0.00) | 0.772 (± 0.00) | 0.764 (± 0.00) | 0.845 (± 0.00) | 0.760 (± 0.02) | 0.760 (± 0.02) |
| 0.90 | 0.822 (± 0.00) | 0.760 (± 0.00) | 0.806 (± 0.00) | 0.696 (± 0.00) | 0.610 (± 0.00) | **0.836 (± 0.00)** | 0.661 (± 0.02) | 0.661 (± 0.02) |
| 0.99 | 0.609 (± 0.00) | 0.300 (± 0.00) | 0.606 (± 0.00) | 0.179 (± 0.00) | 0.132 (± 0.00) | **0.792 (± 0.00)** | 0.294 (± 0.05) | 0.294 (± 0.05) |

*Table 10.* F1 scores for CORA under mechanism *LD-MCAR* and varying $\mu$ (GSPNis not reported as it is not designed for categorical features).

| $\mu$ | GOODIE | FairAC | FP | GNNmi | GCNmf | PCFI | GNNzero | GNNmedian |
|---|---|---|---|---|---|---|---|---|
| 0.00 | 0.875 (± 0.00) | 0.863 (± 0.01) | **0.882 (± 0.00)** | 0.873 (± 0.00) | 0.875 (± 0.00) | 0.868 (± 0.00) | 0.862 (± 0.02) | 0.862 (± 0.02) |
| 0.10 | 0.852 (± 0.00) | 0.851 (± 0.00) | **0.862 (± 0.00)** | 0.857 (± 0.00) | 0.846 (± 0.00) | 0.860 (± 0.00) | 0.858 (± 0.02) | 0.858 (± 0.02) |
| 0.20 | 0.843 (± 0.00) | 0.854 (± 0.00) | **0.859 (± 0.00)** | 0.854 (± 0.00) | 0.850 (± 0.00) | 0.855 (± 0.00) | 0.854 (± 0.02) | 0.854 (± 0.02) |
| 0.30 | 0.843 (± 0.00) | 0.856 (± 0.00) | **0.859 (± 0.00)** | 0.855 (± 0.00) | 0.846 (± 0.00) | 0.852 (± 0.00) | 0.853 (± 0.02) | 0.853 (± 0.02) |
| 0.40 | 0.828 (± 0.00) | 0.854 (± 0.00) | **0.858 (± 0.00)** | 0.853 (± 0.00) | 0.838 (± 0.00) | 0.849 (± 0.00) | 0.849 (± 0.02) | 0.849 (± 0.02) |
| 0.50 | 0.828 (± 0.00) | 0.854 (± 0.00) | 0.855 (± 0.00) | **0.855 (± 0.00)** | 0.848 (± 0.00) | 0.852 (± 0.00) | 0.844 (± 0.02) | 0.844 (± 0.02) |
| 0.60 | 0.812 (± 0.00) | 0.847 (± 0.00) | **0.853 (± 0.00)** | 0.844 (± 0.00) | 0.837 (± 0.00) | 0.841 (± 0.00) | 0.825 (± 0.02) | 0.825 (± 0.02) |
| 0.70 | 0.782 (± 0.00) | 0.841 (± 0.00) | **0.842 (± 0.00)** | 0.831 (± 0.00) | 0.822 (± 0.00) | 0.827 (± 0.00) | 0.810 (± 0.02) | 0.810 (± 0.02) |
| 0.80 | 0.584 (± 0.00) | **0.844 (± 0.00)** | 0.822 (± 0.00) | 0.815 (± 0.00) | 0.792 (± 0.00) | 0.818 (± 0.00) | 0.761 (± 0.01) | 0.761 (± 0.01) |
| 0.90 | 0.297 (± 0.00) | **0.824 (± 0.00)** | 0.777 (± 0.00) | 0.793 (± 0.00) | 0.760 (± 0.00) | 0.778 (± 0.00) | 0.653 (± 0.02) | 0.654 (± 0.02) |
| 0.99 | 0.088 (± 0.00) | 0.066 (± 0.00) | 0.322 (± 0.00) | **0.395 (± 0.00)** | 0.113 (± 0.00) | 0.231 (± 0.00) | 0.204 (± 0.03) | 0.204 (± 0.03) |

*Table 11.* F1 scores for CORA under mechanism *FD-MNAR* and varying $\mu$ (GSPN is not reported as it is not designed for categorical features).

| $\mu$ | GOODIE | FairAC | FP | GNNmi | GCNmf | PCFI | GNNzero | GNNmedian |
|---|---|---|---|---|---|---|---|---|
| 0.00 | 0.875 ($\pm$ 0.00) | 0.863 ($\pm$ 0.01) | **0.882** ($\pm$ **0.00**) | 0.873 ($\pm$ 0.00) | 0.875 ($\pm$ 0.00) | 0.868 ($\pm$ 0.00) | 0.864 ($\pm$ 0.02) | 0.864 ($\pm$ 0.02) |
| 0.10 | 0.872 ($\pm$ 0.01) | 0.862 ($\pm$ 0.01) | **0.873** ($\pm$ **0.01**) | 0.868 ($\pm$ 0.01) | 0.851 ($\pm$ 0.01) | 0.873 ($\pm$ 0.00) | 0.862 ($\pm$ 0.02) | 0.862 ($\pm$ 0.02) |
| 0.20 | **0.879** ($\pm$ **0.00**) | 0.870 ($\pm$ 0.01) | 0.874 ($\pm$ 0.00) | 0.865 ($\pm$ 0.01) | 0.853 ($\pm$ 0.01) | 0.863 ($\pm$ 0.01) | 0.858 ($\pm$ 0.01) | 0.858 ($\pm$ 0.01) |
| 0.30 | **0.880** ($\pm$ **0.00**) | 0.864 ($\pm$ 0.01) | 0.869 ($\pm$ 0.00) | 0.867 ($\pm$ 0.01) | 0.847 ($\pm$ 0.01) | 0.864 ($\pm$ 0.01) | 0.864 ($\pm$ 0.01) | 0.864 ($\pm$ 0.01) |
| 0.40 | **0.869** ($\pm$ **0.01**) | 0.855 ($\pm$ 0.01) | 0.864 ($\pm$ 0.01) | 0.856 ($\pm$ 0.01) | 0.849 ($\pm$ 0.00) | 0.866 ($\pm$ 0.01) | 0.858 ($\pm$ 0.02) | 0.858 ($\pm$ 0.02) |
| 0.50 | 0.865 ($\pm$ 0.01) | 0.860 ($\pm$ 0.01) | **0.866** ($\pm$ **0.01**) | 0.859 ($\pm$ 0.01) | 0.854 ($\pm$ 0.01) | 0.863 ($\pm$ 0.01) | 0.854 ($\pm$ 0.02) | 0.854 ($\pm$ 0.02) |
| 0.60 | **0.866** ($\pm$ **0.01**) | 0.853 ($\pm$ 0.01) | 0.865 ($\pm$ 0.01) | 0.863 ($\pm$ 0.01) | 0.829 ($\pm$ 0.02) | 0.864 ($\pm$ 0.01) | 0.851 ($\pm$ 0.01) | 0.851 ($\pm$ 0.01) |
| 0.70 | 0.859 ($\pm$ 0.01) | 0.847 ($\pm$ 0.00) | **0.862** ($\pm$ **0.01**) | 0.853 ($\pm$ 0.00) | 0.695 ($\pm$ 0.14) | 0.860 ($\pm$ 0.00) | 0.846 ($\pm$ 0.01) | 0.846 ($\pm$ 0.01) |
| 0.80 | **0.865** ($\pm$ **0.01**) | 0.845 ($\pm$ 0.01) | 0.861 ($\pm$ 0.01) | 0.837 ($\pm$ 0.00) | 0.785 ($\pm$ 0.05) | 0.857 ($\pm$ 0.01) | 0.817 ($\pm$ 0.02) | 0.817 ($\pm$ 0.02) |
| 0.90 | 0.854 ($\pm$ 0.01) | 0.833 ($\pm$ 0.01) | **0.855** ($\pm$ **0.00**) | 0.833 ($\pm$ 0.00) | 0.465 ($\pm$ 0.21) | 0.854 ($\pm$ 0.01) | 0.819 ($\pm$ 0.01) | 0.819 ($\pm$ 0.01) |
| 0.99 | 0.822 ($\pm$ 0.01) | 0.066 ($\pm$ 0.00) | 0.810 ($\pm$ 0.02) | 0.098 ($\pm$ 0.01) | 0.230 ($\pm$ 0.05) | **0.837** ($\pm$ **0.02**) | 0.670 ($\pm$ 0.02) | 0.670 ($\pm$ 0.02) |

*Table 12.* F1 scores for CORA under mechanism *CD-MNAR* and varying $\mu$ (GSPN is not reported as it is not designed for categorical features).

| $\mu$ | GOODIE | FairAC | FP | GNNmi | GCNmf | PCFI | GNNzero | GNNmedian |
|---|---|---|---|---|---|---|---|---|
| 0.00 | 0.875 ($\pm$ 0.00) | 0.863 ($\pm$ 0.01) | **0.882** ($\pm$ **0.00**) | 0.873 ($\pm$ 0.00) | 0.875 ($\pm$ 0.00) | 0.868 ($\pm$ 0.00) | 0.863 ($\pm$ 0.02) | 0.863 ($\pm$ 0.02) |
| 0.10 | **0.875** ($\pm$ **0.00**) | 0.864 ($\pm$ 0.01) | 0.870 ($\pm$ 0.01) | 0.862 ($\pm$ 0.01) | 0.850 ($\pm$ 0.00) | 0.869 ($\pm$ 0.01) | 0.863 ($\pm$ 0.02) | 0.863 ($\pm$ 0.02) |
| 0.20 | **0.881** ($\pm$ **0.01**) | 0.865 ($\pm$ 0.00) | 0.874 ($\pm$ 0.01) | 0.868 ($\pm$ 0.01) | 0.856 ($\pm$ 0.01) | 0.869 ($\pm$ 0.01) | 0.860 ($\pm$ 0.02) | 0.860 ($\pm$ 0.02) |
| 0.30 | **0.882** ($\pm$ **0.00**) | 0.858 ($\pm$ 0.00) | 0.873 ($\pm$ 0.00) | 0.871 ($\pm$ 0.01) | 0.854 ($\pm$ 0.00) | 0.866 ($\pm$ 0.01) | 0.860 ($\pm$ 0.02) | 0.860 ($\pm$ 0.02) |
| 0.40 | **0.884** ($\pm$ **0.01**) | 0.862 ($\pm$ 0.01) | 0.870 ($\pm$ 0.00) | 0.864 ($\pm$ 0.00) | 0.853 ($\pm$ 0.01) | 0.865 ($\pm$ 0.01) | 0.853 ($\pm$ 0.02) | 0.853 ($\pm$ 0.02) |
| 0.50 | 0.867 ($\pm$ 0.01) | 0.852 ($\pm$ 0.01) | **0.867** ($\pm$ **0.00**) | 0.861 ($\pm$ 0.00) | 0.844 ($\pm$ 0.02) | 0.861 ($\pm$ 0.01) | 0.855 ($\pm$ 0.02) | 0.855 ($\pm$ 0.02) |
| 0.60 | **0.864** ($\pm$ **0.00**) | 0.847 ($\pm$ 0.00) | 0.860 ($\pm$ 0.01) | 0.856 ($\pm$ 0.00) | 0.849 ($\pm$ 0.01) | 0.857 ($\pm$ 0.01) | 0.842 ($\pm$ 0.02) | 0.842 ($\pm$ 0.02) |
| 0.70 | 0.860 ($\pm$ 0.01) | 0.845 ($\pm$ 0.01) | **0.864** ($\pm$ **0.01**) | 0.852 ($\pm$ 0.01) | 0.753 ($\pm$ 0.12) | 0.856 ($\pm$ 0.01) | 0.840 ($\pm$ 0.02) | 0.840 ($\pm$ 0.02) |
| 0.80 | 0.853 ($\pm$ 0.01) | 0.844 ($\pm$ 0.02) | **0.862** ($\pm$ **0.01**) | 0.852 ($\pm$ 0.01) | 0.551 ($\pm$ 0.10) | 0.861 ($\pm$ 0.01) | 0.822 ($\pm$ 0.03) | 0.822 ($\pm$ 0.03) |
| 0.90 | 0.848 ($\pm$ 0.01) | 0.835 ($\pm$ 0.01) | 0.852 ($\pm$ 0.00) | 0.831 ($\pm$ 0.01) | 0.271 ($\pm$ 0.23) | **0.855** ($\pm$ **0.01**) | 0.771 ($\pm$ 0.03) | 0.771 ($\pm$ 0.03) |
| 0.99 | 0.836 ($\pm$ 0.01) | 0.810 ($\pm$ 0.01) | 0.828 ($\pm$ 0.01) | 0.788 ($\pm$ 0.02) | 0.135 ($\pm$ 0.05) | **0.849** ($\pm$ **0.01**) | 0.727 ($\pm$ 0.04) | 0.725 ($\pm$ 0.03) |

*Table 13.* F1 scores for CITESEER under mechanism *U-MCAR* and varying $\mu$ (GSPN is not reported as it is not designed for categorical features).

| $\mu$ | GOODIE | FairAC | FP | GNNmi | GCNmf | PCFI | GNNzero | GNNmedian |
|---|---|---|---|---|---|---|---|---|
| 0.00 | 0.687 ($\pm$ 0.00) | 0.700 ($\pm$ 0.00) | 0.710 ($\pm$ 0.02) | 0.704 ($\pm$ 0.02) | 0.707 ($\pm$ 0.00) | 0.706 ($\pm$ 0.02) | **0.726** ($\pm$ **0.02**) | 0.726 ($\pm$ 0.02) |
| 0.10 | 0.682 ($\pm$ 0.00) | 0.693 ($\pm$ 0.00) | 0.707 ($\pm$ 0.00) | 0.705 ($\pm$ 0.00) | 0.692 ($\pm$ 0.00) | 0.708 ($\pm$ 0.00) | **0.732** ($\pm$ **0.02**) | 0.732 ($\pm$ 0.02) |
| 0.20 | 0.684 ($\pm$ 0.00) | 0.693 ($\pm$ 0.00) | 0.706 ($\pm$ 0.00) | 0.695 ($\pm$ 0.00) | 0.698 ($\pm$ 0.00) | 0.705 ($\pm$ 0.00) | **0.728** ($\pm$ **0.02**) | 0.728 ($\pm$ 0.02) |
| 0.30 | 0.691 ($\pm$ 0.00) | 0.691 ($\pm$ 0.00) | 0.705 ($\pm$ 0.00) | 0.696 ($\pm$ 0.00) | 0.697 ($\pm$ 0.00) | 0.706 ($\pm$ 0.00) | **0.723** ($\pm$ **0.03**) | 0.723 ($\pm$ 0.03) |
| 0.40 | 0.685 ($\pm$ 0.00) | 0.700 ($\pm$ 0.00) | 0.706 ($\pm$ 0.00) | 0.698 ($\pm$ 0.00) | 0.684 ($\pm$ 0.00) | 0.708 ($\pm$ 0.00) | **0.724** ($\pm$ **0.02**) | 0.724 ($\pm$ 0.02) |
| 0.50 | 0.669 ($\pm$ 0.00) | 0.697 ($\pm$ 0.00) | 0.702 ($\pm$ 0.00) | 0.695 ($\pm$ 0.00) | 0.675 ($\pm$ 0.00) | 0.711 ($\pm$ 0.00) | **0.722** ($\pm$ **0.02**) | 0.722 ($\pm$ 0.02) |
| 0.60 | 0.680 ($\pm$ 0.00) | 0.695 ($\pm$ 0.00) | 0.697 ($\pm$ 0.00) | 0.699 ($\pm$ 0.00) | 0.700 ($\pm$ 0.00) | 0.707 ($\pm$ 0.00) | **0.712** ($\pm$ **0.02**) | 0.712 ($\pm$ 0.02) |
| 0.70 | 0.699 ($\pm$ 0.00) | 0.688 ($\pm$ 0.00) | 0.694 ($\pm$ 0.00) | 0.700 ($\pm$ 0.00) | 0.507 ($\pm$ 0.00) | 0.701 ($\pm$ 0.00) | **0.710** ($\pm$ **0.02**) | 0.710 ($\pm$ 0.02) |
| 0.80 | 0.675 ($\pm$ 0.00) | 0.687 ($\pm$ 0.00) | 0.694 ($\pm$ 0.00) | 0.696 ($\pm$ 0.00) | 0.368 ($\pm$ 0.00) | **0.707** ($\pm$ **0.00**) | 0.701 ($\pm$ 0.01) | 0.701 ($\pm$ 0.01) |
| 0.90 | 0.684 ($\pm$ 0.00) | 0.680 ($\pm$ 0.00) | 0.686 ($\pm$ 0.00) | 0.680 ($\pm$ 0.00) | 0.215 ($\pm$ 0.00) | **0.694** ($\pm$ **0.00**) | 0.678 ($\pm$ 0.02) | 0.678 ($\pm$ 0.02) |
| 0.99 | 0.588 ($\pm$ 0.00) | 0.584 ($\pm$ 0.00) | 0.613 ($\pm$ 0.00) | 0.539 ($\pm$ 0.00) | 0.102 ($\pm$ 0.00) | **0.636** ($\pm$ **0.00**) | 0.519 ($\pm$ 0.03) | 0.519 ($\pm$ 0.03) |

*Table 14.* F1 scores for CITESEER under mechanism *S-MCAR* and varying $\mu$ (GSPN is not reported as it is not designed for categorical features).

| $\mu$ | GOODIE | FairAC | FP | GNNmi | GCNmf | PCFI | GNNzero | GNNmedian |
|---|---|---|---|---|---|---|---|---|
| 0.00 | 0.687 ($\pm$ 0.00) | 0.700 ($\pm$ 0.00) | 0.710 ($\pm$ 0.02) | - | 0.707 ($\pm$ 0.00) | 0.706 ($\pm$ 0.02) | **0.726** ($\pm$ **0.02**) | 0.726 ($\pm$ 0.02) |
| 0.10 | 0.670 ($\pm$ 0.00) | 0.688 ($\pm$ 0.00) | 0.711 ($\pm$ 0.00) | 0.703 ($\pm$ 0.00) | 0.708 ($\pm$ 0.00) | 0.708 ($\pm$ 0.00) | **0.726** ($\pm$ **0.03**) | 0.726 ($\pm$ 0.03) |
| 0.20 | 0.675 ($\pm$ 0.00) | 0.685 ($\pm$ 0.00) | 0.707 ($\pm$ 0.00) | 0.697 ($\pm$ 0.00) | 0.707 ($\pm$ 0.00) | 0.706 ($\pm$ 0.00) | **0.725** ($\pm$ **0.03**) | 0.725 ($\pm$ 0.03) |
| 0.30 | 0.673 ($\pm$ 0.00) | 0.681 ($\pm$ 0.00) | 0.705 ($\pm$ 0.00) | 0.692 ($\pm$ 0.00) | 0.693 ($\pm$ 0.00) | 0.701 ($\pm$ 0.00) | **0.714** ($\pm$ **0.02**) | 0.714 ($\pm$ 0.02) |
| 0.40 | 0.677 ($\pm$ 0.00) | 0.667 ($\pm$ 0.00) | 0.698 ($\pm$ 0.00) | 0.682 ($\pm$ 0.00) | 0.682 ($\pm$ 0.00) | 0.698 ($\pm$ 0.00) | **0.704** ($\pm$ **0.03**) | 0.704 ($\pm$ 0.03) |
| 0.50 | 0.658 ($\pm$ 0.00) | 0.659 ($\pm$ 0.00) | 0.685 ($\pm$ 0.00) | 0.680 ($\pm$ 0.00) | 0.676 ($\pm$ 0.00) | 0.683 ($\pm$ 0.00) | **0.689** ($\pm$ **0.03**) | 0.689 ($\pm$ 0.03) |
| 0.60 | 0.667 ($\pm$ 0.00) | 0.659 ($\pm$ 0.00) | 0.676 ($\pm$ 0.00) | 0.656 ($\pm$ 0.00) | 0.659 ($\pm$ 0.00) | **0.680** ($\pm$ **0.00**) | 0.659 ($\pm$ 0.02) | 0.659 ($\pm$ 0.02) |
| 0.70 | 0.655 ($\pm$ 0.00) | 0.646 ($\pm$ 0.00) | 0.656 ($\pm$ 0.00) | 0.629 ($\pm$ 0.00) | 0.624 ($\pm$ 0.00) | **0.662** ($\pm$ **0.00**) | 0.617 ($\pm$ 0.02) | 0.617 ($\pm$ 0.02) |
| 0.80 | 0.621 ($\pm$ 0.00) | 0.593 ($\pm$ 0.00) | **0.629** ($\pm$ **0.00**) | 0.575 ($\pm$ 0.00) | 0.531 ($\pm$ 0.00) | 0.628 ($\pm$ 0.00) | 0.553 ($\pm$ 0.03) | 0.553 ($\pm$ 0.03) |
| 0.90 | 0.568 ($\pm$ 0.00) | 0.508 ($\pm$ 0.00) | 0.552 ($\pm$ 0.00) | 0.449 ($\pm$ 0.00) | 0.352 ($\pm$ 0.00) | **0.584** ($\pm$ **0.00**) | 0.455 ($\pm$ 0.03) | 0.455 ($\pm$ 0.03) |
| 0.99 | 0.425 ($\pm$ 0.00) | 0.258 ($\pm$ 0.00) | 0.381 ($\pm$ 0.00) | 0.188 ($\pm$ 0.00) | 0.159 ($\pm$ 0.00) | **0.495** ($\pm$ **0.00**) | 0.186 ($\pm$ 0.01) | 0.186 ($\pm$ 0.01) |

*Table 15.* F1 scores for CITESEER under mechanism *LD-MCAR* and varying $\mu$ (GSPN is not reported as it is not designed for categorical features).

| $\mu$ | GOODIE | FairAC | FP | GNNmi | GCNmf | PCFI | GNNzero | GNNmedian |
|---|---|---|---|---|---|---|---|---|
| 0.00 | 0.687 ($\pm$ 0.00) | 0.700 ($\pm$ 0.00) | 0.710 ($\pm$ 0.02) | 0.704 ($\pm$ 0.02) | 0.707 ($\pm$ 0.00) | 0.706 ($\pm$ 0.02) | **0.726** ($\pm$ **0.02**) | 0.726 ($\pm$ 0.02) |
| 0.10 | 0.671 ($\pm$ 0.00) | 0.687 ($\pm$ 0.00) | 0.698 ($\pm$ 0.00) | 0.694 ($\pm$ 0.00) | 0.693 ($\pm$ 0.00) | 0.702 ($\pm$ 0.00) | **0.723** ($\pm$ **0.02**) | 0.723 ($\pm$ 0.02) |
| 0.20 | 0.670 ($\pm$ 0.00) | 0.686 ($\pm$ 0.00) | 0.699 ($\pm$ 0.00) | 0.691 ($\pm$ 0.00) | 0.696 ($\pm$ 0.00) | 0.698 ($\pm$ 0.00) | **0.713** ($\pm$ **0.02**) | 0.713 ($\pm$ 0.02) |
| 0.30 | 0.666 ($\pm$ 0.00) | 0.682 ($\pm$ 0.00) | 0.697 ($\pm$ 0.00) | 0.691 ($\pm$ 0.00) | 0.694 ($\pm$ 0.00) | 0.699 ($\pm$ 0.00) | **0.711** ($\pm$ **0.03**) | 0.711 ($\pm$ 0.03) |
| 0.40 | 0.652 ($\pm$ 0.00) | 0.683 ($\pm$ 0.00) | 0.698 ($\pm$ 0.00) | 0.691 ($\pm$ 0.00) | 0.688 ($\pm$ 0.00) | 0.701 ($\pm$ 0.00) | **0.715** ($\pm$ **0.02**) | 0.715 ($\pm$ 0.02) |
| 0.50 | 0.650 ($\pm$ 0.00) | 0.690 ($\pm$ 0.00) | 0.699 ($\pm$ 0.00) | 0.693 ($\pm$ 0.00) | 0.688 ($\pm$ 0.00) | **0.702** ($\pm$ **0.00**) | 0.694 ($\pm$ 0.02) | 0.694 ($\pm$ 0.02) |
| 0.60 | 0.622 ($\pm$ 0.00) | 0.686 ($\pm$ 0.00) | 0.685 ($\pm$ 0.00) | 0.685 ($\pm$ 0.00) | 0.681 ($\pm$ 0.00) | **0.704** ($\pm$ **0.00**) | 0.684 ($\pm$ 0.02) | 0.684 ($\pm$ 0.02) |
| 0.70 | 0.613 ($\pm$ 0.00) | 0.687 ($\pm$ 0.00) | 0.686 ($\pm$ 0.00) | 0.674 ($\pm$ 0.00) | 0.677 ($\pm$ 0.00) | **0.700** ($\pm$ **0.00**) | 0.685 ($\pm$ 0.03) | 0.685 ($\pm$ 0.03) |
| 0.80 | 0.582 ($\pm$ 0.00) | 0.671 ($\pm$ 0.00) | 0.677 ($\pm$ 0.00) | 0.664 ($\pm$ 0.00) | 0.534 ($\pm$ 0.00) | **0.686** ($\pm$ **0.00**) | 0.674 ($\pm$ 0.02) | 0.674 ($\pm$ 0.02) |
| 0.90 | 0.456 ($\pm$ 0.00) | **0.671** ($\pm$ **0.00**) | 0.650 ($\pm$ 0.00) | 0.650 ($\pm$ 0.00) | 0.607 ($\pm$ 0.00) | 0.648 ($\pm$ 0.00) | 0.593 ($\pm$ 0.02) | 0.593 ($\pm$ 0.02) |
| 0.99 | 0.171 ($\pm$ 0.00) | 0.257 ($\pm$ 0.00) | 0.298 ($\pm$ 0.00) | 0.346 ($\pm$ 0.00) | 0.195 ($\pm$ 0.00) | **0.348** ($\pm$ **0.00**) | 0.184 ($\pm$ 0.02) | 0.194 ($\pm$ 0.03) |

*Table 16.* F1 scores for CITESEER under mechanism *FD-MNAR* and varying $\mu$ (GSPN is not reported as it is not designed for categorical features).

| $\mu$ | GOODIE | FairAC | FP | GNNmi | GCNmf | PCFI | GNNzero | GNNmedian |
|---|---|---|---|---|---|---|---|---|
| 0.00 | 0.687 ($\pm$ 0.00) | 0.700 ($\pm$ 0.00) | 0.710 ($\pm$ 0.02) | 0.704 ($\pm$ 0.02) | 0.707 ($\pm$ 0.00) | 0.706 ($\pm$ 0.02) | **0.728** ($\pm$ **0.02**) | 0.728 ($\pm$ 0.02) |
| 0.10 | 0.689 ($\pm$ 0.03) | 0.691 ($\pm$ 0.03) | 0.706 ($\pm$ 0.02) | 0.699 ($\pm$ 0.02) | 0.699 ($\pm$ 0.02) | 0.708 ($\pm$ 0.03) | **0.729** ($\pm$ **0.02**) | 0.729 ($\pm$ 0.02) |
| 0.20 | 0.686 ($\pm$ 0.02) | 0.698 ($\pm$ 0.02) | 0.703 ($\pm$ 0.02) | 0.697 ($\pm$ 0.02) | 0.696 ($\pm$ 0.02) | 0.704 ($\pm$ 0.02) | **0.720** ($\pm$ **0.02**) | 0.720 ($\pm$ 0.02) |
| 0.30 | 0.701 ($\pm$ 0.04) | 0.690 ($\pm$ 0.03) | 0.701 ($\pm$ 0.03) | 0.693 ($\pm$ 0.02) | 0.704 ($\pm$ 0.02) | 0.700 ($\pm$ 0.03) | **0.721** ($\pm$ **0.03**) | 0.721 ($\pm$ 0.03) |
| 0.40 | 0.696 ($\pm$ 0.04) | 0.699 ($\pm$ 0.04) | 0.695 ($\pm$ 0.02) | 0.695 ($\pm$ 0.02) | 0.692 ($\pm$ 0.03) | 0.701 ($\pm$ 0.03) | **0.717** ($\pm$ **0.02**) | 0.717 ($\pm$ 0.02) |
| 0.50 | 0.707 ($\pm$ 0.03) | 0.688 ($\pm$ 0.04) | 0.698 ($\pm$ 0.03) | 0.693 ($\pm$ 0.03) | 0.690 ($\pm$ 0.02) | 0.702 ($\pm$ 0.03) | **0.727** ($\pm$ **0.02**) | 0.727 ($\pm$ 0.02) |
| 0.60 | 0.708 ($\pm$ 0.02) | 0.694 ($\pm$ 0.03) | 0.691 ($\pm$ 0.03) | 0.693 ($\pm$ 0.03) | 0.696 ($\pm$ 0.02) | 0.702 ($\pm$ 0.03) | **0.712** ($\pm$ **0.03**) | 0.712 ($\pm$ 0.03) |
| 0.70 | 0.678 ($\pm$ 0.04) | 0.688 ($\pm$ 0.03) | 0.688 ($\pm$ 0.03) | 0.686 ($\pm$ 0.02) | 0.649 ($\pm$ 0.03) | 0.690 ($\pm$ 0.04) | **0.705** ($\pm$ **0.02**) | 0.705 ($\pm$ 0.02) |
| 0.80 | 0.695 ($\pm$ 0.03) | 0.689 ($\pm$ 0.04) | 0.689 ($\pm$ 0.02) | 0.685 ($\pm$ 0.02) | 0.437 ($\pm$ 0.27) | 0.694 ($\pm$ 0.03) | **0.696** ($\pm$ **0.03**) | 0.696 ($\pm$ 0.03) |
| 0.90 | 0.653 ($\pm$ 0.03) | 0.681 ($\pm$ 0.04) | 0.682 ($\pm$ 0.02) | 0.687 ($\pm$ 0.03) | 0.257 ($\pm$ 0.17) | **0.689** ($\pm$ **0.02**) | 0.676 ($\pm$ 0.02) | 0.676 ($\pm$ 0.02) |
| 0.99 | 0.601 ($\pm$ 0.01) | 0.566 ($\pm$ 0.01) | 0.611 ($\pm$ 0.01) | 0.535 ($\pm$ 0.02) | 0.118 ($\pm$ 0.04) | **0.633** ($\pm$ **0.01**) | 0.538 ($\pm$ 0.03) | 0.538 ($\pm$ 0.03) |

*Table 17.* F1 scores for CITESEER under mechanism *CD-MNAR* and varying $\mu$ (GSPNis not reported as it is not designed for categorical features).

| $\mu$ | GOODIE | FairAC | FP | GNNmi | GCNmf | PCFI | GNNzero | GNNmedian |
|---|---|---|---|---|---|---|---|---|
| 0.00 | 0.687 ($\pm$ 0.00) | 0.700 ($\pm$ 0.05) | 0.710 ($\pm$ 0.02) | 0.704 ($\pm$ 0.02) | 0.707 ($\pm$ 0.00) | 0.706 ($\pm$ 0.02) | **0.726** ($\pm$ **0.02**) | 0.726 ($\pm$ 0.02) |
| 0.10 | 0.692 ($\pm$ 0.04) | 0.696 ($\pm$ 0.04) | 0.708 ($\pm$ 0.02) | 0.705 ($\pm$ 0.02) | 0.702 ($\pm$ 0.03) | 0.705 ($\pm$ 0.02) | **0.729** ($\pm$ **0.02**) | 0.729 ($\pm$ 0.02) |
| 0.20 | 0.690 ($\pm$ 0.04) | 0.689 ($\pm$ 0.04) | 0.703 ($\pm$ 0.03) | 0.702 ($\pm$ 0.02) | 0.705 ($\pm$ 0.02) | 0.704 ($\pm$ 0.02) | **0.727** ($\pm$ **0.02**) | 0.727 ($\pm$ 0.02) |
| 0.30 | 0.700 ($\pm$ 0.02) | 0.689 ($\pm$ 0.04) | 0.708 ($\pm$ 0.03) | 0.706 ($\pm$ 0.02) | 0.708 ($\pm$ 0.02) | 0.705 ($\pm$ 0.02) | **0.728** ($\pm$ **0.02**) | 0.728 ($\pm$ 0.02) |
| 0.40 | 0.687 ($\pm$ 0.04) | 0.695 ($\pm$ 0.04) | 0.707 ($\pm$ 0.03) | 0.704 ($\pm$ 0.02) | 0.703 ($\pm$ 0.03) | 0.704 ($\pm$ 0.03) | **0.725** ($\pm$ **0.02**) | 0.725 ($\pm$ 0.02) |
| 0.50 | 0.675 ($\pm$ 0.03) | 0.692 ($\pm$ 0.03) | 0.699 ($\pm$ 0.03) | 0.700 ($\pm$ 0.03) | 0.697 ($\pm$ 0.02) | 0.706 ($\pm$ 0.03) | **0.718** ($\pm$ **0.02**) | 0.718 ($\pm$ 0.02) |
| 0.60 | 0.689 ($\pm$ 0.03) | 0.689 ($\pm$ 0.03) | 0.702 ($\pm$ 0.03) | 0.699 ($\pm$ 0.03) | 0.693 ($\pm$ 0.03) | 0.706 ($\pm$ 0.03) | **0.714** ($\pm$ **0.02**) | 0.714 ($\pm$ 0.02) |
| 0.70 | 0.681 ($\pm$ 0.03) | 0.685 ($\pm$ 0.03) | 0.692 ($\pm$ 0.03) | 0.691 ($\pm$ 0.03) | 0.522 ($\pm$ 0.20) | 0.696 ($\pm$ 0.03) | **0.702** ($\pm$ **0.03**) | 0.702 ($\pm$ 0.03) |
| 0.80 | 0.676 ($\pm$ 0.05) | 0.685 ($\pm$ 0.03) | 0.690 ($\pm$ 0.03) | 0.689 ($\pm$ 0.02) | 0.359 ($\pm$ 0.15) | **0.696** ($\pm$ **0.04**) | 0.689 ($\pm$ 0.03) | 0.689 ($\pm$ 0.03) |
| 0.90 | 0.665 ($\pm$ 0.02) | **0.681** ($\pm$ **0.03**) | 0.677 ($\pm$ 0.03) | 0.666 ($\pm$ 0.03) | 0.113 ($\pm$ 0.06) | 0.681 ($\pm$ 0.03) | 0.638 ($\pm$ 0.02) | 0.638 ($\pm$ 0.02) |
| 0.99 | 0.645 ($\pm$ 0.03) | 0.631 ($\pm$ 0.02) | 0.652 ($\pm$ 0.02) | 0.621 ($\pm$ 0.02) | 0.104 ($\pm$ 0.06) | **0.660** ($\pm$ **0.02**) | 0.593 ($\pm$ 0.03) | 0.592 ($\pm$ 0.03) |

*Table 18.* F1 scores for PUBMED under mechanism *U-MCAR* and varying $\mu$ (GSPNis not reported as it is not designed for categorical features).

| $\mu$ | GOODIE | FairAC | FP | GNNmi | GCNmf | PCFI | GNNzero | GNNmedian |
|---|---|---|---|---|---|---|---|---|
| 0.00 | 0.784 ($\pm$ 0.01) | 0.831 ($\pm$ 0.00) | **0.883** ($\pm$ **0.00**) | 0.881 ($\pm$ 0.00) | 0.877 ($\pm$ 0.00) | 0.882 ($\pm$ 0.00) | 0.875 ($\pm$ 0.00) | 0.875 ($\pm$ 0.00) |
| 0.10 | 0.787 ($\pm$ 0.00) | 0.830 ($\pm$ 0.00) | 0.877 ($\pm$ 0.00) | **0.879** ($\pm$ **0.00**) | 0.830 ($\pm$ 0.00) | 0.874 ($\pm$ 0.00) | 0.871 ($\pm$ 0.00) | 0.871 ($\pm$ 0.00) |
| 0.20 | 0.786 ($\pm$ 0.00) | 0.831 ($\pm$ 0.00) | 0.868 ($\pm$ 0.00) | **0.873** ($\pm$ **0.00**) | 0.832 ($\pm$ 0.00) | 0.868 ($\pm$ 0.00) | 0.866 ($\pm$ 0.00) | 0.866 ($\pm$ 0.00) |
| 0.30 | 0.785 ($\pm$ 0.00) | 0.830 ($\pm$ 0.00) | 0.870 ($\pm$ 0.00) | **0.872** ($\pm$ **0.00**) | 0.827 ($\pm$ 0.00) | 0.864 ($\pm$ 0.00) | 0.862 ($\pm$ 0.00) | 0.860 ($\pm$ 0.00) |
| 0.40 | 0.782 ($\pm$ 0.00) | 0.828 ($\pm$ 0.00) | 0.861 ($\pm$ 0.00) | **0.869** ($\pm$ **0.00**) | 0.828 ($\pm$ 0.00) | 0.858 ($\pm$ 0.00) | 0.857 ($\pm$ 0.01) | 0.857 ($\pm$ 0.00) |
| 0.50 | 0.784 ($\pm$ 0.00) | 0.827 ($\pm$ 0.00) | 0.856 ($\pm$ 0.00) | **0.862** ($\pm$ **0.00**) | 0.778 ($\pm$ 0.00) | 0.852 ($\pm$ 0.00) | 0.851 ($\pm$ 0.01) | 0.852 ($\pm$ 0.00) |
| 0.60 | 0.777 ($\pm$ 0.00) | 0.828 ($\pm$ 0.00) | 0.851 ($\pm$ 0.00) | **0.855** ($\pm$ **0.00**) | 0.805 ($\pm$ 0.00) | 0.849 ($\pm$ 0.00) | 0.846 ($\pm$ 0.00) | 0.845 ($\pm$ 0.00) |
| 0.70 | 0.772 ($\pm$ 0.00) | 0.824 ($\pm$ 0.00) | **0.847** ($\pm$ **0.00**) | 0.845 ($\pm$ 0.00) | 0.726 ($\pm$ 0.00) | 0.844 ($\pm$ 0.00) | 0.834 ($\pm$ 0.01) | 0.835 ($\pm$ 0.01) |
| 0.80 | 0.756 ($\pm$ 0.00) | 0.819 ($\pm$ 0.00) | 0.836 ($\pm$ 0.00) | 0.832 ($\pm$ 0.00) | 0.443 ($\pm$ 0.00) | **0.837** ($\pm$ **0.00**) | 0.820 ($\pm$ 0.00) | 0.816 ($\pm$ 0.00) |
| 0.90 | 0.700 ($\pm$ 0.00) | 0.806 ($\pm$ 0.00) | 0.822 ($\pm$ 0.00) | 0.803 ($\pm$ 0.00) | 0.315 ($\pm$ 0.00) | **0.832** ($\pm$ **0.00**) | 0.791 ($\pm$ 0.01) | 0.786 ($\pm$ 0.01) |
| 0.99 | 0.452 ($\pm$ 0.00) | 0.262 ($\pm$ 0.00) | 0.793 ($\pm$ 0.00) | 0.327 ($\pm$ 0.00) | 0.315 ($\pm$ 0.00) | **0.814** ($\pm$ **0.00**) | 0.674 ($\pm$ 0.02) | 0.693 ($\pm$ 0.01) |

*Table 19.* F1 scores for PUBMED under mechanism *S-MCAR* and varying $\mu$ (GSPNis not reported as it is not designed for categorical features).

| $\mu$ | GOODIE | FairAC | FP | GNNmi | GCNmf | PCFI | GNNzero | GNNmedian |
|---|---|---|---|---|---|---|---|---|
| 0.00 | 0.784 ($\pm$ 0.01) | 0.831 ($\pm$ 0.00) | **0.883** ($\pm$ **0.00**) | - | 0.877 ($\pm$ 0.00) | 0.882 ($\pm$ 0.00) | 0.875 ($\pm$ 0.00) | 0.875 ($\pm$ 0.00) |
| 0.10 | 0.786 ($\pm$ 0.00) | 0.831 ($\pm$ 0.00) | 0.875 ($\pm$ 0.00) | **0.875** ($\pm$ **0.00**) | 0.870 ($\pm$ 0.00) | 0.871 ($\pm$ 0.00) | 0.868 ($\pm$ 0.01) | 0.866 ($\pm$ 0.01) |
| 0.20 | 0.783 ($\pm$ 0.00) | 0.827 ($\pm$ 0.00) | 0.869 ($\pm$ 0.00) | **0.870** ($\pm$ **0.00**) | 0.861 ($\pm$ 0.00) | 0.867 ($\pm$ 0.00) | 0.860 ($\pm$ 0.01) | 0.859 ($\pm$ 0.01) |
| 0.30 | 0.785 ($\pm$ 0.00) | 0.832 ($\pm$ 0.00) | 0.863 ($\pm$ 0.00) | **0.865** ($\pm$ **0.00**) | 0.861 ($\pm$ 0.00) | 0.863 ($\pm$ 0.00) | 0.853 ($\pm$ 0.00) | 0.852 ($\pm$ 0.00) |
| 0.40 | 0.785 ($\pm$ 0.00) | 0.828 ($\pm$ 0.00) | 0.856 ($\pm$ 0.00) | **0.857** ($\pm$ **0.00**) | 0.848 ($\pm$ 0.00) | 0.856 ($\pm$ 0.00) | 0.846 ($\pm$ 0.01) | 0.847 ($\pm$ 0.01) |
| 0.50 | 0.775 ($\pm$ 0.00) | 0.827 ($\pm$ 0.00) | 0.853 ($\pm$ 0.00) | **0.854** ($\pm$ **0.00**) | 0.808 ($\pm$ 0.00) | 0.848 ($\pm$ 0.00) | 0.838 ($\pm$ 0.00) | 0.837 ($\pm$ 0.00) |
| 0.60 | 0.774 ($\pm$ 0.00) | 0.822 ($\pm$ 0.00) | 0.843 ($\pm$ 0.00) | **0.845** ($\pm$ **0.00**) | 0.798 ($\pm$ 0.00) | 0.843 ($\pm$ 0.00) | 0.829 ($\pm$ 0.00) | 0.827 ($\pm$ 0.00) |
| 0.70 | 0.760 ($\pm$ 0.00) | 0.813 ($\pm$ 0.00) | 0.832 ($\pm$ 0.00) | 0.827 ($\pm$ 0.00) | 0.762 ($\pm$ 0.00) | **0.836** ($\pm$ **0.00**) | 0.815 ($\pm$ 0.00) | 0.814 ($\pm$ 0.00) |
| 0.80 | 0.744 ($\pm$ 0.00) | 0.806 ($\pm$ 0.00) | 0.828 ($\pm$ 0.00) | 0.808 ($\pm$ 0.00) | 0.683 ($\pm$ 0.00) | **0.832** ($\pm$ **0.00**) | 0.785 ($\pm$ 0.01) | 0.788 ($\pm$ 0.01) |
| 0.90 | 0.706 ($\pm$ 0.00) | 0.786 ($\pm$ 0.00) | 0.815 ($\pm$ 0.00) | 0.743 ($\pm$ 0.00) | 0.421 ($\pm$ 0.00) | **0.825** ($\pm$ **0.00**) | 0.727 ($\pm$ 0.01) | 0.729 ($\pm$ 0.00) |
| 0.99 | 0.441 ($\pm$ 0.00) | 0.259 ($\pm$ 0.00) | 0.765 ($\pm$ 0.00) | 0.333 ($\pm$ 0.00) | 0.310 ($\pm$ 0.00) | **0.794** ($\pm$ **0.00**) | 0.446 ($\pm$ 0.03) | 0.458 ($\pm$ 0.02) |

*Table 20.* F1 scores for PUBMED under mechanism *LD-MCAR* and varying $\mu$ (GSPN is not reported as it is not designed for categorical features).

| $\mu$ | GOODIE | FairAC | FP | GNNmi | GCNmf | PCFI | GNNzero | GNNmedian |
|---|---|---|---|---|---|---|---|---|
| 0.00 | 0.784 (± 0.01) | 0.831 (± 0.00) | **0.883** (± 0.00) | 0.881 (± 0.00) | 0.877 (± 0.00) | 0.882 (± 0.00) | 0.875 (± 0.00) | 0.876 (± 0.00) |
| 0.10 | 0.738 (± 0.00) | 0.824 (± 0.00) | 0.855 (± 0.00) | **0.857** (± 0.00) | 0.830 (± 0.00) | 0.852 (± 0.00) | 0.848 (± 0.00) | 0.846 (± 0.00) |
| 0.20 | 0.700 (± 0.00) | 0.820 (± 0.00) | 0.845 (± 0.00) | **0.851** (± 0.00) | 0.828 (± 0.00) | 0.844 (± 0.00) | 0.837 (± 0.00) | 0.836 (± 0.00) |
| 0.30 | 0.607 (± 0.00) | 0.823 (± 0.00) | 0.843 (± 0.00) | **0.844** (± 0.00) | 0.823 (± 0.00) | 0.836 (± 0.00) | 0.822 (± 0.00) | 0.822 (± 0.00) |
| 0.40 | 0.534 (± 0.00) | 0.821 (± 0.00) | 0.834 (± 0.00) | **0.842** (± 0.00) | 0.818 (± 0.00) | 0.830 (± 0.00) | 0.821 (± 0.01) | 0.821 (± 0.01) |
| 0.50 | 0.509 (± 0.00) | 0.814 (± 0.00) | 0.818 (± 0.00) | **0.823** (± 0.00) | 0.797 (± 0.00) | 0.820 (± 0.00) | 0.808 (± 0.01) | 0.806 (± 0.01) |
| 0.60 | 0.422 (± 0.00) | 0.812 (± 0.00) | 0.808 (± 0.00) | **0.816** (± 0.00) | 0.787 (± 0.00) | 0.812 (± 0.00) | 0.790 (± 0.00) | 0.793 (± 0.01) |
| 0.70 | 0.415 (± 0.00) | 0.802 (± 0.00) | 0.797 (± 0.00) | **0.811** (± 0.00) | 0.779 (± 0.00) | 0.801 (± 0.00) | 0.778 (± 0.01) | 0.774 (± 0.01) |
| 0.80 | 0.396 (± 0.00) | 0.779 (± 0.00) | 0.749 (± 0.00) | **0.783** (± 0.00) | 0.713 (± 0.00) | 0.754 (± 0.00) | 0.738 (± 0.01) | 0.749 (± 0.02) |
| 0.90 | 0.306 (± 0.00) | 0.574 (± 0.00) | 0.693 (± 0.00) | **0.700** (± 0.00) | 0.391 (± 0.00) | 0.683 (± 0.00) | 0.664 (± 0.01) | 0.667 (± 0.02) |
| 0.99 | 0.198 (± 0.00) | 0.266 (± 0.00) | 0.303 (± 0.00) | 0.330 (± 0.00) | 0.306 (± 0.00) | 0.305 (± 0.00) | **0.346** (± 0.02) | 0.345 (± 0.02) |

*Table 21.* F1 scores for PUBMED under mechanism *FD-MNAR* and varying $\mu$ (GSPN is not reported as it is not designed for categorical features).

| $\mu$ | GOODIE | FairAC | FP | GNNmi | GCNmf | PCFI | GNNzero | GNNmedian |
|---|---|---|---|---|---|---|---|---|
| 0.00 | 0.784 (± 0.01) | 0.831 (± 0.00) | **0.883** (± 0.00) | 0.881 (± 0.00) | 0.877 (± 0.00) | 0.882 (± 0.00) | 0.875 (± 0.00) | 0.874 (± 0.00) |
| 0.10 | 0.785 (± 0.02) | 0.832 (± 0.00) | 0.876 (± 0.01) | **0.880** (± 0.01) | 0.834 (± 0.00) | 0.874 (± 0.01) | 0.867 (± 0.01) | 0.868 (± 0.00) |
| 0.20 | 0.785 (± 0.02) | 0.834 (± 0.00) | 0.869 (± 0.00) | **0.875** (± 0.00) | 0.832 (± 0.00) | 0.869 (± 0.01) | 0.864 (± 0.01) | 0.864 (± 0.00) |
| 0.30 | 0.785 (± 0.02) | 0.830 (± 0.00) | 0.865 (± 0.00) | **0.870** (± 0.00) | 0.829 (± 0.00) | 0.860 (± 0.00) | 0.858 (± 0.00) | 0.858 (± 0.01) |
| 0.40 | 0.780 (± 0.01) | 0.827 (± 0.00) | 0.860 (± 0.00) | **0.866** (± 0.00) | 0.733 (± 0.11) | 0.856 (± 0.00) | 0.853 (± 0.01) | 0.854 (± 0.00) |
| 0.50 | 0.775 (± 0.02) | 0.822 (± 0.00) | 0.853 (± 0.00) | **0.859** (± 0.00) | 0.720 (± 0.12) | 0.850 (± 0.00) | 0.844 (± 0.01) | 0.846 (± 0.00) |
| 0.60 | 0.763 (± 0.02) | 0.824 (± 0.01) | 0.847 (± 0.01) | **0.850** (± 0.00) | 0.746 (± 0.04) | 0.842 (± 0.00) | 0.836 (± 0.01) | 0.836 (± 0.00) |
| 0.70 | 0.745 (± 0.03) | 0.813 (± 0.00) | 0.836 (± 0.00) | 0.834 (± 0.00) | 0.579 (± 0.25) | **0.837** (± 0.00) | 0.827 (± 0.00) | 0.826 (± 0.00) |
| 0.80 | 0.745 (± 0.03) | 0.819 (± 0.00) | 0.759 (± 0.04) | **0.829** (± 0.00) | 0.555 (± 0.14) | 0.764 (± 0.00) | 0.805 (± 0.01) | 0.805 (± 0.01) |
| 0.90 | 0.336 (± 0.01) | 0.806 (± 0.00) | 0.693 (± 0.01) | **0.812** (± 0.00) | 0.529 (± 0.13) | 0.653 (± 0.00) | 0.780 (± 0.01) | 0.777 (± 0.01) |
| 0.99 | 0.278 (± 0.01) | 0.282 (± 0.01) | 0.303 (± 0.05) | 0.347 (± 0.00) | 0.399 (± 0.33) | 0.335 (± 0.01) | 0.659 (± 0.02) | **0.669** (± 0.02) |

*Table 22.* F1 scores for PUBMED under mechanism *CD-MNAR* and varying $\mu$ (GSPN is not reported as it is not designed for categorical features).

| $\mu$ | GOODIE | FairAC | FP | GNNmi | GCNmf | PCFI | GNNzero | GNNmedian |
|---|---|---|---|---|---|---|---|---|
| 0.00 | 0.784 (± 0.01) | 0.831 (± 0.00) | **0.883** (± 0.00) | 0.881 (± 0.00) | 0.877 (± 0.00) | 0.882 (± 0.00) | 0.874 (± 0.00) | 0.875 (± 0.00) |
| 0.10 | 0.789 (± 0.02) | 0.829 (± 0.00) | 0.878 (± 0.00) | **0.880** (± 0.00) | 0.835 (± 0.00) | 0.877 (± 0.00) | 0.866 (± 0.01) | 0.869 (± 0.00) |
| 0.20 | 0.783 (± 0.01) | 0.830 (± 0.00) | 0.870 (± 0.00) | **0.876** (± 0.00) | 0.834 (± 0.00) | 0.867 (± 0.01) | 0.862 (± 0.00) | 0.861 (± 0.00) |
| 0.30 | 0.783 (± 0.02) | 0.828 (± 0.00) | 0.863 (± 0.00) | **0.871** (± 0.00) | 0.823 (± 0.00) | 0.866 (± 0.00) | 0.860 (± 0.00) | 0.859 (± 0.00) |
| 0.40 | 0.777 (± 0.02) | 0.826 (± 0.00) | 0.858 (± 0.00) | **0.863** (± 0.00) | 0.830 (± 0.00) | 0.857 (± 0.01) | 0.854 (± 0.00) | 0.852 (± 0.00) |
| 0.50 | 0.779 (± 0.01) | 0.825 (± 0.00) | 0.853 (± 0.00) | **0.858** (± 0.00) | 0.826 (± 0.00) | 0.853 (± 0.00) | 0.847 (± 0.00) | 0.849 (± 0.00) |
| 0.60 | 0.769 (± 0.02) | 0.824 (± 0.00) | 0.847 (± 0.01) | **0.848** (± 0.01) | 0.784 (± 0.04) | 0.848 (± 0.00) | 0.840 (± 0.01) | 0.840 (± 0.00) |
| 0.70 | 0.752 (± 0.03) | 0.816 (± 0.00) | 0.837 (± 0.00) | 0.835 (± 0.00) | 0.765 (± 0.02) | **0.837** (± 0.00) | 0.827 (± 0.00) | 0.825 (± 0.00) |
| 0.80 | 0.742 (± 0.03) | 0.813 (± 0.00) | 0.828 (± 0.00) | 0.817 (± 0.00) | 0.323 (± 0.10) | **0.836** (± 0.00) | 0.810 (± 0.01) | 0.809 (± 0.00) |
| 0.90 | 0.605 (± 0.13) | 0.628 (± 0.24) | 0.812 (± 0.00) | 0.770 (± 0.00) | 0.280 (± 0.05) | **0.823** (± 0.00) | 0.760 (± 0.01) | 0.763 (± 0.01) |
| 0.99 | 0.557 (± 0.14) | 0.260 (± 0.00) | 0.800 (± 0.00) | 0.689 (± 0.01) | 0.418 (± 0.04) | **0.818** (± 0.00) | 0.717 (± 0.01) | 0.728 (± 0.02) |

*Table 23.* F1 scores for SYNTHETIC under mechanism *U-MCAR* and varying $\mu$

| $\mu$ | GOODIE | GSPN | FairAC | FP | GNNmi | GCNmf | PCFI | GNNzero | GNNmedian | GNNmim |
|---|---|---|---|---|---|---|---|---|---|---|
| 0.00 | 0.812 (± 0.00) | 0.865 (± 0.00) | 0.815 (± 0.00) | 0.980 (± 0.00) | 0.982 (± 0.00) | 0.978 (± 0.00) | 0.977 (± 0.00) | 0.978 (± 0.01) | 0.978 (± 0.01) | **0.983** (± 0.01) |
| 0.10 | 0.810 (± 0.00) | 0.822 (± 0.00) | 0.825 (± 0.00) | **0.910** (± 0.00) | 0.902 (± 0.00) | 0.875 (± 0.00) | 0.898 (± 0.00) | 0.902 (± 0.02) | 0.903 (± 0.02) | 0.901 (± 0.00) |
| 0.20 | 0.792 (± 0.00) | 0.759 (± 0.00) | 0.808 (± 0.00) | 0.863 (± 0.00) | **0.870** (± 0.00) | 0.790 (± 0.00) | 0.855 (± 0.00) | 0.853 (± 0.02) | 0.853 (± 0.02) | 0.861 (± 0.00) |
| 0.30 | 0.758 (± 0.00) | 0.768 (± 0.00) | 0.762 (± 0.00) | 0.795 (± 0.00) | 0.808 (± 0.00) | 0.770 (± 0.00) | 0.805 (± 0.00) | 0.800 (± 0.03) | 0.801 (± 0.03) | **0.815** (± 0.00) |
| 0.40 | 0.758 (± 0.00) | 0.749 (± 0.00) | 0.759 (± 0.00) | 0.764 (± 0.00) | 0.771 (± 0.00) | 0.745 (± 0.00) | 0.763 (± 0.00) | 0.766 (± 0.02) | 0.766 (± 0.02) | **0.791** (± 0.00) |
| 0.50 | 0.747 (± 0.00) | 0.721 (± 0.00) | 0.642 (± 0.00) | 0.745 (± 0.00) | 0.745 (± 0.00) | 0.710 (± 0.00) | **0.748** (± 0.00) | 0.732 (± 0.04) | 0.730 (± 0.04) | 0.739 (± 0.00) |
| 0.60 | **0.773** (± 0.00) | 0.708 (± 0.00) | 0.680 (± 0.00) | 0.720 (± 0.00) | 0.737 (± 0.00) | 0.692 (± 0.00) | 0.717 (± 0.00) | 0.714 (± 0.04) | 0.710 (± 0.04) | 0.714 (± 0.00) |
| 0.70 | **0.742** (± 0.00) | 0.629 (± 0.00) | 0.611 (± 0.00) | 0.683 (± 0.00) | 0.689 (± 0.00) | 0.673 (± 0.00) | 0.678 (± 0.00) | 0.687 (± 0.03) | 0.693 (± 0.03) | 0.693 (± 0.00) |
| 0.80 | **0.771** (± 0.00) | 0.579 (± 0.00) | 0.621 (± 0.00) | 0.632 (± 0.00) | 0.638 (± 0.00) | 0.601 (± 0.00) | 0.638 (± 0.00) | 0.610 (± 0.05) | 0.621 (± 0.05) | 0.649 (± 0.00) |
| 0.90 | **0.776** (± 0.00) | 0.544 (± 0.00) | 0.567 (± 0.00) | 0.605 (± 0.00) | 0.602 (± 0.00) | 0.592 (± 0.00) | 0.588 (± 0.00) | 0.589 (± 0.04) | 0.599 (± 0.04) | 0.590 (± 0.00) |
| 0.99 | **0.762** (± 0.00) | 0.499 (± 0.00) | 0.391 (± 0.00) | 0.542 (± 0.00) | 0.367 (± 0.00) | 0.471 (± 0.00) | 0.547 (± 0.00) | 0.548 (± 0.04) | 0.411 (± 0.07) | 0.535 (± 0.00) |

*Table 24.* F1 scores for SYNTHETIC under mechanism *S-MCAR* and varying $\mu$

| $\mu$ | GOODIE | GSPN | FairAC | FP | GNNmi | GCNmf | PCFI | GNNzero | GNNmedian | GNNmim |
|---|---|---|---|---|---|---|---|---|---|---|
| 0.00 | 0.812 (± 0.00) | 0.865 (± 0.00) | 0.815 (± 0.00) | 0.980 (± 0.00) | 0.982 (± 0.00) | 0.978 (± 0.00) | 0.977 (± 0.00) | 0.978 (± 0.01) | 0.978 (± 0.01) | **0.983 (± 0.01)** |
| 0.10 | 0.756 (± 0.00) | 0.748 (± 0.00) | 0.723 (± 0.00) | 0.903 (± 0.00) | **0.912 (± 0.00)** | 0.903 (± 0.00) | 0.900 (± 0.00) | 0.909 (± 0.01) | 0.911 (± 0.01) | 0.898 (± 0.00) |
| 0.20 | 0.769 (± 0.00) | 0.733 (± 0.00) | 0.727 (± 0.00) | **0.883 (± 0.00)** | 0.883 (± 0.00) | 0.872 (± 0.00) | 0.870 (± 0.00) | 0.844 (± 0.02) | 0.843 (± 0.02) | 0.875 (± 0.00) |
| 0.30 | 0.742 (± 0.00) | 0.737 (± 0.00) | 0.700 (± 0.00) | 0.830 (± 0.00) | **0.842 (± 0.00)** | 0.841 (± 0.00) | 0.831 (± 0.00) | 0.817 (± 0.02) | 0.813 (± 0.01) | 0.833 (± 0.00) |
| 0.40 | 0.716 (± 0.00) | 0.712 (± 0.00) | 0.683 (± 0.00) | **0.810 (± 0.00)** | 0.798 (± 0.00) | 0.752 (± 0.00) | 0.793 (± 0.00) | 0.775 (± 0.02) | 0.777 (± 0.02) | 0.799 (± 0.00) |
| 0.50 | 0.700 (± 0.00) | 0.711 (± 0.00) | 0.704 (± 0.00) | 0.785 (± 0.00) | **0.788 (± 0.00)** | 0.705 (± 0.00) | 0.780 (± 0.00) | 0.746 (± 0.02) | 0.748 (± 0.02) | 0.779 (± 0.00) |
| 0.60 | 0.658 (± 0.00) | 0.674 (± 0.00) | 0.695 (± 0.00) | 0.747 (± 0.00) | **0.761 (± 0.00)** | 0.726 (± 0.00) | 0.738 (± 0.00) | 0.718 (± 0.03) | 0.705 (± 0.04) | 0.756 (± 0.00) |
| 0.70 | 0.618 (± 0.00) | 0.675 (± 0.00) | 0.652 (± 0.00) | 0.687 (± 0.00) | 0.703 (± 0.00) | 0.665 (± 0.00) | 0.700 (± 0.00) | 0.663 (± 0.03) | 0.667 (± 0.02) | **0.727 (± 0.00)** |
| 0.80 | 0.584 (± 0.00) | 0.649 (± 0.00) | 0.616 (± 0.00) | 0.653 (± 0.00) | 0.667 (± 0.00) | 0.645 (± 0.00) | 0.638 (± 0.00) | 0.647 (± 0.05) | 0.656 (± 0.04) | **0.676 (± 0.00)** |
| 0.90 | 0.527 (± 0.00) | 0.588 (± 0.00) | 0.589 (± 0.00) | 0.597 (± 0.00) | 0.597 (± 0.00) | 0.578 (± 0.00) | 0.591 (± 0.00) | **0.601 (± 0.02)** | 0.593 (± 0.02) | 0.582 (± 0.00) |
| 0.99 | 0.337 (± 0.00) | 0.455 (± 0.00) | 0.338 (± 0.00) | **0.515 (± 0.00)** | 0.425 (± 0.00) | 0.403 (± 0.00) | 0.513 (± 0.00) | 0.488 (± 0.02) | 0.444 (± 0.05) | 0.477 (± 0.00) |

*Table 25.* F1 scores for SYNTHETIC under mechanism *LD-MCAR* and varying $\mu$

| $\mu$ | GOODIE | GSPN | FairAC | FP | GNNmi | GCNmf | PCFI | GNNzero | GNNmedian | GNNmim |
|---|---|---|---|---|---|---|---|---|---|---|
| 0.00 | 0.812 (± 0.00) | 0.865 (± 0.00) | 0.815 (± 0.00) | 0.980 (± 0.00) | **0.982 (± 0.00)** | 0.978 (± 0.00) | 0.977 (± 0.00) | 0.978 (± 0.01) | 0.978 (± 0.01) | 0.886 (± 0.00) |
| 0.10 | 0.778 (± 0.00) | 0.785 (± 0.00) | 0.792 (± 0.00) | 0.860 (± 0.00) | 0.857 (± 0.00) | 0.845 (± 0.00) | 0.860 (± 0.00) | **0.978 (± 0.01)** | 0.978 (± 0.01) | 0.829 (± 0.00) |
| 0.20 | 0.760 (± 0.00) | 0.731 (± 0.00) | 0.705 (± 0.00) | **0.788 (± 0.00)** | 0.770 (± 0.00) | 0.741 (± 0.00) | 0.772 (± 0.00) | 0.699 (± 0.02) | 0.699 (± 0.02) | 0.780 (± 0.00) |
| 0.30 | 0.730 (± 0.00) | 0.666 (± 0.00) | 0.718 (± 0.00) | 0.736 (± 0.00) | 0.733 (± 0.00) | 0.730 (± 0.00) | 0.734 (± 0.00) | 0.605 (± 0.03) | 0.605 (± 0.03) | **0.738 (± 0.00)** |
| 0.40 | **0.736 (± 0.00)** | 0.625 (± 0.00) | 0.607 (± 0.00) | 0.661 (± 0.00) | 0.659 (± 0.00) | 0.673 (± 0.00) | 0.649 (± 0.00) | 0.605 (± 0.03) | 0.605 (± 0.03) | 0.703 (± 0.00) |
| 0.50 | **0.761 (± 0.00)** | 0.547 (± 0.00) | 0.542 (± 0.00) | 0.619 (± 0.00) | 0.618 (± 0.00) | 0.628 (± 0.00) | 0.613 (± 0.00) | 0.605 (± 0.03) | 0.605 (± 0.03) | 0.682 (± 0.00) |
| 0.60 | **0.768 (± 0.00)** | 0.594 (± 0.00) | 0.543 (± 0.00) | 0.621 (± 0.00) | 0.613 (± 0.00) | 0.619 (± 0.00) | 0.605 (± 0.00) | 0.528 (± 0.03) | 0.528 (± 0.03) | 0.667 (± 0.00) |
| 0.70 | **0.759 (± 0.00)** | 0.603 (± 0.00) | 0.586 (± 0.00) | 0.617 (± 0.00) | 0.607 (± 0.00) | 0.591 (± 0.00) | 0.594 (± 0.00) | 0.536 (± 0.03) | 0.536 (± 0.03) | 0.675 (± 0.00) |
| 0.80 | **0.758 (± 0.00)** | 0.613 (± 0.00) | 0.486 (± 0.00) | 0.617 (± 0.00) | 0.622 (± 0.00) | 0.631 (± 0.00) | 0.620 (± 0.00) | 0.536 (± 0.03) | 0.536 (± 0.03) | 0.666 (± 0.00) |
| 0.90 | **0.775 (± 0.00)** | 0.544 (± 0.00) | 0.529 (± 0.00) | 0.623 (± 0.00) | 0.633 (± 0.00) | 0.623 (± 0.00) | 0.606 (± 0.00) | 0.535 (± 0.02) | 0.536 (± 0.03) | 0.678 (± 0.00) |
| 0.99 | **0.764 (± 0.00)** | 0.569 (± 0.00) | 0.557 (± 0.00) | 0.609 (± 0.00) | 0.611 (± 0.00) | 0.643 (± 0.00) | 0.612 (± 0.00) | 0.646 (± 0.03) | 0.638 (± 0.03) | 0.667 (± 0.00) |

*Table 26.* F1 scores for SYNTHETIC under mechanism *FD-MNAR* and varying $\mu$

| $\mu$ | GOODIE | GSPN | FairAC | FP | GNNmi | GCNmf | PCFI | GNNzero | GNNmedian | GNNmim |
|---|---|---|---|---|---|---|---|---|---|---|
| 0.00 | 0.812 (± 0.00) | 0.865 (± 0.00) | 0.815 (± 0.00) | 0.980 (± 0.00) | 0.982 (± 0.00) | 0.978 (± 0.00) | 0.977 (± 0.00) | 0.976 (± 0.01) | 0.976 (± 0.01) | **0.983 (± 0.01)** |
| 0.10 | 0.751 (± 0.05) | 0.750 (± 0.03) | 0.761 (± 0.02) | 0.893 (± 0.01) | **0.900 (± 0.02)** | 0.878 (± 0.02) | 0.895 (± 0.01) | 0.891 (± 0.02) | 0.894 (± 0.02) | 0.895 (± 0.01) |
| 0.20 | 0.750 (± 0.03) | 0.721 (± 0.01) | 0.699 (± 0.04) | 0.836 (± 0.02) | 0.845 (± 0.02) | 0.785 (± 0.04) | 0.847 (± 0.02) | 0.849 (± 0.03) | **0.854 (± 0.02)** | 0.843 (± 0.04) |
| 0.30 | 0.691 (± 0.04) | 0.678 (± 0.02) | 0.667 (± 0.03) | 0.810 (± 0.01) | 0.812 (± 0.01) | 0.771 (± 0.03) | 0.789 (± 0.01) | 0.819 (± 0.02) | **0.821 (± 0.01)** | 0.812 (± 0.01) |
| 0.40 | 0.693 (± 0.03) | 0.678 (± 0.03) | 0.682 (± 0.03) | 0.791 (± 0.02) | 0.798 (± 0.00) | 0.763 (± 0.02) | 0.791 (± 0.00) | 0.785 (± 0.02) | 0.793 (± 0.02) | **0.806 (± 0.01)** |
| 0.50 | 0.673 (± 0.04) | 0.668 (± 0.01) | 0.676 (± 0.03) | 0.753 (± 0.01) | 0.758 (± 0.02) | 0.713 (± 0.03) | 0.752 (± 0.01) | 0.741 (± 0.02) | 0.737 (± 0.02) | **0.763 (± 0.01)** |
| 0.60 | 0.620 (± 0.02) | 0.608 (± 0.02) | 0.610 (± 0.02) | 0.708 (± 0.01) | 0.715 (± 0.00) | 0.685 (± 0.02) | 0.702 (± 0.02) | 0.714 (± 0.01) | 0.719 (± 0.01) | **0.727 (± 0.01)** |
| 0.70 | 0.494 (± 0.07) | 0.580 (± 0.06) | 0.588 (± 0.02) | 0.651 (± 0.03) | 0.670 (± 0.04) | 0.631 (± 0.03) | 0.653 (± 0.04) | 0.676 (± 0.02) | 0.673 (± 0.03) | **0.688 (± 0.02)** |
| 0.80 | 0.425 (± 0.07) | 0.607 (± 0.04) | 0.577 (± 0.01) | 0.611 (± 0.01) | 0.627 (± 0.02) | 0.589 (± 0.03) | 0.596 (± 0.01) | 0.619 (± 0.01) | 0.624 (± 0.01) | **0.639 (± 0.02)** |
| 0.90 | 0.362 (± 0.02) | **0.625 (± 0.02)** | 0.512 (± 0.05) | 0.575 (± 0.02) | 0.595 (± 0.02) | 0.573 (± 0.02) | 0.582 (± 0.01) | 0.594 (± 0.04) | 0.601 (± 0.02) | 0.612 (± 0.00) |
| 0.99 | 0.429 (± 0.13) | 0.570 (± 0.02) | 0.423 (± 0.11) | 0.547 (± 0.02) | 0.536 (± 0.01) | 0.490 (± 0.05) | 0.551 (± 0.01) | 0.569 (± 0.03) | 0.545 (± 0.04) | **0.576 (± 0.02)** |

*Table 27.* F1 scores for SYNTHETIC under mechanism *CD-MNAR* and varying $\mu$

| $\mu$ | GOODIE | GSPN | FairAC | FP | GNNmi | GCNmf | PCFI | GNNzero | GNNmedian | GNNmim |
|---|---|---|---|---|---|---|---|---|---|---|
| 0.00 | 0.812 (± 0.00) | 0.865 (± 0.00) | 0.815 (± 0.00) | 0.980 (± 0.00) | 0.982 (± 0.00) | 0.978 (± 0.00) | 0.977 (± 0.00) | 0.978 (± 0.01) | 0.978 (± 0.01) | **0.983 (± 0.01)** |
| 0.10 | 0.756 (± 0.04) | 0.757 (± 0.02) | 0.752 (± 0.02) | 0.913 (± 0.02) | **0.918 (± 0.02)** | 0.882 (± 0.02) | 0.912 (± 0.01) | 0.912 (± 0.02) | 0.912 (± 0.02) | 0.913 (± 0.02) |
| 0.20 | 0.730 (± 0.05) | 0.718 (± 0.02) | 0.674 (± 0.05) | 0.856 (± 0.03) | **0.868 (± 0.03)** | 0.800 (± 0.04) | 0.861 (± 0.04) | 0.864 (± 0.02) | 0.865 (± 0.02) | 0.865 (± 0.03) |
| 0.30 | 0.663 (± 0.05) | 0.716 (± 0.02) | 0.689 (± 0.03) | 0.803 (± 0.02) | 0.820 (± 0.02) | 0.768 (± 0.03) | 0.810 (± 0.03) | 0.807 (± 0.02) | 0.804 (± 0.02) | **0.830 (± 0.03)** |
| 0.40 | 0.530 (± 0.16) | 0.678 (± 0.01) | 0.718 (± 0.03) | 0.744 (± 0.01) | 0.749 (± 0.00) | 0.753 (± 0.01) | 0.739 (± 0.03) | 0.756 (± 0.01) | 0.742 (± 0.01) | **0.776 (± 0.01)** |
| 0.50 | 0.487 (± 0.12) | 0.662 (± 0.03) | 0.655 (± 0.04) | 0.697 (± 0.03) | 0.695 (± 0.03) | 0.683 (± 0.04) | 0.699 (± 0.04) | 0.689 (± 0.03) | 0.657 (± 0.02) | **0.725 (± 0.01)** |
| 0.60 | 0.575 (± 0.06) | 0.696 (± 0.03) | 0.577 (± 0.02) | 0.683 (± 0.03) | 0.658 (± 0.04) | 0.666 (± 0.03) | 0.645 (± 0.03) | 0.694 (± 0.04) | 0.638 (± 0.03) | **0.731 (± 0.03)** |
| 0.70 | 0.553 (± 0.03) | 0.616 (± 0.03) | 0.583 (± 0.02) | 0.613 (± 0.02) | 0.600 (± 0.04) | 0.617 (± 0.04) | 0.592 (± 0.05) | 0.642 (± 0.03) | 0.603 (± 0.04) | **0.668 (± 0.01)** |
| 0.80 | 0.486 (± 0.06) | 0.638 (± 0.03) | 0.592 (± 0.03) | 0.588 (± 0.02) | 0.596 (± 0.03) | 0.570 (± 0.02) | 0.563 (± 0.03) | 0.618 (± 0.02) | 0.580 (± 0.04) | **0.655 (± 0.02)** |
| 0.90 | 0.432 (± 0.08) | 0.618 (± 0.05) | 0.479 (± 0.10) | 0.586 (± 0.04) | 0.607 (± 0.03) | 0.556 (± 0.03) | 0.553 (± 0.01) | 0.598 (± 0.03) | 0.557 (± 0.04) | **0.635 (± 0.04)** |
| 0.99 | 0.468 (± 0.03) | 0.545 (± 0.06) | 0.396 (± 0.08) | **0.594 (± 0.01)** | 0.537 (± 0.01) | 0.475 (± 0.06) | 0.549 (± 0.03) | 0.550 (± 0.03) | 0.485 (± 0.06) | 0.568 (± 0.01) |

*Table 28.* F1 scores for AIR under mechanism *U-MCAR* and varying $\mu$

| $\mu$ | GOODIE | GSPN | FairAC | FP | GNNmi | GCNmf | PCFI | GNNzero | GNNmedian | GNNmim |
|---|---|---|---|---|---|---|---|---|---|---|
| 0.00 | 0.724 ($\pm$ 0.00) | 0.798 ($\pm$ 0.02) | 0.733 ($\pm$ 0.00) | 0.925 ($\pm$ 0.00) | 0.922 ($\pm$ 0.01) | 0.922 ($\pm$ 0.00) | 0.901 ($\pm$ 0.00) | 0.916 ($\pm$ 0.02) | 0.916 ($\pm$ 0.02) | **0.930** ($\pm$ **0.00**) |
| 0.10 | 0.665 ($\pm$ 0.00) | 0.710 ($\pm$ 0.00) | 0.733 ($\pm$ 0.00) | 0.897 ($\pm$ 0.00) | 0.891 ($\pm$ 0.00) | 0.768 ($\pm$ 0.00) | 0.888 ($\pm$ 0.00) | **0.904** ($\pm$ **0.03**) | 0.902 ($\pm$ 0.03) | 0.899 ($\pm$ 0.00) |
| 0.20 | 0.669 ($\pm$ 0.00) | 0.582 ($\pm$ 0.00) | 0.709 ($\pm$ 0.00) | 0.855 ($\pm$ 0.00) | 0.833 ($\pm$ 0.00) | 0.747 ($\pm$ 0.00) | 0.856 ($\pm$ 0.00) | **0.874** ($\pm$ **0.03**) | 0.865 ($\pm$ 0.03) | 0.859 ($\pm$ 0.00) |
| 0.30 | 0.669 ($\pm$ 0.00) | 0.502 ($\pm$ 0.00) | 0.715 ($\pm$ 0.00) | 0.836 ($\pm$ 0.00) | 0.837 ($\pm$ 0.00) | 0.712 ($\pm$ 0.00) | 0.836 ($\pm$ 0.00) | 0.837 ($\pm$ 0.04) | **0.857** ($\pm$ **0.03**) | 0.852 ($\pm$ 0.00) |
| 0.40 | 0.714 ($\pm$ 0.00) | 0.532 ($\pm$ 0.00) | 0.700 ($\pm$ 0.00) | 0.805 ($\pm$ 0.00) | 0.829 ($\pm$ 0.00) | 0.712 ($\pm$ 0.00) | 0.797 ($\pm$ 0.00) | 0.813 ($\pm$ 0.02) | **0.839** ($\pm$ **0.02**) | 0.833 ($\pm$ 0.00) |
| 0.50 | 0.666 ($\pm$ 0.00) | 0.553 ($\pm$ 0.00) | 0.669 ($\pm$ 0.00) | 0.803 ($\pm$ 0.00) | 0.805 ($\pm$ 0.00) | 0.711 ($\pm$ 0.00) | 0.808 ($\pm$ 0.00) | **0.832** ($\pm$ **0.04**) | 0.815 ($\pm$ 0.03) | 0.767 ($\pm$ 0.00) |
| 0.60 | 0.663 ($\pm$ 0.00) | 0.452 ($\pm$ 0.00) | 0.691 ($\pm$ 0.00) | 0.778 ($\pm$ 0.00) | 0.762 ($\pm$ 0.00) | 0.701 ($\pm$ 0.00) | 0.767 ($\pm$ 0.00) | 0.795 ($\pm$ 0.04) | **0.807** ($\pm$ **0.06**) | 0.744 ($\pm$ 0.00) |
| 0.70 | 0.714 ($\pm$ 0.00) | 0.495 ($\pm$ 0.00) | 0.686 ($\pm$ 0.00) | 0.724 ($\pm$ 0.00) | 0.736 ($\pm$ 0.00) | 0.656 ($\pm$ 0.00) | **0.759** ($\pm$ **0.00**) | 0.753 ($\pm$ 0.07) | 0.746 ($\pm$ 0.05) | 0.736 ($\pm$ 0.00) |
| 0.80 | 0.666 ($\pm$ 0.00) | 0.559 ($\pm$ 0.00) | 0.667 ($\pm$ 0.00) | 0.712 ($\pm$ 0.00) | 0.677 ($\pm$ 0.00) | 0.647 ($\pm$ 0.00) | 0.637 ($\pm$ 0.00) | 0.709 ($\pm$ 0.03) | **0.715** ($\pm$ **0.03**) | 0.713 ($\pm$ 0.00) |
| 0.90 | 0.700 ($\pm$ 0.00) | 0.541 ($\pm$ 0.00) | 0.670 ($\pm$ 0.00) | 0.585 ($\pm$ 0.00) | 0.593 ($\pm$ 0.00) | 0.669 ($\pm$ 0.00) | 0.619 ($\pm$ 0.00) | 0.598 ($\pm$ 0.06) | 0.628 ($\pm$ 0.04) | **0.705** ($\pm$ **0.00**) |
| 0.99 | **0.693** ($\pm$ **0.00**) | 0.409 ($\pm$ 0.00) | 0.658 ($\pm$ 0.00) | 0.431 ($\pm$ 0.00) | 0.384 ($\pm$ 0.00) | 0.651 ($\pm$ 0.00) | 0.431 ($\pm$ 0.00) | 0.440 ($\pm$ 0.05) | 0.397 ($\pm$ 0.04) | 0.664 ($\pm$ 0.00) |

*Table 29.* F1 scores for AIR under mechanism *S-MCAR* and varying $\mu$

| $\mu$ | GOODIE | GSPN | FairAC | FP | GNNmi | GCNmf | PCFI | GNNzero | GNNmedian | GNNmim |
|---|---|---|---|---|---|---|---|---|---|---|
| 0.00 | 0.724 ($\pm$ 0.00) | 0.798 ($\pm$ 0.02) | 0.733 ($\pm$ 0.00) | 0.925 ($\pm$ 0.00) | 0.922 ($\pm$ 0.01) | 0.922 ($\pm$ 0.00) | 0.901 ($\pm$ 0.00) | 0.916 ($\pm$ 0.02) | 0.916 ($\pm$ 0.02) | **0.930** ($\pm$ **0.00**) |
| 0.10 | 0.568 ($\pm$ 0.00) | 0.644 ($\pm$ 0.00) | 0.733 ($\pm$ 0.00) | 0.894 ($\pm$ 0.00) | 0.899 ($\pm$ 0.00) | 0.895 ($\pm$ 0.00) | **0.900** ($\pm$ **0.00**) | 0.879 ($\pm$ 0.02) | **0.900** ($\pm$ **0.02**) | 0.891 ($\pm$ 0.00) |
| 0.20 | 0.573 ($\pm$ 0.00) | 0.597 ($\pm$ 0.00) | 0.733 ($\pm$ 0.00) | 0.876 ($\pm$ 0.00) | 0.883 ($\pm$ 0.00) | 0.851 ($\pm$ 0.00) | **0.899** ($\pm$ **0.00**) | 0.860 ($\pm$ 0.03) | 0.865 ($\pm$ 0.03) | 0.890 ($\pm$ 0.00) |
| 0.30 | 0.630 ($\pm$ 0.00) | 0.527 ($\pm$ 0.00) | 0.665 ($\pm$ 0.00) | 0.852 ($\pm$ 0.00) | 0.847 ($\pm$ 0.00) | 0.820 ($\pm$ 0.00) | **0.855** ($\pm$ **0.00**) | 0.838 ($\pm$ 0.04) | 0.853 ($\pm$ 0.03) | 0.835 ($\pm$ 0.00) |
| 0.40 | 0.571 ($\pm$ 0.00) | 0.508 ($\pm$ 0.00) | 0.728 ($\pm$ 0.00) | 0.833 ($\pm$ 0.00) | 0.819 ($\pm$ 0.00) | 0.795 ($\pm$ 0.00) | **0.846** ($\pm$ **0.00**) | 0.812 ($\pm$ 0.03) | 0.796 ($\pm$ 0.04) | 0.842 ($\pm$ 0.00) |
| 0.50 | 0.562 ($\pm$ 0.00) | 0.530 ($\pm$ 0.00) | 0.742 ($\pm$ 0.00) | 0.789 ($\pm$ 0.00) | 0.770 ($\pm$ 0.00) | **0.829** ($\pm$ **0.00**) | 0.812 ($\pm$ 0.00) | 0.769 ($\pm$ 0.03) | 0.778 ($\pm$ 0.03) | 0.817 ($\pm$ 0.00) |
| 0.60 | 0.549 ($\pm$ 0.00) | 0.532 ($\pm$ 0.00) | 0.739 ($\pm$ 0.00) | 0.756 ($\pm$ 0.00) | 0.737 ($\pm$ 0.00) | **0.809** ($\pm$ **0.00**) | 0.761 ($\pm$ 0.00) | 0.736 ($\pm$ 0.06) | 0.718 ($\pm$ 0.04) | 0.797 ($\pm$ 0.00) |
| 0.70 | 0.603 ($\pm$ 0.00) | 0.532 ($\pm$ 0.00) | 0.706 ($\pm$ 0.00) | 0.682 ($\pm$ 0.00) | 0.661 ($\pm$ 0.00) | **0.767** ($\pm$ **0.00**) | 0.666 ($\pm$ 0.00) | 0.709 ($\pm$ 0.05) | 0.693 ($\pm$ 0.03) | 0.756 ($\pm$ 0.00) |
| 0.80 | 0.610 ($\pm$ 0.00) | 0.476 ($\pm$ 0.00) | 0.657 ($\pm$ 0.00) | 0.607 ($\pm$ 0.00) | 0.605 ($\pm$ 0.00) | 0.721 ($\pm$ 0.00) | 0.601 ($\pm$ 0.00) | 0.614 ($\pm$ 0.04) | 0.603 ($\pm$ 0.04) | **0.734** ($\pm$ **0.00**) |
| 0.90 | 0.504 ($\pm$ 0.00) | 0.389 ($\pm$ 0.00) | 0.692 ($\pm$ 0.00) | 0.549 ($\pm$ 0.00) | 0.505 ($\pm$ 0.00) | 0.677 ($\pm$ 0.00) | 0.522 ($\pm$ 0.00) | 0.537 ($\pm$ 0.03) | 0.511 ($\pm$ 0.02) | **0.699** ($\pm$ **0.00**) |
| 0.99 | 0.435 ($\pm$ 0.00) | 0.332 ($\pm$ 0.00) | **0.652** ($\pm$ **0.00**) | 0.450 ($\pm$ 0.00) | 0.333 ($\pm$ 0.00) | 0.643 ($\pm$ 0.00) | 0.353 ($\pm$ 0.00) | 0.351 ($\pm$ 0.01) | 0.354 ($\pm$ 0.01) | 0.652 ($\pm$ 0.00) |

*Table 30.* F1 scores for AIR under mechanism *LD-MCAR* and varying $\mu$

| $\mu$ | GOODIE | GSPN | FairAC | FP | GNNmi | GCNmf | PCFI | GNNzero | GNNmedian | GNNmim |
|---|---|---|---|---|---|---|---|---|---|---|
| 0.00 | 0.724 ($\pm$ 0.00) | 0.798 ($\pm$ 0.02) | 0.733 ($\pm$ 0.00) | 0.925 ($\pm$ 0.00) | 0.922 ($\pm$ 0.01) | 0.922 ($\pm$ 0.00) | 0.901 ($\pm$ 0.00) | 0.916 ($\pm$ 0.02) | 0.916 ($\pm$ 0.02) | **0.930** ($\pm$ **0.00**) |
| 0.10 | 0.714 ($\pm$ 0.00) | 0.730 ($\pm$ 0.00) | 0.706 ($\pm$ 0.00) | 0.804 ($\pm$ 0.00) | 0.819 ($\pm$ 0.00) | 0.700 ($\pm$ 0.00) | 0.820 ($\pm$ 0.00) | 0.825 ($\pm$ 0.05) | 0.825 ($\pm$ 0.05) | **0.876** ($\pm$ **0.00**) |
| 0.20 | 0.714 ($\pm$ 0.00) | 0.730 ($\pm$ 0.00) | 0.703 ($\pm$ 0.00) | 0.804 ($\pm$ 0.00) | 0.819 ($\pm$ 0.00) | 0.677 ($\pm$ 0.00) | 0.820 ($\pm$ 0.00) | 0.825 ($\pm$ 0.05) | 0.825 ($\pm$ 0.05) | **0.887** ($\pm$ **0.00**) |
| 0.30 | 0.710 ($\pm$ 0.00) | 0.651 ($\pm$ 0.00) | 0.613 ($\pm$ 0.00) | 0.721 ($\pm$ 0.00) | 0.697 ($\pm$ 0.00) | 0.696 ($\pm$ 0.00) | 0.726 ($\pm$ 0.00) | 0.725 ($\pm$ 0.07) | 0.725 ($\pm$ 0.07) | **0.744** ($\pm$ **0.00**) |
| 0.40 | 0.701 ($\pm$ 0.00) | 0.587 ($\pm$ 0.00) | 0.617 ($\pm$ 0.00) | 0.717 ($\pm$ 0.00) | 0.687 ($\pm$ 0.00) | 0.691 ($\pm$ 0.00) | 0.701 ($\pm$ 0.00) | 0.719 ($\pm$ 0.05) | 0.719 ($\pm$ 0.05) | **0.794** ($\pm$ **0.00**) |
| 0.50 | 0.717 ($\pm$ 0.00) | 0.504 ($\pm$ 0.00) | 0.458 ($\pm$ 0.00) | 0.528 ($\pm$ 0.00) | 0.571 ($\pm$ 0.00) | 0.625 ($\pm$ 0.00) | 0.564 ($\pm$ 0.00) | 0.556 ($\pm$ 0.08) | 0.556 ($\pm$ 0.08) | **0.722** ($\pm$ **0.00**) |
| 0.60 | 0.717 ($\pm$ 0.00) | 0.504 ($\pm$ 0.00) | 0.450 ($\pm$ 0.00) | 0.528 ($\pm$ 0.00) | 0.571 ($\pm$ 0.00) | 0.625 ($\pm$ 0.00) | 0.564 ($\pm$ 0.00) | 0.556 ($\pm$ 0.08) | 0.556 ($\pm$ 0.08) | **0.737** ($\pm$ **0.00**) |
| 0.70 | **0.717** ($\pm$ **0.00**) | 0.498 ($\pm$ 0.00) | 0.446 ($\pm$ 0.00) | 0.540 ($\pm$ 0.00) | 0.553 ($\pm$ 0.00) | 0.668 ($\pm$ 0.00) | 0.518 ($\pm$ 0.00) | 0.498 ($\pm$ 0.04) | 0.498 ($\pm$ 0.04) | 0.662 ($\pm$ 0.00) |
| 0.80 | **0.703** ($\pm$ **0.00**) | 0.557 ($\pm$ 0.00) | 0.430 ($\pm$ 0.00) | 0.515 ($\pm$ 0.00) | 0.481 ($\pm$ 0.00) | 0.676 ($\pm$ 0.00) | 0.457 ($\pm$ 0.00) | 0.495 ($\pm$ 0.05) | 0.495 ($\pm$ 0.05) | 0.680 ($\pm$ 0.00) |
| 0.90 | **0.703** ($\pm$ **0.00**) | 0.498 ($\pm$ 0.00) | 0.338 ($\pm$ 0.00) | 0.515 ($\pm$ 0.00) | 0.481 ($\pm$ 0.00) | 0.676 ($\pm$ 0.00) | 0.457 ($\pm$ 0.00) | 0.495 ($\pm$ 0.05) | 0.495 ($\pm$ 0.05) | 0.674 ($\pm$ 0.00) |
| 0.99 | 0.660 ($\pm$ 0.00) | 0.468 ($\pm$ 0.00) | 0.338 ($\pm$ 0.00) | 0.515 ($\pm$ 0.00) | 0.481 ($\pm$ 0.00) | 0.682 ($\pm$ 0.00) | 0.457 ($\pm$ 0.00) | 0.675 ($\pm$ 0.05) | **0.688** ($\pm$ **0.05**) | 0.673 ($\pm$ 0.00) |

*Table 31.* F1 scores for AIR under mechanism *FD-MNAR* and varying $\mu$

| $\mu$ | GOODIE | GSPN | FairAC | FP | GNNmi | GCNmf | PCFI | GNNzero | GNNmedian | GNNmim |
|---|---|---|---|---|---|---|---|---|---|---|
| 0.00 | 0.724 ($\pm$ 0.00) | 0.798 ($\pm$ 0.02) | 0.733 ($\pm$ 0.00) | 0.925 ($\pm$ 0.00) | 0.922 ($\pm$ 0.01) | 0.922 ($\pm$ 0.00) | 0.901 ($\pm$ 0.00) | 0.916 ($\pm$ 0.02) | 0.916 ($\pm$ 0.02) | **0.930** ($\pm$ **0.00**) |
| 0.10 | 0.618 ($\pm$ 0.10) | 0.758 ($\pm$ 0.05) | 0.709 ($\pm$ 0.03) | 0.895 ($\pm$ 0.01) | 0.891 ($\pm$ 0.04) | 0.772 ($\pm$ 0.02) | 0.883 ($\pm$ 0.03) | 0.890 ($\pm$ 0.03) | 0.897 ($\pm$ 0.03) | **0.906** ($\pm$ **0.02**) |
| 0.20 | 0.595 ($\pm$ 0.10) | 0.776 ($\pm$ 0.05) | 0.668 ($\pm$ 0.08) | 0.883 ($\pm$ 0.03) | 0.879 ($\pm$ 0.01) | 0.756 ($\pm$ 0.03) | 0.867 ($\pm$ 0.02) | 0.852 ($\pm$ 0.02) | **0.888** ($\pm$ **0.02**) | 0.887 ($\pm$ 0.01) |
| 0.30 | 0.580 ($\pm$ 0.12) | 0.536 ($\pm$ 0.15) | 0.721 ($\pm$ 0.01) | 0.852 ($\pm$ 0.03) | 0.859 ($\pm$ 0.01) | 0.745 ($\pm$ 0.03) | 0.833 ($\pm$ 0.02) | 0.845 ($\pm$ 0.02) | 0.864 ($\pm$ 0.03) | **0.875** ($\pm$ **0.01**) |
| 0.40 | 0.677 ($\pm$ 0.03) | 0.575 ($\pm$ 0.09) | 0.716 ($\pm$ 0.02) | 0.852 ($\pm$ 0.02) | **0.855** ($\pm$ **0.03**) | 0.725 ($\pm$ 0.02) | 0.840 ($\pm$ 0.04) | 0.839 ($\pm$ 0.02) | 0.848 ($\pm$ 0.04) | 0.852 ($\pm$ 0.02) |
| 0.50 | 0.587 ($\pm$ 0.13) | 0.620 ($\pm$ 0.08) | 0.719 ($\pm$ 0.02) | 0.837 ($\pm$ 0.03) | 0.832 ($\pm$ 0.03) | 0.698 ($\pm$ 0.04) | 0.829 ($\pm$ 0.04) | 0.806 ($\pm$ 0.01) | 0.822 ($\pm$ 0.04) | **0.852** ($\pm$ **0.03**) |
| 0.60 | 0.556 ($\pm$ 0.16) | 0.686 ($\pm$ 0.05) | 0.692 ($\pm$ 0.02) | **0.837** ($\pm$ **0.02**) | 0.808 ($\pm$ 0.06) | 0.711 ($\pm$ 0.03) | 0.793 ($\pm$ 0.02) | 0.780 ($\pm$ 0.04) | 0.783 ($\pm$ 0.05) | 0.817 ($\pm$ 0.03) |
| 0.70 | 0.556 ($\pm$ 0.16) | 0.634 ($\pm$ 0.02) | 0.717 ($\pm$ 0.02) | 0.769 ($\pm$ 0.03) | **0.779** ($\pm$ **0.05**) | 0.685 ($\pm$ 0.01) | 0.750 ($\pm$ 0.04) | 0.745 ($\pm$ 0.05) | 0.771 ($\pm$ 0.05) | 0.770 ($\pm$ 0.03) |
| 0.80 | 0.556 ($\pm$ 0.16) | 0.665 ($\pm$ 0.02) | 0.665 ($\pm$ 0.03) | 0.654 ($\pm$ 0.05) | 0.709 ($\pm$ 0.03) | 0.667 ($\pm$ 0.03) | 0.660 ($\pm$ 0.08) | 0.718 ($\pm$ 0.07) | 0.719 ($\pm$ 0.04) | **0.786** ($\pm$ **0.02**) |
| 0.90 | 0.582 ($\pm$ 0.09) | 0.645 ($\pm$ 0.04) | 0.662 ($\pm$ 0.01) | 0.658 ($\pm$ 0.05) | 0.661 ($\pm$ 0.02) | 0.659 ($\pm$ 0.03) | 0.530 ($\pm$ 0.05) | 0.670 ($\pm$ 0.06) | 0.655 ($\pm$ 0.05) | **0.710** ($\pm$ **0.05**) |
| 0.99 | 0.638 ($\pm$ 0.05) | 0.635 ($\pm$ 0.02) | 0.637 ($\pm$ 0.04) | 0.557 ($\pm$ 0.04) | 0.528 ($\pm$ 0.03) | **0.674** ($\pm$ **0.02**) | 0.508 ($\pm$ 0.07) | 0.549 ($\pm$ 0.06) | 0.565 ($\pm$ 0.04) | 0.616 ($\pm$ 0.05) |

*Table 32.* F1 scores for AIR under mechanism *CD-MNAR* and varying $\mu$

| $\mu$ | GOODIE | GSPN | FairAC | FP | GNNmi | GCNmf | PCFI | GNNzero | GNNmedian | GNNmim |
|---|---|---|---|---|---|---|---|---|---|---|
| 0.00 | 0.724 (± 0.00) | 0.798 (± 0.02) | 0.733 (± 0.00) | 0.925 (± 0.00) | 0.922 (± 0.01) | 0.922 (± 0.00) | 0.901 (± 0.00) | 0.916 (± 0.02) | 0.916 (± 0.02) | **0.930** (± **0.00**) |
| 0.10 | 0.598 (± 0.11) | 0.667 (± 0.02) | 0.722 (± 0.02) | 0.891 (± 0.02) | 0.887 (± 0.04) | 0.851 (± 0.01) | 0.891 (± 0.03) | 0.860 (± 0.04) | 0.883 (± 0.04) | **0.895** (± **0.04**) |
| 0.20 | 0.556 (± 0.16) | 0.632 (± 0.19) | 0.697 (± 0.02) | **0.869** (± **0.02**) | 0.848 (± 0.06) | 0.778 (± 0.04) | 0.841 (± 0.01) | 0.853 (± 0.04) | 0.836 (± 0.04) | 0.864 (± 0.01) |
| 0.30 | 0.556 (± 0.16) | 0.526 (± 0.13) | 0.722 (± 0.03) | 0.845 (± 0.01) | 0.825 (± 0.04) | 0.689 (± 0.02) | 0.841 (± 0.03) | 0.855 (± 0.02) | 0.806 (± 0.04) | **0.891** (± **0.04**) |
| 0.40 | 0.480 (± 0.16) | 0.691 (± 0.14) | 0.601 (± 0.12) | 0.833 (± 0.02) | 0.805 (± 0.03) | 0.722 (± 0.02) | 0.860 (± 0.03) | 0.856 (± 0.02) | 0.811 (± 0.02) | **0.860** (± **0.03**) |
| 0.50 | 0.536 (± 0.16) | 0.607 (± 0.09) | 0.705 (± 0.02) | 0.813 (± 0.02) | 0.769 (± 0.04) | 0.674 (± 0.01) | 0.783 (± 0.04) | 0.790 (± 0.05) | 0.777 (± 0.05) | **0.833** (± **0.03**) |
| 0.60 | 0.622 (± 0.06) | 0.636 (± 0.04) | 0.694 (± 0.01) | 0.759 (± 0.05) | 0.708 (± 0.07) | 0.681 (± 0.01) | 0.766 (± 0.06) | **0.814** (± **0.03**) | 0.774 (± 0.07) | 0.766 (± 0.06) |
| 0.70 | 0.580 (± 0.10) | 0.672 (± 0.07) | 0.681 (± 0.01) | **0.757** (± **0.03**) | 0.724 (± 0.04) | 0.644 (± 0.02) | 0.753 (± 0.05) | 0.755 (± 0.06) | 0.720 (± 0.04) | 0.726 (± 0.05) |
| 0.80 | 0.563 (± 0.12) | 0.681 (± 0.05) | 0.676 (± 0.01) | 0.733 (± 0.02) | 0.655 (± 0.02) | 0.658 (± 0.02) | 0.712 (± 0.01) | 0.735 (± 0.05) | 0.686 (± 0.06) | **0.769** (± **0.03**) |
| 0.90 | 0.655 (± 0.03) | 0.615 (± 0.04) | 0.653 (± 0.01) | **0.693** (± **0.04**) | 0.579 (± 0.04) | 0.643 (± 0.04) | 0.692 (± 0.06) | 0.678 (± 0.03) | 0.613 (± 0.04) | 0.668 (± 0.02) |
| 0.99 | 0.654 (± 0.03) | 0.522 (± 0.04) | **0.660** (± **0.05**) | 0.524 (± 0.07) | 0.473 (± 0.05) | 0.650 (± 0.06) | 0.424 (± 0.06) | 0.523 (± 0.06) | 0.411 (± 0.03) | 0.631 (± 0.07) |

*Table 33.* F1 scores for ELECTRIC under mechanism *U-MCAR* and varying $\mu$

| $\mu$ | GOODIE | GSPN | FairAC | FP | GNNmi | GCNmf | PCFI | GNNzero | GNNmedian | GNNmim |
|---|---|---|---|---|---|---|---|---|---|---|
| 0.00 | 0.588 (± 0.00) | 0.915 (± 0.00) | **0.963** (± **0.01**) | 0.885 (± 0.00) | 0.929 (± 0.00) | 0.861 (± 0.00) | 0.903 (± 0.00) | 0.912 (± 0.01) | 0.909 (± 0.01) | 0.938 (± 0.01) |
| 0.10 | 0.589 (± 0.00) | 0.827 (± 0.00) | **0.931** (± **0.00**) | 0.865 (± 0.00) | 0.864 (± 0.00) | 0.887 (± 0.00) | 0.889 (± 0.00) | 0.855 (± 0.03) | 0.854 (± 0.02) | 0.923 (± 0.00) |
| 0.20 | 0.589 (± 0.00) | 0.806 (± 0.00) | **0.935** (± **0.00**) | 0.821 (± 0.00) | 0.807 (± 0.00) | 0.876 (± 0.00) | 0.877 (± 0.00) | 0.805 (± 0.03) | 0.807 (± 0.03) | 0.877 (± 0.00) |
| 0.30 | 0.588 (± 0.00) | 0.770 (± 0.00) | **0.924** (± **0.00**) | 0.758 (± 0.00) | 0.780 (± 0.00) | 0.889 (± 0.00) | 0.872 (± 0.00) | 0.742 (± 0.03) | 0.781 (± 0.04) | 0.868 (± 0.00) |
| 0.40 | 0.590 (± 0.00) | 0.703 (± 0.00) | **0.906** (± **0.00**) | 0.711 (± 0.00) | 0.728 (± 0.00) | 0.874 (± 0.00) | 0.865 (± 0.00) | 0.710 (± 0.02) | 0.746 (± 0.04) | 0.859 (± 0.00) |
| 0.50 | 0.587 (± 0.00) | 0.626 (± 0.00) | **0.922** (± **0.00**) | 0.676 (± 0.00) | 0.693 (± 0.00) | 0.864 (± 0.00) | 0.841 (± 0.00) | 0.676 (± 0.03) | 0.721 (± 0.04) | 0.804 (± 0.00) |
| 0.60 | 0.584 (± 0.00) | 0.567 (± 0.00) | **0.881** (± **0.00**) | 0.598 (± 0.00) | 0.614 (± 0.00) | 0.877 (± 0.00) | 0.793 (± 0.00) | 0.597 (± 0.04) | 0.663 (± 0.06) | 0.779 (± 0.00) |
| 0.70 | 0.582 (± 0.00) | 0.506 (± 0.00) | **0.868** (± **0.00**) | 0.548 (± 0.00) | 0.553 (± 0.00) | 0.831 (± 0.00) | 0.771 (± 0.00) | 0.528 (± 0.02) | 0.601 (± 0.06) | 0.766 (± 0.00) |
| 0.80 | 0.592 (± 0.00) | 0.397 (± 0.00) | **0.852** (± **0.00**) | 0.496 (± 0.00) | 0.522 (± 0.00) | 0.807 (± 0.00) | 0.730 (± 0.00) | 0.465 (± 0.03) | 0.509 (± 0.06) | 0.728 (± 0.00) |
| 0.90 | 0.593 (± 0.00) | 0.389 (± 0.00) | **0.744** (± **0.00**) | 0.361 (± 0.00) | 0.423 (± 0.00) | 0.701 (± 0.00) | 0.628 (± 0.00) | 0.407 (± 0.04) | 0.395 (± 0.02) | 0.646 (± 0.00) |
| 0.99 | 0.592 (± 0.00) | 0.289 (± 0.00) | 0.260 (± 0.00) | 0.285 (± 0.00) | 0.282 (± 0.00) | **0.630** (± **0.00**) | 0.333 (± 0.00) | 0.278 (± 0.01) | 0.276 (± 0.01) | 0.412 (± 0.00) |

*Table 34.* F1 scores for ELECTRIC under mechanism *S-MCAR* and varying $\mu$

| $\mu$ | GOODIE | GSPN | FairAC | FP | GNNmi | GCNmf | PCFI | GNNzero | GNNmedian | GNNmim |
|---|---|---|---|---|---|---|---|---|---|---|
| 0.00 | 0.588 (± 0.00) | 0.915 (± 0.00) | **0.963** (± **0.01**) | 0.885 (± 0.00) | 0.929 (± 0.00) | 0.861 (± 0.00) | 0.903 (± 0.00) | 0.909 (± 0.01) | 0.912 (± 0.01) | 0.938 (± 0.01) |
| 0.10 | 0.493 (± 0.00) | 0.891 (± 0.00) | **0.959** (± **0.00**) | 0.831 (± 0.00) | 0.853 (± 0.00) | 0.862 (± 0.00) | 0.854 (± 0.00) | 0.872 (± 0.01) | 0.873 (± 0.02) | 0.904 (± 0.00) |
| 0.20 | 0.484 (± 0.00) | 0.855 (± 0.00) | **0.945** (± **0.00**) | 0.821 (± 0.00) | 0.851 (± 0.00) | 0.867 (± 0.00) | 0.870 (± 0.00) | 0.833 (± 0.01) | 0.842 (± 0.03) | 0.878 (± 0.00) |
| 0.30 | 0.478 (± 0.00) | 0.816 (± 0.00) | **0.935** (± **0.00**) | 0.768 (± 0.00) | 0.796 (± 0.00) | 0.872 (± 0.00) | 0.856 (± 0.00) | 0.776 (± 0.02) | 0.805 (± 0.02) | 0.855 (± 0.00) |
| 0.40 | 0.483 (± 0.00) | 0.756 (± 0.00) | **0.940** (± **0.00**) | 0.703 (± 0.00) | 0.734 (± 0.00) | 0.842 (± 0.00) | 0.871 (± 0.00) | 0.736 (± 0.03) | 0.754 (± 0.01) | 0.801 (± 0.00) |
| 0.50 | 0.431 (± 0.00) | 0.708 (± 0.00) | **0.926** (± **0.00**) | 0.656 (± 0.00) | 0.665 (± 0.00) | 0.839 (± 0.00) | 0.844 (± 0.00) | 0.682 (± 0.02) | 0.712 (± 0.01) | 0.810 (± 0.00) |
| 0.60 | 0.397 (± 0.00) | 0.632 (± 0.00) | **0.898** (± **0.00**) | 0.619 (± 0.00) | 0.617 (± 0.00) | 0.813 (± 0.00) | 0.808 (± 0.00) | 0.627 (± 0.03) | 0.651 (± 0.01) | 0.787 (± 0.00) |
| 0.70 | 0.435 (± 0.00) | 0.563 (± 0.00) | **0.870** (± **0.00**) | 0.528 (± 0.00) | 0.545 (± 0.00) | 0.799 (± 0.00) | 0.776 (± 0.00) | 0.543 (± 0.04) | 0.586 (± 0.05) | 0.711 (± 0.00) |
| 0.80 | 0.490 (± 0.00) | 0.522 (± 0.00) | **0.806** (± **0.00**) | 0.475 (± 0.00) | 0.455 (± 0.00) | 0.764 (± 0.00) | 0.770 (± 0.00) | 0.477 (± 0.03) | 0.493 (± 0.02) | 0.676 (± 0.00) |
| 0.90 | 0.374 (± 0.00) | 0.392 (± 0.00) | **0.771** (± **0.00**) | 0.420 (± 0.00) | 0.394 (± 0.00) | 0.738 (± 0.00) | 0.496 (± 0.00) | 0.374 (± 0.03) | 0.381 (± 0.03) | 0.567 (± 0.00) |
| 0.99 | 0.260 (± 0.00) | 0.265 (± 0.00) | 0.260 (± 0.00) | 0.269 (± 0.00) | 0.277 (± 0.00) | **0.639** (± **0.00**) | 0.285 (± 0.00) | 0.267 (± 0.01) | 0.267 (± 0.01) | 0.479 (± 0.00) |

*Table 35.* F1 scores for ELECTRIC under mechanism *LD-MCAR* and varying $\mu$

| $\mu$ | GOODIE | GSPN | FairAC | FP | GNNmi | GCNmf | PCFI | GNNzero | GNNmedian | GNNmim |
|---|---|---|---|---|---|---|---|---|---|---|
| 0.00 | 0.588 (± 0.00) | 0.915 (± 0.00) | **0.963** (± **0.01**) | 0.885 (± 0.00) | 0.929 (± 0.00) | 0.861 (± 0.00) | 0.903 (± 0.00) | 0.908 (± 0.01) | 0.911 (± 0.01) | 0.920 (± 0.00) |
| 0.10 | 0.585 (± 0.00) | 0.794 (± 0.00) | **0.910** (± **0.00**) | 0.828 (± 0.00) | 0.843 (± 0.00) | 0.890 (± 0.00) | 0.894 (± 0.00) | 0.804 (± 0.03) | 0.804 (± 0.03) | 0.867 (± 0.00) |
| 0.20 | 0.584 (± 0.00) | 0.687 (± 0.00) | **0.920** (± **0.00**) | 0.710 (± 0.00) | 0.762 (± 0.00) | 0.860 (± 0.00) | 0.842 (± 0.00) | 0.804 (± 0.03) | 0.805 (± 0.03) | 0.815 (± 0.00) |
| 0.30 | 0.591 (± 0.00) | 0.604 (± 0.00) | 0.650 (± 0.00) | 0.672 (± 0.00) | 0.693 (± 0.00) | 0.815 (± 0.00) | **0.820** (± **0.00**) | 0.635 (± 0.01) | 0.635 (± 0.01) | 0.793 (± 0.00) |
| 0.40 | 0.587 (± 0.00) | 0.475 (± 0.00) | 0.630 (± 0.00) | 0.475 (± 0.00) | 0.494 (± 0.00) | **0.729** (± **0.00**) | 0.723 (± 0.00) | 0.263 (± 0.01) | 0.263 (± 0.01) | 0.685 (± 0.00) |
| 0.50 | 0.589 (± 0.00) | 0.301 (± 0.00) | **0.630** (± **0.00**) | 0.260 (± 0.00) | 0.260 (± 0.00) | 0.630 (± 0.00) | 0.260 (± 0.00) | 0.265 (± 0.01) | 0.265 (± 0.01) | 0.532 (± 0.00) |
| 0.60 | 0.593 (± 0.00) | 0.271 (± 0.00) | **0.630** (± **0.00**) | 0.260 (± 0.00) | 0.260 (± 0.00) | 0.630 (± 0.00) | 0.260 (± 0.00) | 0.265 (± 0.01) | 0.265 (± 0.01) | 0.517 (± 0.00) |
| 0.70 | 0.589 (± 0.00) | 0.310 (± 0.00) | 0.260 (± 0.00) | 0.260 (± 0.00) | 0.260 (± 0.00) | **0.629** (± **0.00**) | 0.260 (± 0.00) | 0.267 (± 0.01) | 0.267 (± 0.01) | 0.571 (± 0.00) |
| 0.80 | 0.593 (± 0.00) | 0.343 (± 0.00) | 0.260 (± 0.00) | 0.260 (± 0.00) | 0.263 (± 0.00) | **0.630** (± **0.00**) | 0.260 (± 0.00) | 0.260 (± 0.00) | 0.260 (± 0.00) | 0.544 (± 0.00) |
| 0.90 | 0.589 (± 0.00) | 0.315 (± 0.00) | 0.260 (± 0.00) | 0.260 (± 0.00) | 0.263 (± 0.00) | **0.630** (± **0.00**) | 0.260 (± 0.00) | 0.260 (± 0.00) | 0.260 (± 0.00) | 0.538 (± 0.00) |
| 0.99 | 0.589 (± 0.00) | 0.330 (± 0.00) | 0.260 (± 0.00) | 0.260 (± 0.00) | 0.260 (± 0.00) | **0.630** (± **0.00**) | 0.260 (± 0.00) | 0.382 (± 0.01) | 0.423 (± 0.02) | 0.552 (± 0.00) |

*Table 36.* F1 scores for ELECTRIC under mechanism *FD-MNAR* and varying $\mu$

| $\mu$ | GOODIE | GSPN | FairAC | FP | GNNmi | GCMf | PCFI | GNNzero | GNNmedian | GNNmim |
|---|---|---|---|---|---|---|---|---|---|---|
| 0.00 | 0.588 (± 0.00) | 0.915 (± 0.00) | **0.963 (± 0.01)** | 0.885 (± 0.00) | 0.929 (± 0.00) | 0.861 (± 0.00) | 0.903 (± 0.00) | 0.911 (± 0.01) | 0.913 (± 0.01) | 0.938 (± 0.01) |
| 0.10 | 0.468 (± 0.15) | 0.879 (± 0.01) | **0.944 (± 0.02)** | 0.862 (± 0.03) | 0.844 (± 0.02) | 0.878 (± 0.03) | 0.870 (± 0.04) | 0.840 (± 0.03) | 0.851 (± 0.02) | 0.916 (± 0.01) |
| 0.20 | 0.491 (± 0.13) | 0.850 (± 0.01) | **0.938 (± 0.01)** | 0.808 (± 0.02) | 0.813 (± 0.02) | 0.867 (± 0.01) | 0.859 (± 0.02) | 0.789 (± 0.02) | 0.802 (± 0.03) | 0.906 (± 0.00) |
| 0.30 | 0.496 (± 0.13) | 0.800 (± 0.02) | **0.922 (± 0.03)** | 0.744 (± 0.02) | 0.793 (± 0.01) | 0.864 (± 0.03) | 0.861 (± 0.01) | 0.727 (± 0.02) | 0.798 (± 0.03) | 0.877 (± 0.01) |
| 0.40 | 0.506 (± 0.12) | 0.772 (± 0.04) | **0.906 (± 0.03)** | 0.701 (± 0.03) | 0.751 (± 0.03) | 0.850 (± 0.02) | 0.839 (± 0.01) | 0.674 (± 0.02) | 0.726 (± 0.03) | 0.864 (± 0.01) |
| 0.50 | 0.438 (± 0.12) | 0.743 (± 0.01) | **0.877 (± 0.01)** | 0.648 (± 0.03) | 0.707 (± 0.02) | 0.842 (± 0.02) | 0.817 (± 0.03) | 0.642 (± 0.05) | 0.699 (± 0.08) | 0.837 (± 0.02) |
| 0.60 | 0.331 (± 0.05) | 0.688 (± 0.02) | **0.836 (± 0.03)** | 0.594 (± 0.02) | 0.663 (± 0.01) | 0.807 (± 0.05) | 0.775 (± 0.02) | 0.590 (± 0.03) | 0.607 (± 0.03) | 0.806 (± 0.01) |
| 0.70 | 0.461 (± 0.14) | 0.626 (± 0.01) | **0.834 (± 0.04)** | 0.514 (± 0.04) | 0.590 (± 0.02) | 0.776 (± 0.02) | 0.761 (± 0.02) | 0.482 (± 0.03) | 0.433 (± 0.09) | 0.760 (± 0.02) |
| 0.80 | 0.435 (± 0.12) | 0.570 (± 0.02) | 0.742 (± 0.06) | 0.463 (± 0.01) | 0.490 (± 0.04) | **0.743 (± 0.04)** | 0.700 (± 0.02) | 0.436 (± 0.01) | 0.328 (± 0.04) | 0.707 (± 0.01) |
| 0.90 | 0.275 (± 0.01) | 0.484 (± 0.01) | 0.560 (± 0.22) | 0.330 (± 0.06) | 0.426 (± 0.03) | **0.663 (± 0.03)** | 0.500 (± 0.17) | 0.352 (± 0.03) | 0.276 (± 0.01) | 0.620 (± 0.02) |
| 0.99 | 0.342 (± 0.12) | 0.377 (± 0.03) | 0.260 (± 0.00) | 0.260 (± 0.00) | 0.347 (± 0.01) | **0.629 (± 0.00)** | 0.274 (± 0.01) | 0.286 (± 0.02) | 0.273 (± 0.01) | 0.537 (± 0.08) |

*Table 37.* F1 scores for ELECTRIC under mechanism *CD-MNAR* and varying $\mu$

| $\mu$ | GOODIE | GSPN | FairAC | FP | GNNmi | GCMf | PCFI | GNNzero | GNNmedian | GNNmim |
|---|---|---|---|---|---|---|---|---|---|---|
| 0.00 | 0.588 (± 0.00) | 0.915 (± 0.00) | **0.963 (± 0.01)** | 0.885 (± 0.00) | 0.929 (± 0.00) | 0.861 (± 0.00) | 0.903 (± 0.00) | 0.908 (± 0.01) | 0.908 (± 0.01) | 0.938 (± 0.01) |
| 0.10 | 0.486 (± 0.12) | 0.888 (± 0.01) | **0.962 (± 0.01)** | 0.869 (± 0.01) | 0.874 (± 0.01) | 0.908 (± 0.02) | 0.885 (± 0.03) | 0.839 (± 0.03) | 0.860 (± 0.02) | 0.922 (± 0.00) |
| 0.20 | 0.476 (± 0.15) | 0.851 (± 0.02) | **0.931 (± 0.02)** | 0.815 (± 0.03) | 0.802 (± 0.01) | 0.879 (± 0.01) | 0.879 (± 0.01) | 0.805 (± 0.03) | 0.801 (± 0.00) | 0.902 (± 0.03) |
| 0.30 | 0.478 (± 0.16) | 0.819 (± 0.04) | **0.922 (± 0.00)** | 0.789 (± 0.03) | 0.789 (± 0.01) | 0.872 (± 0.01) | 0.880 (± 0.01) | 0.770 (± 0.05) | 0.736 (± 0.01) | 0.890 (± 0.00) |
| 0.40 | 0.431 (± 0.11) | 0.807 (± 0.02) | **0.902 (± 0.01)** | 0.775 (± 0.01) | 0.762 (± 0.01) | 0.835 (± 0.02) | 0.865 (± 0.02) | 0.749 (± 0.03) | 0.685 (± 0.05) | 0.869 (± 0.02) |
| 0.50 | 0.450 (± 0.09) | 0.758 (± 0.02) | **0.867 (± 0.03)** | 0.722 (± 0.02) | 0.748 (± 0.01) | 0.835 (± 0.03) | 0.827 (± 0.03) | 0.656 (± 0.03) | 0.633 (± 0.04) | 0.850 (± 0.02) |
| 0.60 | 0.436 (± 0.10) | 0.706 (± 0.01) | **0.853 (± 0.05)** | 0.663 (± 0.02) | 0.608 (± 0.01) | 0.847 (± 0.03) | 0.780 (± 0.02) | 0.664 (± 0.03) | 0.593 (± 0.04) | 0.836 (± 0.02) |
| 0.70 | 0.337 (± 0.03) | 0.604 (± 0.03) | **0.812 (± 0.03)** | 0.585 (± 0.03) | 0.538 (± 0.03) | 0.770 (± 0.09) | 0.729 (± 0.01) | 0.560 (± 0.02) | 0.514 (± 0.02) | 0.765 (± 0.01) |
| 0.80 | 0.411 (± 0.09) | 0.594 (± 0.02) | **0.824 (± 0.08)** | 0.540 (± 0.01) | 0.486 (± 0.01) | 0.703 (± 0.04) | 0.671 (± 0.01) | 0.513 (± 0.01) | 0.469 (± 0.02) | 0.742 (± 0.02) |
| 0.90 | 0.392 (± 0.11) | 0.531 (± 0.02) | **0.735 (± 0.07)** | 0.473 (± 0.03) | 0.449 (± 0.02) | 0.686 (± 0.06) | 0.600 (± 0.04) | 0.434 (± 0.02) | 0.445 (± 0.04) | 0.683 (± 0.02) |
| 0.99 | 0.304 (± 0.02) | 0.329 (± 0.04) | 0.264 (± 0.01) | 0.303 (± 0.02) | 0.294 (± 0.01) | **0.629 (± 0.00)** | 0.312 (± 0.04) | 0.305 (± 0.02) | 0.292 (± 0.02) | 0.561 (± 0.02) |

*Table 38.* F1 scores for TADPOLE under mechanism *U-MCAR* and varying $\mu$

| $\mu$ | GOODIE | GSPN | FairAC | FP | GNNmi | GCMf | PCFI | GNNzero | GNNmedian | GNNmim |
|---|---|---|---|---|---|---|---|---|---|---|
| 0.00 | 0.804 (± 0.00) | 0.648 (± 0.01) | 0.790 (± 0.00) | 0.806 (± 0.00) | 0.832 (± 0.02) | 0.786 (± 0.00) | 0.792 (± 0.00) | **0.847 (± 0.03)** | 0.847 (± 0.03) | 0.809 (± 0.00) |
| 0.10 | 0.789 (± 0.00) | 0.590 (± 0.00) | 0.795 (± 0.00) | 0.801 (± 0.00) | 0.832 (± 0.00) | 0.809 (± 0.00) | 0.821 (± 0.00) | **0.841 (± 0.03)** | 0.837 (± 0.03) | 0.820 (± 0.00) |
| 0.20 | 0.808 (± 0.00) | 0.590 (± 0.00) | 0.803 (± 0.00) | 0.823 (± 0.00) | **0.836 (± 0.00)** | 0.779 (± 0.00) | 0.802 (± 0.00) | 0.833 (± 0.03) | 0.827 (± 0.04) | 0.799 (± 0.00) |
| 0.30 | 0.814 (± 0.00) | 0.567 (± 0.00) | 0.791 (± 0.00) | 0.806 (± 0.00) | **0.825 (± 0.00)** | 0.757 (± 0.00) | 0.803 (± 0.00) | 0.811 (± 0.03) | 0.813 (± 0.03) | 0.802 (± 0.00) |
| 0.40 | 0.804 (± 0.00) | 0.610 (± 0.00) | **0.831 (± 0.00)** | 0.800 (± 0.00) | 0.820 (± 0.00) | 0.794 (± 0.00) | 0.799 (± 0.00) | 0.830 (± 0.01) | 0.819 (± 0.02) | 0.805 (± 0.00) |
| 0.50 | 0.752 (± 0.00) | 0.581 (± 0.00) | 0.813 (± 0.00) | 0.809 (± 0.00) | **0.830 (± 0.00)** | 0.799 (± 0.00) | 0.810 (± 0.00) | 0.797 (± 0.03) | 0.790 (± 0.03) | 0.814 (± 0.00) |
| 0.60 | 0.756 (± 0.00) | 0.575 (± 0.00) | 0.808 (± 0.00) | 0.785 (± 0.00) | 0.797 (± 0.00) | 0.722 (± 0.00) | 0.791 (± 0.00) | **0.810 (± 0.05)** | 0.771 (± 0.04) | 0.799 (± 0.00) |
| 0.70 | 0.610 (± 0.00) | 0.552 (± 0.00) | 0.795 (± 0.00) | 0.740 (± 0.00) | 0.772 (± 0.00) | 0.729 (± 0.00) | 0.762 (± 0.00) | 0.779 (± 0.04) | 0.767 (± 0.03) | **0.802 (± 0.00)** |
| 0.80 | 0.669 (± 0.00) | 0.552 (± 0.00) | **0.804 (± 0.00)** | 0.757 (± 0.00) | 0.728 (± 0.00) | 0.669 (± 0.00) | 0.775 (± 0.00) | 0.760 (± 0.05) | 0.736 (± 0.04) | 0.764 (± 0.00) |
| 0.90 | 0.759 (± 0.00) | 0.590 (± 0.00) | 0.241 (± 0.00) | 0.758 (± 0.00) | 0.408 (± 0.00) | 0.608 (± 0.00) | 0.767 (± 0.00) | **0.786 (± 0.02)** | 0.704 (± 0.02) | 0.763 (± 0.00) |
| 0.99 | **0.707 (± 0.00)** | 0.523 (± 0.00) | 0.241 (± 0.00) | 0.241 (± 0.00) | 0.241 (± 0.00) | 0.353 (± 0.00) | 0.241 (± 0.00) | 0.507 (± 0.22) | 0.241 (± 0.00) | 0.700 (± 0.00) |

*Table 39.* F1 scores for TADPOLE under mechanism *S-MCAR* and varying $\mu$

| $\mu$ | GOODIE | GSPN | FairAC | FP | GNNmi | GCMf | PCFI | GNNzero | GNNmedian | GNNmim |
|---|---|---|---|---|---|---|---|---|---|---|
| 0.00 | 0.804 (± 0.00) | 0.648 (± 0.01) | 0.790 (± 0.00) | 0.806 (± 0.00) | 0.832 (± 0.02) | 0.786 (± 0.00) | 0.792 (± 0.00) | **0.847 (± 0.03)** | 0.847 (± 0.03) | 0.831 (± 0.04) |
| 0.10 | 0.554 (± 0.00) | 0.542 (± 0.00) | 0.805 (± 0.00) | 0.803 (± 0.00) | 0.815 (± 0.00) | 0.751 (± 0.00) | 0.804 (± 0.00) | **0.848 (± 0.02)** | 0.846 (± 0.02) | 0.810 (± 0.00) |
| 0.20 | 0.497 (± 0.00) | 0.486 (± 0.00) | 0.818 (± 0.00) | 0.818 (± 0.00) | 0.811 (± 0.00) | 0.737 (± 0.00) | 0.814 (± 0.00) | **0.846 (± 0.02)** | 0.845 (± 0.03) | 0.794 (± 0.00) |
| 0.30 | 0.523 (± 0.00) | 0.502 (± 0.00) | 0.775 (± 0.00) | 0.799 (± 0.00) | 0.825 (± 0.00) | 0.777 (± 0.00) | 0.818 (± 0.00) | 0.837 (± 0.02) | **0.838 (± 0.02)** | 0.775 (± 0.00) |
| 0.40 | 0.482 (± 0.00) | 0.581 (± 0.00) | 0.800 (± 0.00) | 0.797 (± 0.00) | 0.794 (± 0.00) | 0.719 (± 0.00) | 0.784 (± 0.00) | 0.820 (± 0.03) | **0.823 (± 0.04)** | 0.790 (± 0.00) |
| 0.50 | 0.501 (± 0.00) | 0.523 (± 0.00) | 0.757 (± 0.00) | 0.777 (± 0.00) | 0.769 (± 0.00) | 0.739 (± 0.00) | 0.798 (± 0.00) | **0.803 (± 0.02)** | 0.797 (± 0.00) | 0.795 (± 0.00) |
| 0.60 | 0.539 (± 0.00) | 0.498 (± 0.00) | 0.802 (± 0.00) | 0.769 (± 0.00) | 0.734 (± 0.00) | 0.693 (± 0.00) | 0.804 (± 0.00) | 0.804 (± 0.05) | 0.799 (± 0.04) | **0.816 (± 0.00)** |
| 0.70 | 0.480 (± 0.00) | 0.453 (± 0.00) | 0.748 (± 0.00) | 0.719 (± 0.00) | 0.738 (± 0.00) | 0.642 (± 0.00) | 0.752 (± 0.00) | 0.784 (± 0.03) | 0.777 (± 0.05) | **0.795 (± 0.00)** |
| 0.80 | 0.502 (± 0.00) | 0.422 (± 0.00) | 0.689 (± 0.00) | 0.736 (± 0.00) | 0.703 (± 0.00) | 0.555 (± 0.00) | 0.730 (± 0.00) | 0.739 (± 0.02) | 0.740 (± 0.06) | **0.812 (± 0.00)** |
| 0.90 | 0.377 (± 0.00) | 0.280 (± 0.00) | 0.503 (± 0.00) | 0.680 (± 0.00) | 0.650 (± 0.00) | 0.420 (± 0.00) | 0.739 (± 0.00) | 0.662 (± 0.07) | 0.557 (± 0.06) | **0.742 (± 0.00)** |
| 0.99 | 0.272 (± 0.00) | 0.249 (± 0.00) | 0.241 (± 0.00) | **0.384 (± 0.00)** | 0.241 (± 0.00) | 0.241 (± 0.00) | 0.241 (± 0.00) | 0.323 (± 0.05) | 0.241 (± 0.00) | 0.370 (± 0.00) |

*Table 40.* F1 scores for TADPOLE under mechanism *LD-MCAR* and varying $\mu$

| $\mu$ | GOODIE | GSPN | FairAC | FP | GNNmi | GCNmf | PCFI | GNNzero | GNNmedian | GNNmim |
|---|---|---|---|---|---|---|---|---|---|---|
| 0.00 | 0.804 ($\pm$ 0.00) | 0.648 ($\pm$ 0.01) | 0.790 ($\pm$ 0.00) | 0.806 ($\pm$ 0.00) | 0.832 ($\pm$ 0.02) | 0.786 ($\pm$ 0.00) | 0.792 ($\pm$ 0.00) | **0.847** ($\pm$ **0.03**) | 0.847 ($\pm$ 0.03) | 0.831 ($\pm$ 0.04) |
| 0.10 | 0.786 ($\pm$ 0.00) | 0.550 ($\pm$ 0.00) | 0.765 ($\pm$ 0.00) | 0.760 ($\pm$ 0.00) | 0.793 ($\pm$ 0.00) | 0.789 ($\pm$ 0.00) | 0.785 ($\pm$ 0.00) | 0.809 ($\pm$ 0.03) | 0.809 ($\pm$ 0.03) | **0.815** ($\pm$ **0.00**) |
| 0.20 | 0.785 ($\pm$ 0.00) | 0.462 ($\pm$ 0.00) | 0.758 ($\pm$ 0.00) | 0.777 ($\pm$ 0.00) | 0.786 ($\pm$ 0.00) | 0.763 ($\pm$ 0.00) | 0.804 ($\pm$ 0.00) | **0.810** ($\pm$ **0.04**) | 0.810 ($\pm$ 0.04) | 0.806 ($\pm$ 0.00) |
| 0.30 | 0.654 ($\pm$ 0.00) | 0.517 ($\pm$ 0.00) | 0.766 ($\pm$ 0.00) | 0.788 ($\pm$ 0.00) | 0.784 ($\pm$ 0.00) | 0.779 ($\pm$ 0.00) | 0.782 ($\pm$ 0.00) | **0.802** ($\pm$ **0.04**) | 0.802 ($\pm$ 0.04) | 0.800 ($\pm$ 0.00) |
| 0.40 | 0.685 ($\pm$ 0.00) | 0.550 ($\pm$ 0.00) | 0.780 ($\pm$ 0.00) | 0.764 ($\pm$ 0.00) | 0.780 ($\pm$ 0.00) | 0.779 ($\pm$ 0.00) | 0.774 ($\pm$ 0.00) | **0.795** ($\pm$ **0.03**) | 0.795 ($\pm$ 0.03) | 0.780 ($\pm$ 0.00) |
| 0.50 | 0.778 ($\pm$ 0.00) | 0.558 ($\pm$ 0.00) | 0.700 ($\pm$ 0.00) | 0.728 ($\pm$ 0.00) | 0.776 ($\pm$ 0.00) | 0.746 ($\pm$ 0.00) | 0.731 ($\pm$ 0.00) | 0.773 ($\pm$ 0.04) | 0.773 ($\pm$ 0.04) | **0.785** ($\pm$ **0.00**) |
| 0.60 | **0.783** ($\pm$ **0.00**) | 0.508 ($\pm$ 0.00) | 0.731 ($\pm$ 0.00) | 0.708 ($\pm$ 0.00) | 0.729 ($\pm$ 0.00) | 0.760 ($\pm$ 0.00) | 0.714 ($\pm$ 0.00) | 0.767 ($\pm$ 0.03) | 0.767 ($\pm$ 0.03) | 0.745 ($\pm$ 0.00) |
| 0.70 | 0.725 ($\pm$ 0.00) | 0.545 ($\pm$ 0.00) | 0.684 ($\pm$ 0.00) | 0.638 ($\pm$ 0.00) | 0.663 ($\pm$ 0.00) | 0.704 ($\pm$ 0.00) | 0.710 ($\pm$ 0.00) | **0.739** ($\pm$ **0.03**) | 0.739 ($\pm$ 0.03) | 0.722 ($\pm$ 0.00) |
| 0.80 | **0.656** ($\pm$ **0.00**) | 0.442 ($\pm$ 0.00) | 0.576 ($\pm$ 0.00) | 0.391 ($\pm$ 0.00) | 0.442 ($\pm$ 0.00) | 0.543 ($\pm$ 0.00) | 0.419 ($\pm$ 0.00) | 0.643 ($\pm$ 0.04) | 0.643 ($\pm$ 0.04) | 0.615 ($\pm$ 0.00) |
| 0.90 | **0.704** ($\pm$ **0.00**) | 0.419 ($\pm$ 0.00) | 0.241 ($\pm$ 0.00) | 0.348 ($\pm$ 0.00) | 0.361 ($\pm$ 0.00) | 0.337 ($\pm$ 0.00) | 0.292 ($\pm$ 0.00) | 0.327 ($\pm$ 0.03) | 0.327 ($\pm$ 0.03) | 0.409 ($\pm$ 0.00) |
| 0.99 | 0.687 ($\pm$ 0.00) | 0.402 ($\pm$ 0.00) | 0.241 ($\pm$ 0.00) | 0.348 ($\pm$ 0.00) | 0.361 ($\pm$ 0.00) | 0.337 ($\pm$ 0.00) | 0.292 ($\pm$ 0.00) | **0.730** ($\pm$ **0.03**) | 0.567 ($\pm$ 0.17) | 0.409 ($\pm$ 0.00) |

*Table 41.* F1 scores for TADPOLE under mechanism *FD-MNAR* and varying $\mu$

| $\mu$ | GOODIE | GSPN | FairAC | FP | GNNmi | GCNmf | PCFI | GNNzero | GNNmedian | GNNmim |
|---|---|---|---|---|---|---|---|---|---|---|
| 0.00 | 0.804 ($\pm$ 0.00) | 0.648 ($\pm$ 0.01) | 0.790 ($\pm$ 0.00) | 0.806 ($\pm$ 0.00) | 0.832 ($\pm$ 0.02) | 0.786 ($\pm$ 0.00) | 0.792 ($\pm$ 0.00) | 0.846 ($\pm$ 0.03) | **0.849** ($\pm$ **0.03**) | 0.831 ($\pm$ 0.04) |
| 0.10 | 0.546 ($\pm$ 0.07) | 0.643 ($\pm$ 0.01) | 0.801 ($\pm$ 0.01) | 0.797 ($\pm$ 0.01) | 0.822 ($\pm$ 0.02) | 0.830 ($\pm$ 0.04) | 0.838 ($\pm$ 0.03) | 0.841 ($\pm$ 0.03) | 0.842 ($\pm$ 0.03) | **0.846** ($\pm$ **0.04**) |
| 0.20 | 0.531 ($\pm$ 0.11) | 0.624 ($\pm$ 0.05) | 0.793 ($\pm$ 0.04) | **0.836** ($\pm$ **0.01**) | 0.810 ($\pm$ 0.01) | 0.832 ($\pm$ 0.02) | 0.827 ($\pm$ 0.01) | 0.832 ($\pm$ 0.03) | 0.817 ($\pm$ 0.03) | 0.796 ($\pm$ 0.00) |
| 0.30 | 0.573 ($\pm$ 0.12) | 0.580 ($\pm$ 0.04) | 0.804 ($\pm$ 0.05) | 0.811 ($\pm$ 0.03) | 0.806 ($\pm$ 0.04) | 0.829 ($\pm$ 0.04) | **0.831** ($\pm$ **0.02**) | 0.827 ($\pm$ 0.03) | 0.802 ($\pm$ 0.03) | 0.828 ($\pm$ 0.03) |
| 0.40 | 0.562 ($\pm$ 0.09) | 0.615 ($\pm$ 0.03) | 0.751 ($\pm$ 0.03) | 0.803 ($\pm$ 0.04) | 0.793 ($\pm$ 0.04) | **0.811** ($\pm$ **0.02**) | 0.802 ($\pm$ 0.03) | 0.806 ($\pm$ 0.03) | 0.803 ($\pm$ 0.03) | 0.781 ($\pm$ 0.02) |
| 0.50 | 0.673 ($\pm$ 0.04) | 0.646 ($\pm$ 0.07) | 0.793 ($\pm$ 0.02) | 0.789 ($\pm$ 0.02) | 0.796 ($\pm$ 0.05) | 0.780 ($\pm$ 0.01) | **0.815** ($\pm$ **0.03**) | 0.809 ($\pm$ 0.04) | 0.805 ($\pm$ 0.04) | 0.784 ($\pm$ 0.03) |
| 0.60 | 0.529 ($\pm$ 0.09) | 0.633 ($\pm$ 0.06) | 0.722 ($\pm$ 0.07) | 0.805 ($\pm$ 0.04) | 0.785 ($\pm$ 0.05) | 0.758 ($\pm$ 0.02) | **0.810** ($\pm$ **0.03**) | 0.803 ($\pm$ 0.04) | 0.792 ($\pm$ 0.04) | 0.795 ($\pm$ 0.03) |
| 0.70 | 0.634 ($\pm$ 0.05) | 0.571 ($\pm$ 0.04) | **0.804** ($\pm$ **0.03**) | 0.795 ($\pm$ 0.04) | 0.746 ($\pm$ 0.06) | 0.720 ($\pm$ 0.06) | 0.795 ($\pm$ 0.05) | 0.776 ($\pm$ 0.05) | 0.748 ($\pm$ 0.05) | 0.780 ($\pm$ 0.03) |
| 0.80 | 0.378 ($\pm$ 0.10) | 0.590 ($\pm$ 0.06) | 0.612 ($\pm$ 0.14) | 0.785 ($\pm$ 0.02) | 0.692 ($\pm$ 0.05) | 0.708 ($\pm$ 0.02) | **0.797** ($\pm$ **0.03**) | 0.776 ($\pm$ 0.04) | 0.720 ($\pm$ 0.05) | 0.765 ($\pm$ 0.00) |
| 0.90 | 0.309 ($\pm$ 0.10) | 0.597 ($\pm$ 0.01) | 0.241 ($\pm$ 0.00) | 0.771 ($\pm$ 0.03) | 0.663 ($\pm$ 0.05) | 0.719 ($\pm$ 0.01) | **0.787** ($\pm$ **0.02**) | 0.779 ($\pm$ 0.03) | 0.703 ($\pm$ 0.05) | 0.777 ($\pm$ 0.06) |
| 0.99 | 0.241 ($\pm$ 0.00) | 0.600 ($\pm$ 0.05) | 0.241 ($\pm$ 0.00) | 0.736 ($\pm$ 0.03) | 0.241 ($\pm$ 0.00) | 0.584 ($\pm$ 0.05) | 0.241 ($\pm$ 0.00) | 0.733 ($\pm$ 0.01) | 0.241 ($\pm$ 0.00) | **0.794** ($\pm$ **0.04**) |

*Table 42.* F1 scores for TADPOLE under mechanism *CD-MNAR* and varying $\mu$

| $\mu$ | GOODIE | GSPN | FairAC | FP | GNNmi | GCNmf | PCFI | GNNzero | GNNmedian | GNNmim |
|---|---|---|---|---|---|---|---|---|---|---|
| 0.00 | 0.804 ($\pm$ 0.00) | 0.648 ($\pm$ 0.01) | 0.790 ($\pm$ 0.00) | 0.806 ($\pm$ 0.00) | 0.832 ($\pm$ 0.02) | 0.786 ($\pm$ 0.00) | 0.792 ($\pm$ 0.00) | **0.847** ($\pm$ **0.03**) | 0.847 ($\pm$ 0.03) | 0.809 ($\pm$ 0.00) |
| 0.10 | 0.553 ($\pm$ 0.06) | 0.534 ($\pm$ 0.09) | 0.793 ($\pm$ 0.05) | 0.813 ($\pm$ 0.03) | 0.829 ($\pm$ 0.04) | 0.792 ($\pm$ 0.03) | 0.806 ($\pm$ 0.03) | **0.842** ($\pm$ **0.02**) | 0.826 ($\pm$ 0.04) | 0.803 ($\pm$ 0.01) |
| 0.20 | 0.485 ($\pm$ 0.06) | 0.515 ($\pm$ 0.04) | 0.804 ($\pm$ 0.03) | 0.812 ($\pm$ 0.03) | 0.832 ($\pm$ 0.03) | 0.810 ($\pm$ 0.02) | 0.806 ($\pm$ 0.02) | **0.849** ($\pm$ **0.01**) | 0.826 ($\pm$ 0.04) | 0.815 ($\pm$ 0.02) |
| 0.30 | 0.441 ($\pm$ 0.02) | 0.584 ($\pm$ 0.06) | 0.805 ($\pm$ 0.03) | 0.785 ($\pm$ 0.02) | 0.811 ($\pm$ 0.03) | 0.786 ($\pm$ 0.02) | 0.812 ($\pm$ 0.02) | **0.828** ($\pm$ **0.03**) | 0.813 ($\pm$ 0.04) | 0.827 ($\pm$ 0.03) |
| 0.40 | 0.502 ($\pm$ 0.07) | 0.671 ($\pm$ 0.03) | 0.828 ($\pm$ 0.01) | 0.818 ($\pm$ 0.03) | 0.808 ($\pm$ 0.02) | 0.793 ($\pm$ 0.02) | 0.814 ($\pm$ 0.03) | 0.824 ($\pm$ 0.02) | 0.826 ($\pm$ 0.03) | **0.830** ($\pm$ **0.01**) |
| 0.50 | 0.448 ($\pm$ 0.02) | 0.621 ($\pm$ 0.04) | 0.784 ($\pm$ 0.02) | 0.804 ($\pm$ 0.04) | 0.799 ($\pm$ 0.04) | 0.756 ($\pm$ 0.03) | 0.803 ($\pm$ 0.05) | 0.819 ($\pm$ 0.02) | 0.800 ($\pm$ 0.04) | **0.828** ($\pm$ **0.04**) |
| 0.60 | 0.457 ($\pm$ 0.01) | 0.529 ($\pm$ 0.03) | 0.791 ($\pm$ 0.01) | 0.781 ($\pm$ 0.03) | 0.803 ($\pm$ 0.03) | 0.710 ($\pm$ 0.07) | 0.797 ($\pm$ 0.03) | **0.823** ($\pm$ **0.04**) | 0.783 ($\pm$ 0.03) | 0.792 ($\pm$ 0.03) |
| 0.70 | 0.485 ($\pm$ 0.07) | 0.590 ($\pm$ 0.09) | 0.639 ($\pm$ 0.29) | 0.797 ($\pm$ 0.05) | 0.787 ($\pm$ 0.04) | 0.710 ($\pm$ 0.05) | **0.822** ($\pm$ **0.02**) | 0.813 ($\pm$ 0.03) | 0.784 ($\pm$ 0.07) | 0.818 ($\pm$ 0.01) |
| 0.80 | 0.376 ($\pm$ 0.10) | 0.605 ($\pm$ 0.04) | 0.434 ($\pm$ 0.27) | 0.785 ($\pm$ 0.05) | 0.767 ($\pm$ 0.09) | 0.744 ($\pm$ 0.01) | 0.798 ($\pm$ 0.05) | **0.819** ($\pm$ **0.04**) | 0.776 ($\pm$ 0.04) | 0.800 ($\pm$ 0.02) |
| 0.90 | 0.362 ($\pm$ 0.09) | 0.563 ($\pm$ 0.03) | 0.241 ($\pm$ 0.00) | 0.788 ($\pm$ 0.01) | 0.730 ($\pm$ 0.08) | 0.689 ($\pm$ 0.05) | 0.776 ($\pm$ 0.06) | 0.771 ($\pm$ 0.06) | 0.704 ($\pm$ 0.05) | **0.803** ($\pm$ **0.05**) |
| 0.99 | 0.324 ($\pm$ 0.12) | 0.547 ($\pm$ 0.08) | 0.241 ($\pm$ 0.00) | 0.255 ($\pm$ 0.02) | 0.241 ($\pm$ 0.00) | 0.348 ($\pm$ 0.05) | 0.241 ($\pm$ 0.00) | 0.558 ($\pm$ 0.15) | 0.241 ($\pm$ 0.00) | **0.652** ($\pm$ **0.04**) |

## H. Complete Result Tables – R2 Regime

This appendix complements the analysis of Research Question 3 (Section 4). It reports the complete set of results for the R2 regime, where training and test data are subject to different missingness mechanisms. We include both numerical tables (F1-score mean ± std over 5 runs) and extended visualizations across all models and datasets.

### H.1. Numerical Results

Table 43 reports the full F1-scores for all models, datasets, and shift configurations considered in the R2 regime.

*Table 43.* F1 (mean ± std over 5 runs). Setup: **R2** missingness distribution shift, where training data are subject to either *FD-MNAR* or *CD-MNAR*, while test data have either no missingness, 25% or 50% of *U-MCAR*

| Task | Train mech. | μ Test | GOODIE | GSPN | FairAC | GCNmf | PCFI | FP | GNNmi | GNNzero | GNNmedian | GNNmim |
|------|-------------|--------|--------|------|--------|-------|------|-----|-------|---------|-----------|--------|
| SYNTHETIC | *FD-MNAR* | 0 | 0.50 (± 0.15) | 0.68 (± 0.01) | 0.69 (± 0.05) | 0.81 (± 0.01) | 0.79 (± 0.02) | 0.80 (± 0.01) | 0.80 (± 0.01) | 0.81 (± 0.02) | 0.80 (± 0.02) | **0.82 (± 0.01)** |
| | *FD-MNAR* | 0.25 | 0.47 (± 0.13) | 0.64 (± 0.03) | 0.69 (± 0.04) | 0.74 (± 0.03) | 0.75 (± 0.03) | 0.76 (± 0.03) | 0.75 (± 0.03) | 0.76 (± 0.01) | 0.76 (± 0.02) | **0.77 (± 0.03)** |
| | *FD-MNAR* | 0.50 | 0.47 (± 0.13) | 0.64 (± 0.02) | 0.65 (± 0.04) | 0.71 (± 0.03) | 0.73 (± 0.02) | 0.71 (± 0.02) | 0.74 (± 0.02) | 0.71 (± 0.03) | 0.72 (± 0.04) | **0.73 (± 0.02)** |
| | *CD-MNAR* | 0 | 0.71 (± 0.07) | 0.70 (± 0.03) | 0.70 (± 0.05) | 0.80 (± 0.04) | 0.81 (± 0.02) | 0.80 (± 0.02) | 0.78 (± 0.02) | 0.82 (± 0.02) | 0.76 (± 0.02) | **0.85 (± 0.04)** |
| | *CD-MNAR* | 0.25 | 0.66 (± 0.05) | 0.68 (± 0.05) | 0.68 (± 0.03) | 0.75 (± 0.06) | 0.78 (± 0.04) | 0.77 (± 0.04) | 0.77 (± 0.02) | 0.78 (± 0.03) | 0.72 (± 0.03) | **0.80 (± 0.03)** |
| | *CD-MNAR* | 0.50 | 0.56 (± 0.10) | 0.64 (± 0.04) | 0.65 (± 0.01) | 0.73 (± 0.02) | 0.72 (± 0.03) | 0.72 (± 0.05) | 0.72 (± 0.01) | 0.72 (± 0.04) | 0.70 (± 0.01) | **0.75 (± 0.03)** |
| AIR | *FD-MNAR* | 0 | 0.50 (± 0.14) | 0.33 (± 0.04) | 0.66 (± 0.07) | 0.83 (± 0.05) | **0.88 (± 0.01)** | 0.86 (± 0.03) | 0.86 (± 0.03) | 0.85 (± 0.01) | 0.84 (± 0.03) | 0.87 (± 0.02) |
| | *FD-MNAR* | 0.25 | 0.51 (± 0.12) | 0.42 (± 0.04) | 0.65 (± 0.08) | 0.68 (± 0.05) | 0.83 (± 0.05) | 0.81 (± 0.02) | 0.81 (± 0.01) | 0.83 (± 0.01) | 0.80 (± 0.02) | **0.85 (± 0.01)** |
| | *FD-MNAR* | 0.50 | 0.52 (± 0.11) | 0.55 (± 0.03) | 0.70 (± 0.03) | 0.71 (± 0.03) | **0.80 (± 0.07)** | 0.79 (± 0.06) | 0.79 (± 0.05) | 0.78 (± 0.04) | 0.78 (± 0.01) | 0.80 (± 0.05) |
| | *CD-MNAR* | 0 | 0.56 (± 0.16) | 0.35 (± 0.02) | 0.65 (± 0.08) | 0.60 (± 0.20) | **0.88 (± 0.01)** | 0.71 (± 0.07) | 0.86 (± 0.06) | 0.83 (± 0.07) | 0.82 (± 0.03) | 0.85 (± 0.00) |
| | *CD-MNAR* | 0.25 | 0.56 (± 0.16) | 0.45 (± 0.50) | 0.70 (± 0.05) | 0.70 (± 0.05) | 0.84 (± 0.05) | 0.75 (± 0.05) | 0.84 (± 0.04) | 0.80 (± 0.05) | 0.79 (± 0.03) | **0.84 (± 0.06)** |
| | *CD-MNAR* | 0.50 | 0.62 (± 0.07) | 0.47 (± 0.04) | 0.68 (± 0.07) | 0.70 (± 0.02) | **0.80 (± 0.05)** | 0.72 (± 0.03) | 0.76 (± 0.05) | 0.76 (± 0.01) | 0.74 (± 0.03) | 0.76 (± 0.02) |
| ELECTRIC | *FD-MNAR* | 0 | 0.45 (± 0.11) | 0.67 (± 0.11) | **0.92 (± 0.02)** | 0.88 (± 0.12) | 0.69 (± 0.00) | 0.76 (± 0.03) | 0.80 (± 0.02) | 0.83 (± 0.05) | 0.79 (± 0.01) | 0.92 (± 0.01) |
| | *FD-MNAR* | 0.25 | 0.53 (± 0.10) | 0.68 (± 0.06) | **0.89 (± 0.00)** | 0.80 (± 0.02) | 0.73 (± 0.03) | 0.69 (± 0.03) | 0.74 (± 0.02) | 0.76 (± 0.03) | 0.73 (± 0.04) | 0.87 (± 0.01) |
| | *FD-MNAR* | 0.50 | 0.50 (± 0.10) | 0.68 (± 0.01) | **0.90 (± 0.02)** | 0.83 (± 0.01) | 0.75 (± 0.03) | 0.62 (± 0.02) | 0.66 (± 0.03) | 0.68 (± 0.02) | 0.66 (± 0.02) | 0.82 (± 0.02) |
| | *CD-MNAR* | 0 | 0.52 (± 0.10) | 0.78 (± 0.04) | 0.92 (± 0.02) | 0.86 (± 0.01) | 0.88 (± 0.01) | 0.83 (± 0.05) | 0.81 (± 0.01) | 0.81 (± 0.01) | 0.79 (± 0.02) | **0.94 (± 0.00)** |
| | *CD-MNAR* | 0.25 | 0.50 (± 0.10) | 0.78 (± 0.01) | **0.88 (± 0.01)** | 0.86 (± 0.02) | 0.85 (± 0.02) | 0.74 (± 0.04) | 0.73 (± 0.03) | 0.72 (± 0.01) | 0.73 (± 0.02) | 0.85 (± 0.03) |
| | *CD-MNAR* | 0.50 | 0.49 (± 0.12) | 0.70 (± 0.02) | **0.87 (± 0.02)** | 0.82 (± 0.03) | 0.81 (± 0.00) | 0.66 (± 0.01) | 0.70 (± 0.03) | 0.65 (± 0.02) | 0.68 (± 0.02) | 0.83 (± 0.02) |
| TADPOLE | *FD-MNAR* | 0 | 0.52 (± 0.07) | 0.53 (± 0.00) | 0.75 (± 0.03) | 0.74 (± 0.05) | 0.79 (± 0.00) | 0.77 (± 0.00) | 0.76 (± 0.01) | 0.79 (± 0.01) | 0.77 (± 0.02) | **0.83 (± 0.02)** |
| | *FD-MNAR* | 0.25 | 0.48 (± 0.03) | 0.48 (± 0.02) | 0.77 (± 0.01) | 0.73 (± 0.01) | **0.82 (± 0.02)** | 0.78 (± 0.03) | 0.76 (± 0.03) | 0.78 (± 0.03) | 0.74 (± 0.03) | 0.81 (± 0.01) |
| | *FD-MNAR* | 0.50 | 0.48 (± 0.04) | 0.53 (± 0.02) | 0.79 (± 0.02) | 0.71 (± 0.04) | 0.78 (± 0.02) | 0.74 (± 0.02) | 0.73 (± 0.03) | 0.74 (± 0.04) | 0.71 (± 0.02) | **0.82 (± 0.03)** |
| | *CD-MNAR* | 0 | 0.60 (± 0.02) | 0.26 (± 0.02) | 0.79 (± 0.06) | 0.75 (± 0.04) | **0.80 (± 0.04)** | 0.80 (± 0.03) | 0.79 (± 0.05) | 0.79 (± 0.04) | 0.75 (± 0.04) | 0.79 (± 0.06) |
| | *CD-MNAR* | 0.25 | 0.47 (± 0.09) | 0.52 (± 0.02) | 0.82 (± 0.05) | 0.78 (± 0.01) | **0.80 (± 0.04)** | **0.80 (± 0.04)** | 0.77 (± 0.04) | 0.78 (± 0.04) | 0.73 (± 0.06) | 0.75 (± 0.03) |
| | *CD-MNAR* | 0.50 | 0.49 (± 0.07) | 0.62 (± 0.05) | 0.81 (± 0.03) | 0.75 (± 0.00) | 0.79 (± 0.01) | **0.82 (± 0.02)** | 0.76 (± 0.03) | 0.76 (± 0.05) | 0.73 (± 0.06) | 0.74 (± 0.02) |

### H.2. Extended Visualizations

In addition to Figure 3 in the main paper, Figures 6 and 7 report the full results for all models under both training mechanisms.

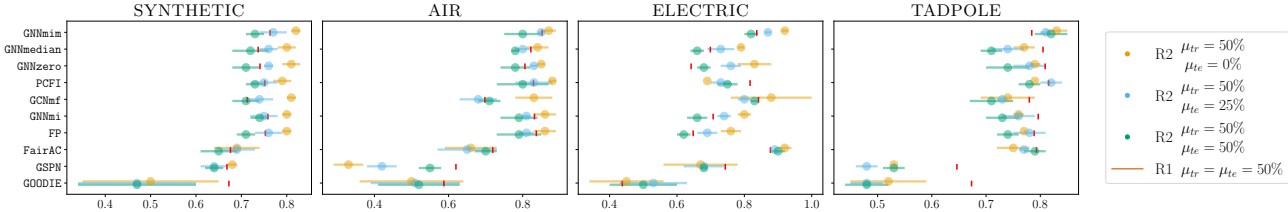

*Figure 6.* Full results for all models trained with *FD-MNAR* at $\mu_{tr} = 50\%$, tested on *U-MCAR* with $\mu_{te} \in \{0\%, 25\%, 50\%\}$. Each panel corresponds to one dataset; each row to one model. Reported values are mean ± std over 5 runs.

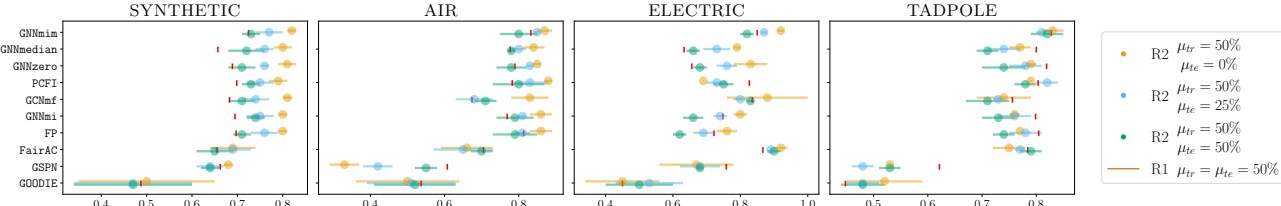

*Figure 7.* Full results for all models trained with *CD-MNAR* at $\mu_{\text{tr}} = 50\%$, tested on *U-MCAR* with $\mu_{\text{te}} \in \{0\%, 25\%, 50\%\}$. Same layout as Figure 6.

# I. Inductive Synthetic Setting

In addition to the transductive experiments reported in the main paper, we also ran a set of experiments in an inductive setting to demonstrate that our model, GNNmim, is not restricted to transductive scenarios. As shown in Figure 8, GNNmim remains competitive with all other baselines even under this inductive setup.

*Figure 8.* Performance of GNNmim and all competitors in an inductive setting. The synthetic dataset is constructed so that test nodes form a separate graph component and are never connected to training nodes, ensuring that no message can propagate between the two sets during training. Despite this strictly inductive setup, GNNmim remains competitive with all baselines.

*Table 44.* F1 scores for INDUCTIVE under mechanism *CDMNAR* and varying $\mu$

| $\mu$ | GOODIE | GSPN | FairAC | FP | GNNmi | GCNmf | PCFI | GNNzero | GNNmedian | GNNmim |
|---|---|---|---|---|---|---|---|---|---|---|
| 0.00 | 0.687 (± 0.166) | 0.713 (± 0.045) | 0.367 (± 0.000) | **0.972** (± **0.011**) | 0.968 (± 0.011) | 0.867 (± 0.023) | 0.970 (± 0.011) | 0.968 (± 0.011) | 0.968 (± 0.011) | 0.967 (± 0.011) |
| 0.10 | 0.672 (± 0.167) | 0.708 (± 0.022) | 0.367 (± 0.000) | 0.880 (± 0.014) | 0.881 (± 0.014) | 0.842 (± 0.010) | 0.876 (± 0.011) | 0.875 (± 0.018) | 0.878 (± 0.020) | **0.883** (± **0.020**) |
| 0.20 | 0.639 (± 0.151) | 0.686 (± 0.048) | 0.367 (± 0.000) | 0.836 (± 0.015) | 0.838 (± 0.022) | 0.796 (± 0.026) | 0.825 (± 0.018) | 0.840 (± 0.022) | **0.842** (± **0.020**) | 0.832 (± 0.019) |
| 0.30 | 0.595 (± 0.122) | 0.636 (± 0.031) | 0.367 (± 0.000) | 0.785 (± 0.020) | 0.785 (± 0.034) | 0.765 (± 0.036) | 0.782 (± 0.023) | 0.796 (± 0.029) | 0.793 (± 0.026) | **0.801** (± **0.020**) |
| 0.40 | 0.598 (± 0.119) | 0.631 (± 0.043) | 0.367 (± 0.000) | 0.734 (± 0.019) | 0.758 (± 0.024) | 0.729 (± 0.021) | 0.731 (± 0.008) | 0.754 (± 0.017) | 0.750 (± 0.023) | **0.759** (± **0.017**) |
| 0.50 | 0.442 (± 0.092) | 0.589 (± 0.029) | 0.367 (± 0.000) | 0.643 (± 0.036) | 0.628 (± 0.040) | 0.647 (± 0.041) | 0.616 (± 0.029) | 0.668 (± 0.023) | 0.632 (± 0.030) | **0.680** (± **0.018**) |
| 0.60 | 0.473 (± 0.063) | 0.605 (± 0.034) | 0.367 (± 0.000) | 0.629 (± 0.031) | 0.597 (± 0.029) | 0.649 (± 0.041) | 0.600 (± 0.052) | 0.687 (± 0.013) | 0.602 (± 0.033) | **0.704** (± **0.021**) |
| 0.70 | 0.401 (± 0.070) | 0.592 (± 0.024) | 0.367 (± 0.000) | 0.574 (± 0.016) | 0.562 (± 0.007) | 0.599 (± 0.064) | 0.471 (± 0.041) | 0.656 (± 0.023) | 0.566 (± 0.018) | **0.664** (± **0.027**) |
| 0.80 | 0.377 (± 0.012) | 0.584 (± 0.026) | 0.367 (± 0.000) | 0.571 (± 0.026) | 0.551 (± 0.020) | 0.567 (± 0.044) | 0.463 (± 0.069) | 0.634 (± 0.025) | 0.557 (± 0.016) | **0.638** (± **0.028**) |
| 0.90 | 0.402 (± 0.062) | 0.592 (± 0.031) | 0.367 (± 0.000) | 0.574 (± 0.048) | 0.544 (± 0.020) | 0.548 (± 0.052) | 0.458 (± 0.046) | 0.650 (± 0.033) | 0.547 (± 0.028) | **0.657** (± **0.020**) |
| 0.99 | 0.395 (± 0.052) | 0.444 (± 0.060) | 0.367 (± 0.000) | 0.380 (± 0.022) | 0.467 (± 0.020) | 0.395 (± 0.035) | 0.367 (± 0.000) | **0.524** (± **0.045**) | 0.464 (± 0.013) | **0.524** (± **0.045**) |

*Table 45.* F1 scores for INDUCTIVE under mechanism *FDMNAR* and varying $\mu$

| $\mu$ | GOODIE | GSPN | FairAC | FP | GNNmi | GCNmf | PCFI | GNNzero | GNNmedian | GNNmim |
|---|---|---|---|---|---|---|---|---|---|---|
| 0.00 | 0.687 (± 0.166) | 0.708 (± 0.045) | 0.367 (± 0.000) | **0.972** (± **0.011**) | 0.967 (± 0.011) | 0.867 (± 0.022) | 0.968 (± 0.013) | 0.967 (± 0.011) | 0.967 (± 0.011) | 0.968 (± 0.011) |
| 0.10 | 0.679 (± 0.166) | 0.711 (± 0.012) | 0.367 (± 0.000) | 0.888 (± 0.013) | 0.879 (± 0.024) | 0.847 (± 0.013) | 0.885 (± 0.014) | 0.882 (± 0.022) | 0.886 (± 0.020) | **0.889** (± **0.017**) |
| 0.20 | 0.646 (± 0.154) | 0.686 (± 0.033) | 0.367 (± 0.000) | **0.834** (± **0.024**) | 0.825 (± 0.024) | 0.799 (± 0.016) | 0.832 (± 0.026) | 0.830 (± 0.022) | 0.825 (± 0.025) | 0.826 (± 0.028) |
| 0.30 | 0.569 (± 0.133) | 0.649 (± 0.013) | 0.367 (± 0.000) | **0.800** (± **0.042**) | 0.786 (± 0.034) | 0.772 (± 0.028) | 0.796 (± 0.025) | 0.789 (± 0.036) | 0.782 (± 0.032) | 0.793 (± 0.036) |
| 0.40 | 0.522 (± 0.134) | 0.608 (± 0.037) | 0.367 (± 0.000) | 0.759 (± 0.021) | **0.761** (± **0.027**) | 0.732 (± 0.032) | 0.753 (± 0.026) | 0.757 (± 0.032) | 0.743 (± 0.028) | 0.742 (± 0.032) |
| 0.50 | 0.492 (± 0.135) | 0.618 (± 0.008) | 0.367 (± 0.000) | 0.714 (± 0.016) | 0.731 (± 0.015) | 0.692 (± 0.027) | 0.710 (± 0.028) | 0.724 (± 0.017) | **0.736** (± **0.018**) | 0.730 (± 0.019) |
| 0.60 | 0.433 (± 0.084) | 0.575 (± 0.025) | 0.367 (± 0.000) | 0.675 (± 0.031) | 0.699 (± 0.032) | 0.676 (± 0.022) | 0.674 (± 0.039) | 0.702 (± 0.030) | 0.687 (± 0.027) | **0.716** (± **0.031**) |
| 0.70 | 0.464 (± 0.090) | 0.582 (± 0.020) | 0.367 (± 0.000) | 0.630 (± 0.031) | 0.643 (± 0.037) | 0.594 (± 0.040) | 0.623 (± 0.035) | 0.651 (± 0.037) | 0.635 (± 0.033) | **0.661** (± **0.019**) |
| 0.80 | 0.429 (± 0.065) | 0.540 (± 0.009) | 0.367 (± 0.000) | 0.586 (± 0.021) | 0.598 (± 0.027) | 0.527 (± 0.053) | 0.560 (± 0.030) | 0.607 (± 0.029) | 0.609 (± 0.019) | **0.620** (± **0.024**) |
| 0.90 | 0.444 (± 0.082) | 0.522 (± 0.034) | 0.367 (± 0.000) | 0.508 (± 0.105) | 0.558 (± 0.049) | 0.486 (± 0.061) | 0.460 (± 0.129) | 0.589 (± 0.042) | 0.575 (± 0.044) | **0.592** (± **0.023**) |
| 0.99 | 0.370 (± 0.005) | 0.538 (± 0.041) | 0.367 (± 0.000) | 0.433 (± 0.093) | 0.515 (± 0.035) | 0.454 (± 0.076) | 0.420 (± 0.105) | **0.561** (± **0.036**) | 0.521 (± 0.040) | 0.550 (± 0.040) |

*Table 46.* F1 scores for INDUCTIVE under mechanism *LDMCAR* and varying $\mu$

| $\mu$ | GOODIE | GSPN | FairAC | FP | GNNmi | GCNmf | PCFI | GNNzero | GNNmedian | GNNmim |
|---|---|---|---|---|---|---|---|---|---|---|
| 0.00 | 0.687 (± 0.166) | 0.713 (± 0.045) | 0.367 (± 0.000) | **0.972** (± **0.011**) | 0.968 (± 0.011) | 0.867 (± 0.023) | 0.970 (± 0.011) | 0.968 (± 0.011) | 0.968 (± 0.011) | 0.967 (± 0.011) |
| 0.10 | 0.687 (± 0.166) | 0.713 (± 0.045) | 0.367 (± 0.000) | **0.972** (± **0.011**) | 0.968 (± 0.011) | 0.867 (± 0.023) | 0.970 (± 0.011) | 0.968 (± 0.011) | 0.968 (± 0.011) | 0.967 (± 0.011) |
| 0.20 | 0.494 (± 0.117) | 0.601 (± 0.039) | 0.367 (± 0.000) | 0.701 (± 0.023) | 0.692 (± 0.031) | 0.673 (± 0.036) | **0.705** (± **0.019**) | 0.692 (± 0.031) | 0.692 (± 0.031) | 0.696 (± 0.029) |
| 0.30 | 0.415 (± 0.076) | 0.537 (± 0.032) | 0.367 (± 0.000) | 0.596 (± 0.010) | **0.624** (± **0.010**) | 0.539 (± 0.028) | 0.606 (± 0.006) | **0.624** (± **0.010**) | **0.624** (± **0.010**) | 0.615 (± 0.011) |
| 0.40 | 0.415 (± 0.076) | 0.543 (± 0.037) | 0.367 (± 0.000) | 0.596 (± 0.010) | **0.624** (± **0.010**) | 0.539 (± 0.028) | 0.606 (± 0.006) | **0.624** (± **0.010**) | **0.624** (± **0.010**) | 0.615 (± 0.011) |
| 0.50 | 0.415 (± 0.076) | 0.537 (± 0.032) | 0.367 (± 0.000) | 0.596 (± 0.010) | **0.624** (± **0.010**) | 0.539 (± 0.028) | 0.606 (± 0.006) | **0.624** (± **0.010**) | **0.624** (± **0.010**) | 0.615 (± 0.011) |
| 0.60 | 0.409 (± 0.053) | 0.495 (± 0.044) | 0.367 (± 0.000) | 0.497 (± 0.015) | **0.555** (± **0.019**) | 0.498 (± 0.022) | 0.501 (± 0.022) | **0.555** (± **0.019**) | **0.555** (± **0.019**) | 0.552 (± 0.027) |
| 0.70 | 0.398 (± 0.037) | 0.428 (± 0.030) | 0.367 (± 0.000) | 0.410 (± 0.027) | 0.524 (± 0.044) | 0.407 (± 0.051) | 0.407 (± 0.025) | 0.524 (± 0.044) | 0.524 (± 0.044) | **0.538** (± **0.023**) |
| 0.80 | 0.398 (± 0.037) | 0.428 (± 0.030) | 0.367 (± 0.000) | 0.410 (± 0.027) | 0.524 (± 0.044) | 0.407 (± 0.051) | 0.407 (± 0.025) | 0.524 (± 0.044) | 0.524 (± 0.044) | **0.538** (± **0.023**) |
| 0.90 | 0.398 (± 0.037) | 0.428 (± 0.030) | 0.367 (± 0.000) | 0.410 (± 0.027) | 0.524 (± 0.044) | 0.407 (± 0.051) | 0.407 (± 0.025) | 0.524 (± 0.044) | 0.524 (± 0.044) | **0.538** (± **0.023**) |
| 0.99 | 0.433 (± 0.069) | 0.549 (± 0.024) | 0.367 (± 0.000) | 0.637 (± 0.036) | 0.659 (± 0.029) | 0.587 (± 0.031) | 0.623 (± 0.027) | **0.660** (± **0.025**) | 0.652 (± 0.025) | 0.658 (± 0.023) |

*Table 47.* F1 scores for INDUCTIVE under mechanism *SMCAR* and varying $\mu$

| $\mu$ | GOODIE | GSPN | FairAC | FP | GNNmi | GCNmf | PCFI | GNNzero | GNNmedian | GNNmim |
|---|---|---|---|---|---|---|---|---|---|---|
| 0.00 | 0.687 (± 0.166) | 0.713 (± 0.045) | 0.367 (± 0.000) | **0.972** (± **0.011**) | 0.968 (± 0.011) | 0.867 (± 0.023) | 0.970 (± 0.011) | 0.968 (± 0.011) | 0.968 (± 0.011) | 0.967 (± 0.011) |
| 0.10 | 0.661 (± 0.148) | 0.687 (± 0.013) | 0.434 (± 0.133) | 0.887 (± 0.012) | **0.894** (± **0.016**) | 0.847 (± 0.025) | 0.891 (± 0.018) | **0.894** (± **0.017**) | 0.890 (± 0.021) | 0.881 (± 0.018) |
| 0.20 | 0.667 (± 0.157) | 0.675 (± 0.036) | 0.367 (± 0.000) | 0.850 (± 0.017) | 0.855 (± 0.027) | 0.820 (± 0.030) | **0.856** (± **0.025**) | 0.847 (± 0.018) | 0.851 (± 0.027) | 0.851 (± 0.028) |
| 0.30 | 0.664 (± 0.155) | 0.679 (± 0.034) | 0.367 (± 0.000) | **0.830** (± **0.016**) | 0.829 (± 0.032) | 0.804 (± 0.028) | 0.822 (± 0.025) | 0.828 (± 0.032) | 0.824 (± 0.034) | 0.827 (± 0.038) |
| 0.40 | 0.557 (± 0.152) | 0.650 (± 0.029) | 0.367 (± 0.000) | 0.785 (± 0.030) | 0.796 (± 0.035) | 0.769 (± 0.018) | 0.785 (± 0.029) | 0.785 (± 0.043) | 0.790 (± 0.039) | **0.802** (± **0.032**) |
| 0.50 | 0.521 (± 0.152) | 0.633 (± 0.045) | 0.367 (± 0.000) | 0.757 (± 0.029) | 0.758 (± 0.018) | 0.735 (± 0.019) | 0.748 (± 0.030) | **0.760** (± **0.018**) | 0.755 (± 0.019) | 0.756 (± 0.009) |
| 0.60 | 0.497 (± 0.135) | 0.636 (± 0.058) | 0.367 (± 0.000) | **0.742** (± **0.030**) | 0.722 (± 0.034) | 0.698 (± 0.021) | 0.723 (± 0.038) | 0.724 (± 0.039) | 0.716 (± 0.031) | 0.730 (± 0.027) |
| 0.70 | 0.461 (± 0.125) | 0.580 (± 0.062) | 0.367 (± 0.000) | 0.670 (± 0.018) | 0.671 (± 0.029) | 0.631 (± 0.036) | 0.666 (± 0.038) | **0.673** (± **0.030**) | 0.672 (± 0.028) | 0.666 (± 0.035) |
| 0.80 | 0.509 (± 0.121) | 0.549 (± 0.071) | 0.367 (± 0.000) | 0.628 (± 0.053) | **0.629** (± **0.025**) | 0.563 (± 0.070) | 0.621 (± 0.044) | 0.623 (± 0.013) | 0.622 (± 0.025) | 0.625 (± 0.037) |
| 0.90 | 0.402 (± 0.071) | 0.455 (± 0.068) | 0.367 (± 0.000) | 0.487 (± 0.070) | 0.580 (± 0.043) | 0.447 (± 0.060) | 0.474 (± 0.092) | **0.597** (± **0.026**) | 0.575 (± 0.039) | 0.580 (± 0.027) |
| 0.99 | 0.367 (± 0.000) | 0.372 (± 0.010) | 0.367 (± 0.000) | 0.367 (± 0.000) | 0.486 (± 0.027) | 0.380 (± 0.019) | 0.367 (± 0.000) | **0.509** (± **0.038**) | 0.476 (± 0.024) | 0.498 (± 0.031) |

*Table 48.* F1 scores for INDUCTIVE under mechanism *UMCAR* and varying $\mu$

| $\mu$ | GOODIE | GSPN | FairAC | FP | GNNmi | GCNmf | PCFI | GNNzero | GNNmedian | GNNmim |
|---|---|---|---|---|---|---|---|---|---|---|
| 0.00 | 0.715 (± 0.096) | 0.705 (± 0.033) | 0.414 (± 0.055) | **0.960** (± **0.009**) | 0.953 (± 0.006) | 0.811 (± 0.030) | **0.960** (± **0.009**) | 0.953 (± 0.006) | 0.953 (± 0.006) | 0.944 (± 0.017) |
| 0.10 | 0.572 (± 0.137) | 0.658 (± 0.031) | 0.412 (± 0.057) | 0.827 (± 0.050) | 0.851 (± 0.043) | 0.769 (± 0.112) | 0.810 (± 0.034) | **0.855** (± **0.044**) | 0.846 (± 0.047) | 0.841 (± 0.051) |
| 0.20 | 0.596 (± 0.165) | 0.638 (± 0.025) | 0.379 (± 0.000) | 0.798 (± 0.033) | **0.799** (± **0.020**) | 0.756 (± 0.032) | 0.788 (± 0.027) | 0.790 (± 0.028) | 0.788 (± 0.021) | 0.785 (± 0.021) |
| 0.30 | 0.594 (± 0.145) | 0.625 (± 0.014) | 0.359 (± 0.040) | **0.771** (± **0.037**) | 0.757 (± 0.046) | 0.674 (± 0.133) | 0.712 (± 0.045) | 0.758 (± 0.049) | **0.771** (± **0.042**) | 0.718 (± 0.047) |
| 0.40 | 0.596 (± 0.132) | 0.625 (± 0.005) | 0.379 (± 0.000) | **0.721** (± **0.055**) | 0.702 (± 0.044) | 0.702 (± 0.055) | 0.664 (± 0.080) | 0.697 (± 0.049) | 0.701 (± 0.048) | 0.718 (± 0.029) |
| 0.50 | 0.487 (± 0.113) | 0.583 (± 0.040) | 0.379 (± 0.000) | 0.608 (± 0.067) | 0.660 (± 0.027) | 0.664 (± 0.053) | 0.568 (± 0.074) | 0.659 (± 0.021) | **0.674** (± **0.022**) | 0.633 (± 0.035) |
| 0.60 | 0.439 (± 0.118) | 0.558 (± 0.034) | 0.379 (± 0.000) | 0.572 (± 0.077) | 0.617 (± 0.038) | 0.606 (± 0.081) | 0.469 (± 0.102) | 0.617 (± 0.038) | **0.622** (± **0.039**) | **0.622** (± **0.062**) |
| 0.70 | 0.390 (± 0.074) | **0.561** (± **0.019**) | 0.379 (± 0.000) | 0.451 (± 0.092) | 0.534 (± 0.073) | 0.511 (± 0.095) | 0.476 (± 0.118) | 0.518 (± 0.076) | 0.541 (± 0.089) | 0.502 (± 0.052) |
| 0.80 | 0.418 (± 0.123) | 0.499 (± 0.029) | 0.379 (± 0.000) | 0.459 (± 0.074) | **0.530** (± **0.060**) | 0.508 (± 0.088) | 0.392 (± 0.087) | 0.490 (± 0.059) | 0.528 (± 0.044) | 0.473 (± 0.052) |
| 0.90 | 0.340 (± 0.048) | 0.493 (± 0.022) | 0.379 (± 0.000) | 0.367 (± 0.046) | **0.550** (± **0.139**) | 0.511 (± 0.082) | 0.362 (± 0.041) | 0.532 (± 0.134) | 0.529 (± 0.131) | 0.501 (± 0.122) |
| 0.99 | 0.341 (± 0.045) | 0.400 (± 0.025) | 0.379 (± 0.000) | 0.379 (± 0.000) | 0.472 (± 0.022) | 0.380 (± 0.003) | 0.384 (± 0.011) | 0.476 (± 0.038) | **0.485** (± **0.018**) | 0.483 (± 0.033) |

# J. Gain using MIM with competitors

Tables 49 through 52 report the performance gain observed when all competitor models described in the main paper are equipped with the MIM mask, mirroring the setup used for `GNNmim`. Consistently, basic imputation methods that replace missing features with a constant, such as `GNNmi` and `GNNmedian`, show a positive and comparable performance increase when supplied with the same mask. This suggests that the improvement comes from the model's ability to selectively ignore the filled or imputed feature values indicated by the mask.

*Table 49.* F1 gain from using mask on SYNTHETIC under mechanism *U-MCAR*

| $\mu$ | FairAC | FP | GCNmf | GNNmedian | GNNmi | GOODIE | GSPN | PCFI | GNNzero |
|------|--------|--------|--------|-----------|--------|--------|--------|--------|---------|
| 0.00 | -0.087 | -0.016 | -0.145 | 0.002 | 0.003 | -0.256 | -0.094 | -0.020 | 0.005 |
| 0.10 | -0.094 | -0.022 | -0.065 | 0.006 | 0.005 | -0.253 | -0.080 | -0.004 | 0.001 |
| 0.20 | -0.102 | -0.013 | -0.005 | 0.002 | 0.004 | -0.215 | -0.052 | -0.001 | 0.008 |
| 0.30 | -0.078 | 0.002 | -0.021 | 0.012 | 0.014 | -0.198 | -0.068 | -0.008 | 0.015 |
| 0.40 | -0.082 | 0.008 | -0.022 | 0.012 | 0.07 | -0.223 | -0.075 | 0.006 | 0.025 |
| 0.50 | 0.011 | -0.006 | -0.010 | 0.005 | 0.09 | -0.268 | -0.079 | -0.018 | 0.007 |
| 0.60 | -0.025 | -0.004 | -0.029 | 0.004 | 0.013 | -0.346 | -0.072 | -0.001 | 0.000 |
| 0.70 | 0.013 | 0.001 | -0.044 | 0.005 | 0.004 | -0.321 | -0.008 | 0.006 | 0.006 |
| 0.80 | -0.070 | -0.008 | 0.009 | 0.002 | 0.015 | -0.429 | 0.015 | -0.014 | 0.039 |
| 0.90 | -0.020 | -0.017 | -0.011 | 0.011 | 0.014 | -0.346 | 0.053 | 0.001 | 0.001 |
| 0.99 | 0.052 | -0.007 | 0.056 | -0.020 | -0.013 | -0.422 | 0.024 | -0.011 | -0.013 |

*Table 50.* F1 gain from using mask on SYNTHETIC under mechanism *S-MCAR*

| $\mu$ | FairAC | FP | GCNmf | GNNmedian | GNNmi | GOODIE | GSPN | PCFI | GNNzero |
|------|--------|--------|--------|-----------|--------|--------|--------|--------|---------|
| 0.00 | -0.080 | -0.016 | -0.145 | 0.002 | 0.003 | -0.256 | -0.091 | 0.05 | 0.005 |
| 0.10 | 0.013 | 0.001 | -0.077 | 0.03 | 0.04 | -0.211 | 0.005 | -0.11 | -0.011 |
| 0.20 | -0.018 | -0.039 | -0.086 | 0.003 | 0.007 | -0.245 | -0.019 | -0.026 | 0.031 |
| 0.30 | 0.000 | -0.026 | -0.083 | 0.006 | 0.015 | -0.234 | -0.013 | -0.015 | 0.016 |
| 0.40 | 0.010 | -0.034 | -0.012 | 0.002 | 0.019 | -0.185 | -0.014 | -0.018 | 0.024 |
| 0.50 | -0.062 | -0.048 | 0.005 | 0.006 | 0.016 | -0.207 | -0.036 | -0.039 | 0.033 |
| 0.60 | -0.045 | -0.028 | -0.038 | 0.018 | 0.032 | -0.161 | 0.001 | -0.026 | 0.038 |
| 0.70 | 0.009 | -0.007 | -0.025 | 0.011 | 0.025 | -0.153 | -0.015 | -0.033 | 0.064 |
| 0.80 | 0.010 | -0.011 | -0.046 | 0.011 | 0.02 | -0.136 | -0.002 | 0.004 | 0.029 |
| 0.90 | -0.045 | 0.003 | -0.018 | 0.002 | -0.002 | -0.071 | 0.043 | -0.000 | -0.019 |
| 0.99 | 0.128 | -0.024 | 0.074 | 0.002 | -0.015 | 0.048 | 0.033 | -0.025 | -0.011 |

*Table 51.* F1 gain from using mask on SYNTHETIC under mechanism *LD-MCAR*

| $\mu$ | FairAC | FP | GCNmf | GNNmedian | GNNmi | GOODIE | GSPN | PCFI | GNNzero |
|------|--------|--------|--------|-----------|--------|--------|--------|--------|---------|
| 0.00 | -0.073 | -0.016 | -0.145 | 0.002 | 0.003 | -0.256 | -0.094 | -0.020 | 0.005 |
| 0.10 | -0.047 | 0.104 | -0.012 | 0.026 | 0.095 | -0.222 | -0.014 | 0.097 | -0.08 |
| 0.20 | -0.105 | -0.078 | -0.081 | 0.004 | 0.075 | -0.251 | -0.092 | -0.067 | 0.081 |
| 0.30 | -0.106 | -0.119 | -0.106 | 0.015 | 0.101 | -0.331 | -0.091 | -0.118 | 0.133 |
| 0.40 | 0.014 | -0.044 | -0.049 | 0.015 | 0.039 | -0.337 | -0.054 | -0.033 | 0.098 |
| 0.50 | 0.080 | -0.002 | -0.004 | 0.015 | 0.002 | -0.362 | 0.027 | 0.003 | 0.077 |
| 0.60 | -0.079 | -0.073 | -0.068 | 0.004 | 0.081 | -0.386 | -0.046 | -0.069 | 0.139 |
| 0.70 | -0.111 | -0.084 | -0.034 | 0.001 | 0.070 | -0.423 | -0.039 | -0.060 | 0.139 |
| 0.80 | 0.001 | -0.084 | -0.074 | 0.001 | 0.085 | -0.422 | -0.056 | -0.086 | 0.130 |
| 0.90 | -0.067 | -0.090 | -0.066 | 0.001 | 0.096 | -0.439 | 0.023 | -0.072 | 0.143 |
| 0.99 | 0.046 | 0.037 | -0.054 | 0.007 | 0.014 | -0.359 | 0.025 | 0.039 | 0.020 |

*Table 52.* F1 gain from using mask on SYNTHETIC under mechanism *FD-MNAR*

| $\mu$ | FairAC | FP | GCNmf | GNNmedian | GNNmi | GOODIE | GSPN | PCFI | GNNzero |
|---|---|---|---|---|---|---|---|---|---|
| 0.00 | -0.080 | -0.018 | -0.141 | 0.002 | 0.003 | -0.256 | -0.081 | -0.018 | 0.005 |
| 0.10 | -0.035 | -0.006 | -0.057 | 0.007 | 0.013 | -0.216 | -0.002 | -0.001 | 0.014 |
| 0.20 | 0.018 | 0.015 | 0.024 | 0.06 | 0.005 | -0.193 | -0.012 | -0.009 | -0.005 |
| 0.30 | 0.021 | 0.002 | -0.005 | 0.002 | 0.007 | -0.138 | 0.015 | 0.016 | -0.018 |
| 0.40 | 0.001 | -0.007 | -0.031 | 0.006 | 0.011 | -0.186 | -0.032 | 0.003 | 0.021 |
| 0.50 | -0.025 | -0.011 | -0.020 | 0.008 | 0.013 | -0.208 | -0.009 | -0.007 | 0.022 |
| 0.60 | 0.011 | 0.006 | -0.019 | 0.012 | 0.008 | -0.121 | 0.030 | 0.013 | 0.013 |
| 0.70 | 0.022 | 0.029 | 0.004 | 0.000 | 0.003 | -0.063 | 0.044 | 0.013 | 0.012 |
| 0.80 | 0.010 | 0.013 | -0.010 | 0.002 | 0.001 | -0.006 | -0.017 | 0.033 | 0.020 |
| 0.90 | 0.053 | 0.032 | -0.032 | 0.005 | 0.011 | 0.048 | -0.023 | 0.020 | 0.018 |
| 0.99 | 0.156 | 0.002 | -0.008 | 0.001 | 0.010 | -0.015 | 0.006 | -0.003 | 0.007 |

*Table 53.* F1 gain from using mask on SYNTHETIC under mechanism *CD-MNAR*

| $\mu$ | FairAC | FP | GCNmf | GNNmedian | GNNmi | GOODIE | GSPN | PCFI | GNNzero |
|---|---|---|---|---|---|---|---|---|---|
| 0.00 | -0.078 | -0.016 | -0.145 | 0.002 | 0.003 | -0.256 | -0.091 | -0.020 | 0.005 |
| 0.10 | -0.025 | -0.002 | -0.060 | 0.004 | 0.010 | -0.239 | -0.019 | -0.005 | 0.001 |
| 0.20 | 0.023 | 0.006 | -0.003 | 0.004 | 0.002 | -0.202 | -0.029 | 0.004 | 0.001 |
| 0.30 | -0.005 | 0.017 | -0.004 | 0.009 | 0.007 | -0.121 | -0.030 | -0.006 | 0.023 |
| 0.40 | -0.045 | 0.017 | -0.015 | 0.014 | 0.017 | 0.005 | -0.024 | 0.021 | 0.020 |
| 0.50 | -0.035 | 0.010 | 0.001 | 0.048 | 0.010 | -0.035 | -0.042 | 0.009 | 0.036 |
| 0.60 | 0.054 | 0.036 | -0.011 | 0.019 | 0.015 | -0.111 | -0.047 | 0.073 | 0.037 |
| 0.70 | 0.038 | 0.051 | 0.001 | 0.025 | 0.028 | -0.064 | 0.031 | 0.072 | 0.026 |
| 0.80 | 0.045 | 0.046 | 0.047 | 0.017 | 0.011 | -0.028 | -0.021 | 0.086 | 0.037 |
| 0.90 | 0.136 | 0.033 | 0.039 | 0.011 | 0.021 | -0.009 | -0.047 | 0.075 | 0.037 |
| 0.99 | 0.098 | -0.041 | 0.057 | 0.017 | 0.015 | -0.050 | 0.044 | 0.013 | 0.018 |

*Table 54.* F1 (mean ± std over 5 runs). Setup: **Reverse R2** missingness distribution shift, where training data are subject to *U-MCAR* ($\mu_{tr} = 0.5$), while test data have either no missingness, 25% or 50% of *FD-MNAR* or *CD-MNAR*.

| Task | Test mech. | $\mu$ Test | GOODIE | GSPN | FairAC | GCNmf | PCFI | FP | GNNmi | GNNzero | GNNmedian | GNNmim |
|---|---|---|---|---|---|---|---|---|---|---|---|---|
| SYNTHETIC | *FD-MNAR* | 0 | 0.61 (± 0.15) | 0.75 (± 0.04) | 0.64 (± 0.12) | 0.78 (± 0.04) | 0.85 (± 0.02) | 0.85 (± 0.02) | 0.86 (± 0.02) | 0.86 (± 0.02) | 0.86 (± 0.02) | **0.87 (± 0.02)** |
| | *FD-MNAR* | 0.25 | 0.57 (± 0.11) | 0.70 (± 0.04) | 0.62 (± 0.11) | 0.74 (± 0.02) | 0.79 (± 0.01) | 0.79 (± 0.02) | 0.78 (± 0.02) | 0.78 (± 0.01) | 0.79 (± 0.01) | **0.79 (± 0.02)** |
| | *FD-MNAR* | 0.50 | 0.55 (± 0.13) | 0.66 (± 0.01) | 0.60 (± 0.12) | 0.69 (± 0.03) | 0.75 (± 0.02) | 0.74 (± 0.03) | 0.75 (± 0.01) | **0.75 (± 0.02)** | 0.75 (± 0.02) | 0.75 (± 0.02) |
| | *CD-MNAR* | 0 | 0.64 (± 0.11) | 0.74 (± 0.04) | 0.69 (± 0.02) | 0.77 (± 0.06) | 0.86 (± 0.02) | 0.85 (± 0.02) | 0.87 (± 0.01) | **0.88 (± 0.02)** | 0.87 (± 0.01) | 0.87 (± 0.03) |
| | *CD-MNAR* | 0.25 | 0.61 (± 0.09) | 0.69 (± 0.03) | 0.67 (± 0.03) | 0.73 (± 0.04) | 0.79 (± 0.02) | 0.78 (± 0.03) | 0.79 (± 0.03) | 0.79 (± 0.03) | 0.80 (± 0.02) | **0.81 (± 0.03)** |
| | *CD-MNAR* | 0.50 | 0.55 (± 0.06) | 0.60 (± 0.02) | 0.61 (± 0.08) | 0.68 (± 0.04) | 0.69 (± 0.02) | 0.69 (± 0.02) | 0.72 (± 0.03) | 0.72 (± 0.02) | 0.72 (± 0.03) | **0.72 (± 0.04)** |
| AIR | *FD-MNAR* | 0 | 0.67 (± 0.05) | 0.88 (± 0.04) | 0.70 (± 0.03) | 0.70 (± 0.03) | **0.91 (± 0.03)** | 0.89 (± 0.03) | 0.90 (± 0.04) | 0.90 (± 0.04) | 0.90 (± 0.02) | 0.90 (± 0.03) |
| | *FD-MNAR* | 0.25 | 0.69 (± 0.02) | 0.85 (± 0.02) | 0.70 (± 0.02) | 0.71 (± 0.03) | **0.90 (± 0.01)** | 0.87 (± 0.02) | 0.89 (± 0.03) | 0.89 (± 0.04) | 0.88 (± 0.02) | 0.88 (± 0.03) |
| | *FD-MNAR* | 0.50 | 0.69 (± 0.02) | 0.81 (± 0.02) | 0.69 (± 0.03) | 0.70 (± 0.04) | 0.87 (± 0.04) | 0.85 (± 0.03) | 0.85 (± 0.04) | 0.85 (± 0.04) | **0.86 (± 0.03)** | 0.84 (± 0.04) |
| | *CD-MNAR* | 0 | 0.66 (± 0.08) | 0.88 (± 0.03) | 0.71 (± 0.05) | 0.71 (± 0.06) | **0.92 (± 0.03)** | 0.90 (± 0.02) | 0.90 (± 0.01) | 0.91 (± 0.01) | 0.91 (± 0.01) | 0.91 (± 0.01) |
| | *CD-MNAR* | 0.25 | 0.66 (± 0.09) | 0.83 (± 0.04) | 0.71 (± 0.04) | 0.70 (± 0.06) | 0.87 (± 0.04) | 0.85 (± 0.05) | 0.88 (± 0.03) | **0.89 (± 0.03)** | 0.87 (± 0.02) | 0.88 (± 0.02) |
| | *CD-MNAR* | 0.50 | 0.69 (± 0.05) | 0.80 (± 0.06) | 0.71 (± 0.04) | 0.69 (± 0.06) | 0.85 (± 0.04) | 0.85 (± 0.06) | 0.85 (± 0.02) | **0.86 (± 0.02)** | 0.84 (± 0.03) | 0.83 (± 0.03) |
| ELECTRIC | *FD-MNAR* | 0 | 0.33 (± 0.05) | 0.62 (± 0.10) | 0.65 (± 0.04) | **0.91 (± 0.03)** | 0.82 (± 0.03) | 0.76 (± 0.04) | 0.71 (± 0.03) | 0.73 (± 0.03) | 0.74 (± 0.03) | 0.63 (± 0.05) |
| | *FD-MNAR* | 0.25 | 0.33 (± 0.04) | 0.61 (± 0.04) | 0.51 (± 0.10) | **0.89 (± 0.03)** | 0.76 (± 0.03) | 0.70 (± 0.03) | 0.70 (± 0.04) | 0.70 (± 0.03) | 0.73 (± 0.02) | 0.72 (± 0.03) |
| | *FD-MNAR* | 0.50 | 0.29 (± 0.03) | 0.60 (± 0.01) | 0.52 (± 0.10) | **0.87 (± 0.03)** | 0.70 (± 0.02) | 0.59 (± 0.01) | 0.64 (± 0.08) | 0.63 (± 0.07) | 0.68 (± 0.02) | 0.66 (± 0.03) |
| | *CD-MNAR* | 0 | 0.33 (± 0.06) | 0.61 (± 0.01) | 0.64 (± 0.01) | **0.90 (± 0.02)** | 0.84 (± 0.01) | 0.76 (± 0.04) | 0.75 (± 0.02) | 0.74 (± 0.02) | 0.74 (± 0.02) | 0.72 (± 0.04) |
| | *CD-MNAR* | 0.25 | 0.37 (± 0.09) | 0.61 (± 0.02) | 0.53 (± 0.08) | **0.89 (± 0.02)** | 0.80 (± 0.04) | 0.67 (± 0.04) | 0.73 (± 0.02) | 0.72 (± 0.01) | 0.72 (± 0.01) | 0.74 (± 0.04) |
| | *CD-MNAR* | 0.50 | 0.32 (± 0.04) | 0.58 (± 0.03) | 0.44 (± 0.00) | **0.84 (± 0.04)** | 0.73 (± 0.03) | 0.59 (± 0.04) | 0.68 (± 0.03) | 0.66 (± 0.03) | 0.66 (± 0.04) | 0.69 (± 0.04) |
| TADPOLE | *FD-MNAR* | 0 | 0.56 (± 0.05) | 0.59 (± 0.02) | 0.74 (± 0.00) | 0.72 (± 0.02) | 0.71 (± 0.03) | 0.77 (± 0.05) | 0.83 (± 0.02) | 0.83 (± 0.01) | 0.84 (± 0.03) | **0.86 (± 0.02)** |
| | *FD-MNAR* | 0.25 | 0.56 (± 0.08) | 0.58 (± 0.04) | 0.73 (± 0.00) | 0.81 (± 0.04) | 0.77 (± 0.04) | 0.78 (± 0.04) | 0.85 (± 0.01) | 0.86 (± 0.01) | 0.85 (± 0.03) | **0.86 (± 0.02)** |
| | *FD-MNAR* | 0.50 | 0.51 (± 0.07) | 0.57 (± 0.01) | 0.70 (± 0.00) | 0.78 (± 0.06) | 0.78 (± 0.05) | 0.76 (± 0.04) | 0.85 (± 0.03) | 0.84 (± 0.02) | 0.85 (± 0.02) | **0.86 (± 0.02)** |
| | *CD-MNAR* | 0 | 0.55 (± 0.07) | 0.61 (± 0.02) | 0.69 (± 0.00) | 0.84 (± 0.02) | 0.80 (± 0.03) | 0.77 (± 0.05) | 0.83 (± 0.02) | 0.83 (± 0.01) | 0.84 (± 0.03) | **0.86 (± 0.02)** |
| | *CD-MNAR* | 0.25 | 0.52 (± 0.06) | 0.58 (± 0.04) | 0.24 (± 0.00) | 0.80 (± 0.04) | 0.79 (± 0.03) | 0.77 (± 0.02) | 0.83 (± 0.03) | 0.83 (± 0.02) | 0.84 (± 0.02) | **0.85 (± 0.03)** |
| | *CD-MNAR* | 0.50 | 0.51 (± 0.11) | 0.55 (± 0.03) | 0.24 (± 0.00) | 0.75 (± 0.07) | 0.80 (± 0.02) | 0.74 (± 0.04) | 0.84 (± 0.01) | 0.83 (± 0.01) | 0.83 (± 0.03) | **0.85 (± 0.03)** |

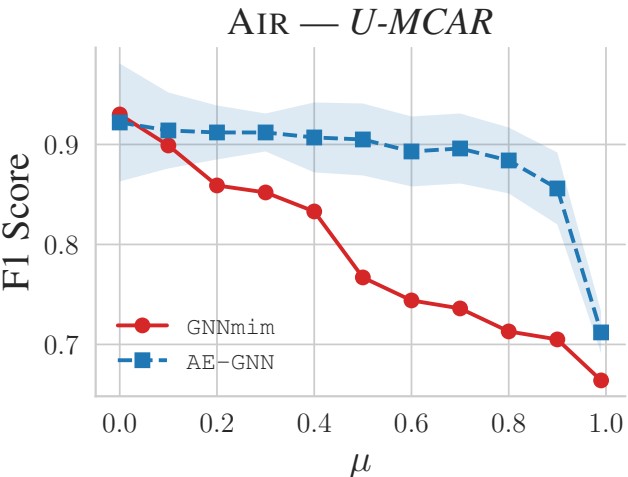

*Figure 9.* F1 score as a function of the missingness rate $\mu$ on the AIR dataset under U-MCAR. GNNmim is trained on the raw 7-dimensional features, while AE-GNN is trained on 256-dimensional autoencoder embeddings of the same features. Despite starting from comparable performance at $\mu = 0$, AE-GNN is virtually unaffected by missingness up to very high rates, confirming that masking learned embeddings does not meaningfully expose the effects of feature missingness.

## K. Feature-Level Missingness on Learned Embeddings

In Section 3 we argued that learned embeddings are unsuitable for evaluating robustness to missing node features, because the information they encode is typically distributed redundantly across many latent dimensions in an overparameterized manner (Arora et al., 2016a;b). As a consequence, masking individual embedding dimensions does not meaningfully expose the effects of missingness. In this appendix, we empirically validate this claim.

**Setup.** We start from the AIR dataset, whose raw node features consist of 7 semantically meaningful environmental measurements. We train a standard autoencoder on the complete feature matrix to project these 7-dimensional raw features into a 256-dimensional latent space, yielding an overparameterized representation that mirrors the typical regime of learned embeddings used in large-scale graph benchmarks. We then apply U-MCAR feature missingness at the same rates $\mu \in [0, 1]$ to both: (i) the raw 7-dimensional features, on which we train GNNmim, and (ii) the 256-dimensional autoencoder embeddings, on which we train an analogous GNN (AE-GNN). At $\mu = 0$, the two models achieve comparable F1 scores ($\sim 0.92$–$0.93$), confirming that the autoencoder does not alter the predictive signal but merely redistributes it across more dimensions.

**Results.** Figure 9 reports the F1 score as a function of $\mu$ for both models. The two curves reveal a striking gap: AE-GNN maintains near-constant performance up to $\mu = 0.7$ (from 0.922 at $\mu = 0$ to 0.896 at $\mu = 0.7$), and degrades sharply only when $\mu \to 1$. In contrast, GNNmim on raw features starts to degrade already at $\mu = 0.2$ ($0.930 \to 0.859$). Remarkably, at $\mu = 0.5$ AE-GNN (0.905) even outperforms GNNmim at $\mu = 0.1$ (0.899): losing 50% of the embedding dimensions is less harmful than losing 10% of the raw features.

**Discussion.** This experiment confirms empirically what we argued conceptually in Section 3: feature-level missingness on embeddings does not constitute a meaningful evaluation challenge, precisely because the original 7-dimensional raw signal is spread redundantly across 256 latent dimensions. Even at high missingness rates, the surviving dimensions retain enough information to reconstruct the predictive signal, making the task essentially unaffected by the missingness mechanism. This further reinforces our position that benchmarks based on learned embeddings are ill-suited for evaluating the robustness of GNNs to missing node features, independently of dataset scale.

## L. Additional Baselines: Classical Imputation and Non-Graph Methods

The main paper compares GNNmim against specialized GNN-based methods for incomplete features. In this appendix we complement this analysis with two further families of baselines, addressing two distinct questions: (i) does explicitly

modeling the missingness indicator outperform classical multivariate imputation used as preprocessing? and (ii) is the graph structure itself necessary, or can simpler non-graph models achieve comparable performance on our datasets?

We focus on the *CD-MNAR* mechanism, as it is the most challenging setting in our protocol: missingness depends on the value of features that are informative for the label, and methods that implicitly assume MAR (such as standard iterative imputation) are expected to be most affected.

### L.1. Comparison with Classical Iterative Imputation

We compare `GNNmim` against `MICE+GNN`, where missing entries are first imputed via Multiple Imputation by Chained Equations (Van Buuren & Groothuis-Oudshoorn, 2011) and the resulting (complete) feature matrix is then used as input to a standard GNN with the same backbone search as `GNNmim`. This isolates the effect of explicitly modeling the missingness mask versus performing high-quality imputation as a preprocessing step.

Figure 10 reports F1 score as a function of the missingness rate $\mu$ under CD-MNAR. Across all four datasets, `GNNmim` is consistently competitive with or outperforms `MICE+GNN`, with the gap widening as $\mu$ increases. The effect is particularly pronounced on ELECTRIC (F1 0.850 vs. 0.561 at $\mu$=0.5) and SYNTHETIC (F1 0.725 vs. 0.621 at $\mu$=0.5), where MICE's MAR assumption is violated by construction. This confirms that under MNAR mechanisms, explicitly exposing the missingness pattern to the model yields more robust performance than even sophisticated imputation as preprocessing.

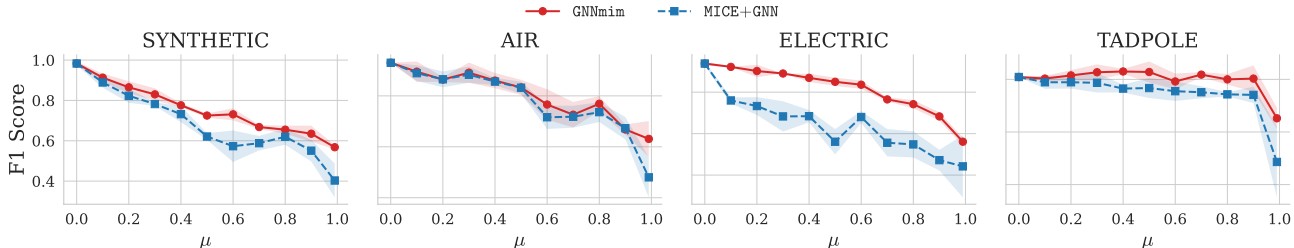

*Figure 10.* F1 score (mean $\pm$ std over 5 runs) as a function of the missingness rate $\mu$ under *CD-MNAR*. `GNNmim` is compared against `MICE+GNN`, where missing values are imputed via Multiple Imputation by Chained Equations before being fed to the same GNN backbone.

### L.2. Comparison with Non-Graph Baselines

To assess whether the graph structure is genuinely useful on our proposed datasets, we additionally compare `GNNmim` against two strong non-graph baselines that operate directly on node features (ignoring the adjacency structure):

- `MLP+MIM`: a multilayer perceptron applied to the concatenation of zero-filled features and the binary missingness mask, with the same MIM principle used in `GNNmim` but without any message passing.

- `XGBoost` (Chen & Guestrin, 2016): gradient-boosted decision trees with native handling of missing values, which is a strong baseline on tabular data with missingness.

Figure 11 reports results under CD-MNAR. `GNNmim` consistently and substantially outperforms both non-graph baselines across all missingness levels. The gap is already large at $\mu$=0 (e.g., SYNTHETIC: `GNNmim` 0.983 vs. `MLP+MIM` 0.752), confirming that graph structure is informative independently of missingness. Crucially, the gap *widens* as $\mu$ increases (ELECTRIC at $\mu$=0.5: `GNNmim` 0.850 vs. `XGBoost` 0.627; TADPOLE at $\mu$=0.9: `GNNmim` 0.803 vs. `MLP+MIM` 0.435), showing that neighborhood aggregation increasingly compensates for feature loss. These results confirm that the graph structure in our proposed datasets is genuinely informative for the prediction task and is not an artifact, while also showing that the MIM principle alone is not sufficient: combining it with message passing is what makes `GNNmim` robust under high missingness.

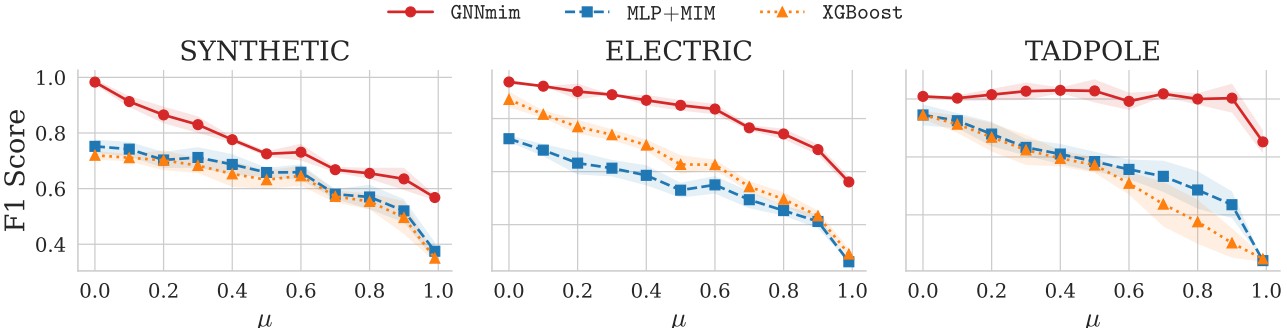

*Figure 11.* F1 score (mean ± std over 5 runs) as a function of the missingness rate $\mu$ under *CD-MNAR*. GNNmim is compared against two non-graph baselines: MLP+MIM (a multilayer perceptron with the same missingness-mask augmentation) and XGBoost with native handling of missing values.

