# OpenReview forum: "Rethinking GNNs and Missing Features: Challenges, Evaluation and a Robust Solution"
_ICML.cc/2026/Conference — ICML 2026 regular_

### Official Review · Reviewer_sqWS · 2026-02-22

**Soundness:** 3
**Presentation:** 2
**Significance:** 3
**Originality:** 2
**Overall Recommendation:** 4
**Confidence:** 4

**Summary:**

The paper studies the problem of graph data with missing node features. The authors provide theoretical analysis to demonstrate the flaws of existing benchmarks and why they are not suitable for missing node features. Four new datasets are introduced to setup a more meaningful evaluation framework. Besides, the authors proposed multiple different missingness mechanisms to reflect more realistic and challenging scenarios. Finally, a model named GNNmim is proposed to demonstrate the robustness over multiple datasets. Experiments show it could outperform multiple SOTA baselines.

**Compliance With Llm Reviewing Policy:**

Affirmed.

**Final Justification:**

Weak accept after the authors claim to include more discussion to address my concerns.

**Key Questions For Authors:**

To prove the invalid prerequisites for OGB dataset with embedding based algorithms, have the authors tried to experiment by comparing the following two tasks?
1. Use the proposed dataset, train a valid node representation learning model to generate node embeddings (need to verify embedding quality). Randomly mask embeddings by different ratios. Train multiple node classification models.
2. Use the same proposed dataset, random masked raw features. Compare the same group of node classification models.

If the outputs of above experiments are totally different, then it may indicate the arguments of embedding based methods are valid. If the outputs show similar trends of degradation when introducing higher masking probability, this might require further theoretical analysis to demonstrate whether the node embeddings-based features are valid or not.

**Limitations:**

yes.

**Strengths And Weaknesses:**

**Strength**
1. The paper has strong motivation on improving the existing benchmarks for missingness problem on graph data and including theoretical analysis to support the claims.
2. The paper introduces multiple new datasets which solved the problems of existing benchmarks. Experiments results also demonstrate this.
3. A new model called GNNmim is introduced which shows strong performance compared to other baselines.

**Weakness**
1. The novelty of the proposed GNNmim method is very limited. The concatenation of missingness indicator matrix and feature could be found in a lot of existing works. For instance BRITS [1] from Neurips 2018 already included this method for time series imputations, and there are multiple other works have also applied this to spatio-temporal data mining domain such as GRIN [2].
2. The authors do not provide concrete examples / reasons / analysis other than some arguments about why embedding based node features are not suitable for missing feature study. This is not super convincing. It is still valid to have embedding based node features on related GNN tasks, for instance one extreme situation could be the entire node feature embeddings are missing on random nodes. Besides, embeddings already encode features semantic and node topological information, applying missingness to certain dimensions still introduces valid lost information to evaluate model’s robustness even though the features are not orthogonal similar to dataset with raw feature.
3. The proposed datasets are small in terms of number of nodes and features, which cannot generalize more complex real-world graph problems.
4. The contribution of the paper is not clear: (1) the theoretical analysis of features scarcity limits the scope of the problem and generalization ability of the proposed methods. The proposed datasets suit the conditions constrained by the authors’ analysis but not reflecting the more complex real world scenario. (2) It seems the paper is more about proposing new datasets rather than corresponding solutions. It would be better if the authors could be clear on the contributions.

[1] Cao, Wei, Dong Wang, Jian Li, Hao Zhou, Lei Li, and Yitan Li. "Brits: Bidirectional recurrent imputation for time series." Advances in neural information processing systems 31 (2018).

[2] Cini, Andrea, Ivan Marisca, and Cesare Alippi. "Filling the G_ap_s: Multivariate Time Series Imputation by Graph Neural Networks." In International Conference on Learning Representations 2022.

---

> ### Author Rebuttal · Authors · 2026-03-31
>
> We thank the Reviewer for the detailed feedback. We have addressed all concerns below. Where figures or tables are needed, we provide anonymous links as per the submission guidelines; however, the textual description is self-contained.
>
> **Novelty**
>
> We fully acknowledge that the MIM principle is not a novel approach, and it was not our intention to present it as such. Note that we included a citation of Van Ness et al. in the paper as a reference not only to the principle, but also to the MIM term and acronym. Van Ness et al. is closer related to our work than references [1,2] cited by the reviewer not only due to the shared terminology, but, more importantly, because Van Ness et al. also point out the appropriateness of MIM in MNAR settings.
> BRITS and GRIN apply indicator-based representations to a fundamentally different problem: multivariate time series imputation, assuming stationarity of the missingness distribution and focusing on the MAR regime. In contrast, GNNmim targets node classification on static graphs under arbitrary missingness mechanisms, including MNAR and train-test distribution shifts, where robustness rather than imputation accuracy is the evaluation criterion. To our knowledge, no prior work applies MIM in this setting.
> That being said, we acknowledge that the relation to prior works using MIM should be better clarified. We will revise the paper to include a dedicated discussion, explicitly covering existing approaches (e.g., BRITS, GRIN) and clarifying that our goal is not to present MIM as a novel method, but to provide a principled re-evaluation of how missingness is investigated (including dataset sparsity, more realistic MNAR settings, and distribution shifts). In such a more realistic framework, the adaptation of an existing base solution (GNNmim) is highly competitive.
>
>
>
> **Generalizability on bigger datasets**
>
> Please refer to our response to Reviewer iL2S (Weakness 1) for a detailed answer to this concern, including new experiments on large-scale datasets.
>
> **Contribution not clear**
>
> Our work explicitly aims to address more realistic and complex scenarios than prior literature. Existing works rely on datasets such as cora, citeseer and pubmed and on MCAR mechanisms, which we show are in fact overly simplistic for studying missingness.
> Our contribution is therefore twofold and tightly connected. First, we identify a fundamental limitation in current evaluation practices and introduce new datasets and protocols that better reflect realistic and more complex missingness scenarios, where the problem is non-trivial. Second, we show that, in these more appropriate settings, the choice of method matters, and we provide a simple yet effective solution (GNNmim) that does not rely on restrictive MAR assumptions. From this perspective, the contribution is not “yet another method”, but rather a reframing of the problem itself; we believe this type of contribution is essential for a top conference: clarifying problem formulations and establishing reliable evaluation standards are necessary steps for meaningful progress in the field.
> Importantly, we do not only propose new datasets: we demonstrate that conclusions drawn from prior evaluations do not hold under more realistic conditions, and we provide a concrete assumption-free baseline that performs strongly in this setting.
> We will revise the paper to make this contribution clearer.
>
>
> **Raw vs. Learned Features**
>
> The core focus of our paper is realistic feature-level missingness, where individual observed attributes are missing, as occurs in healthcare, sensor networks, and IoT applications. This setting requires raw, semantically meaningful features: missingness must correspond to the absence of a measurable quantity (e.g., a patient's weight, a sensor reading). Embeddings are outputs of an algorithm and do not admit this interpretation, there is no realistic mechanism by which individual latent dimensions would be missing in practice. Beyond this conceptual argument, we validate empirically that feature-level missingness on embeddings is also informationally vacuous. Following the reviewer's suggestion, we train an autoencoder on the complete AIR dataset features  and project them into a 256-dimensional latent space, then apply U-MCAR at the same rates to both raw features (GNNmim) and AE embeddings (AE-GNN); plot at[ https://ibb.co/zWJbCBx8 ]. AE-GNN maintains near-constant performance up to μ=0.7 (0.922→0.896), while GNNmim on raw features degrades already at μ=0.2 (0.930→0.859). At μ=0.5, AE-GNN (0.905) even outperforms GNNmim at μ=0.1 (0.899): losing 50% of embedding dimensions is less harmful than losing 10% of raw features. This confirms that feature-level missingness on embeddings does not constitute a meaningful evaluation challenge, precisely because the 7-dimensional raw signal is spread redundantly across 256 latent dimensions. We’ll include this experiment in the paper.

---

> > ### Author Rebuttal · Reviewer_sqWS · 2026-04-01
> >
> > Thank you for the feedbacks. Please make sure to include all the additional details and discussions in the paper. I have raised my scores accordingly.

---

### Official Review · Reviewer_yq2e · 2026-03-10

**Soundness:** 3
**Presentation:** 3
**Significance:** 3
**Originality:** 3
**Overall Recommendation:** 4
**Confidence:** 4

**Summary:**

This paper focuses on the challenges, evaluation systems, and solutions of Graph Neural Networks (GNNs) in scenarios with missing node features. The authors prove that high sparsity substantially limits the information loss caused by missingness, and design evaluation protocols with more realistic missingness mechanisms. Also, the authors propose GNNmim, a simple baseline for node classification with incomplete feature data.

**Compliance With Llm Reviewing Policy:**

Affirmed.

**Final Justification:**

The authors have addressed all of my issues. So I will maintain the score, and accept this paper.

**Key Questions For Authors:**

Please refer to the Weaknesses.

**Limitations:**

No, the authors do not discuss the limitations or potential negative societal impact.

**Strengths And Weaknesses:**

Strengths


1.The organization of this paper is clear.

2.Empirical findings are well visualized.

3.The evaluation design could help realign the subfield toward more realistic robustness.


Weaknesses

1.Theorem 3.2’s lower bound is quite loose and does not involve Y except through an inequality that upper-bounds −Δ by H(X|X̃); its practical interpretation (“provably negligible”) may overstate what the bound guarantees.

2.Absence of strong non-graph baselines under missingness (e.g., MLP/LogReg + MIM/masks, XGBoost/CatBoost with native missing handling) limits conclusions about when graph structure is actually necessary on the new datasets.

3.The evaluation of missingness mechanism shift (R2 regime) only considers the shift from MNAR to MCAR, not covering other common shift types (e.g., MCAR to MNAR).

---

> ### Author Rebuttal · Authors · 2026-03-31
>
> We thank the Reviewer for the detailed feedback. We have addressed all concerns below. Where figures or tables are needed, we provide anonymous links as per the submission guidelines; however, the textual description is self-contained.
>
> **Theorem limitation**
>
> We thank the reviewer for the insightful comment. We agree that the lower bound in Theorem 3.2 can be coarse and omits a possible dependence on Y. Our main goal is to provide a worst-case characterization in terms of feature sparsity and missingness rate. We will revise the wording to avoid overstating the result (e.g., “provably negligible”) in a task-specific sense.
> While loose in general, the bound becomes tight under specific conditions that align with the proof steps: (i) if Y is a deterministic, injective function of X, then the relaxation from (11) to (12) is exact; (ii) if the feature entries X_{i,j} are independent, the decomposition in (13) holds with equality; (iii) if, all X_{i,j} share the same marginal distribution, the bound following (14) is also tight. While these conditions are very strong and not fully realistic, they provide useful insights into the nature of the bound: for missing data to incur a large information loss there has to be significant mutual information to begin with (Cond. (i);  if, at the opposite extreme, Y and X are independent, then Δ=0, and the lower bound is very loose). Conditions (ii) and (iii) are independence and uniformity conditions that enable an expression of the effect of missing feature values in terms of the simple summary statistics of  expected sparsity.
> Thus, while not incorporating all potentially relevant factors, our bound focusses on the effect of sparsity, and shows that in highly sparse settings, sparsity alone can fundamentally limit the impact of missingness, independently of the task. We will incorporate this discussion in the revision.
>
>
> **Missing non-graph baselines**
>
> We added MLP+MIM and XGBoost (native NaN handling) and evaluated them on three datasets under CD-MNAR (figure:[ https://ibb.co/kstgh6Gz ], results will be added to the paper). GNNmim consistently and substantially outperforms both non-graph baselines across all missingness levels. The gap is large even at μ=0 (Synthetic: GNNmim 0.983 vs MLP 0.752), confirming that graph structure is informative independently of missingness. It widens as μ increases (Electric μ=0.5: GNNmim 0.850 vs XGBoost 0.627; Tadpole μ=0.9: GNNmim 0.803 vs MLP 0.435), showing that neighbor aggregation increasingly compensates for feature loss. These results confirm that graph structure in our proposed datasets is genuinely informative and not an artifact.
>
> **Missing R2 cases**
>
> We ran the reverse shift direction (U-MCAR train, μ=0.5 → FD-MNAR or CD-MNAR test, μ ∈ {0, 0.25, 0.50}) across all four datasets (results will be added to the paper, full table:[ https://ibb.co/7xj4rtY7 ]). GNNmim remains competitive or best-performing in many of the settings. On Synthetic and Tadpole it achieves the best F1 in almost all configurations. On Air, PCFI is slightly stronger in some rows but GNNmim remains within standard deviation. Overall, these results confirm that GNNmim's robustness is not limited to the MNAR→MCAR direction and extends to the reverse shift as well.
>
> **Limitations**
>
> We respectfully note that the paper includes an explicit Impact Statement (page 9 of the submission) that discusses both the intended positive contributions and the potential societal implications of this work. Regarding limitations, these are discussed in the Conclusion (Section 6), where we acknowledge the need for larger benchmarks specifically designed for missing features and note that room remains for developing models robust to diverse types of missingness. We will make these limitations more prominent in the final version of the paper.

---

> > ### Author Rebuttal · Reviewer_yq2e · 2026-04-01
> >
> > The authors solved all of my issues about this manuscript.

---

> > > ### Author Response · Authors · 2026-04-01
> > >
> > > Thank you for your positive feedback; we are glad that we were able to address all your concerns.
> > > If you feel that the revisions and clarifications have strengthened the paper, we would greatly appreciate your support in the final evaluation.

---

### Official Review · Reviewer_cbFq · 2026-03-11

**Soundness:** 2
**Presentation:** 3
**Significance:** 2
**Originality:** 2
**Overall Recommendation:** 3
**Confidence:** 4

**Summary:**

This paper tackles the problem of node classification with Graph Neural Networks (GNNs) in the presence of missing node features. The authors make three main contributions. First, they critically analyze existing benchmark datasets, arguing that their high sparsity (due to bag-of-words features) makes them unsuitable for evaluating robustness to missingness, as information loss is negligible except at extreme missingness rates.  Second, they critique the prevailing evaluation protocol of using only uniform MCAR missingness and propose a more challenging and realistic suite of missingness mechanisms, including Label-Dependent MCAR and two MNAR variants. Finally, they propose a simple yet effective method, GNNmim, which applies the Missing Indicator Method (MIM) to GNNs by concatenating a zero-imputed feature matrix with the missingness mask. Through extensive experiments on newly proposed low-sparsity datasets and under the new missingness regimes, they demonstrate that GNNmim is highly robust, often outperforming existing, more complex methods, particularly under distribution shifts in the missingness mechanism.

**Compliance With Llm Reviewing Policy:**

Affirmed.

**Key Questions For Authors:**

Please refer to the weakness.

**Limitations:**

The study is confined to node classification. It is unclear if the findings, particularly the effectiveness of the simple MIM approach, would generalize to other important tasks like link prediction or graph classification where the structure of missingness could be fundamentally different.

**Strengths And Weaknesses:**

Strengths:

(1) The paper addresses a highly relevant problem in graph representation learning. The central critique of using overly sparse, bag-of-words features as a testbed for missingness is a crucial and well-argued point.

(2) The introduction of more realistic missingness mechanisms is a significant step forward.

(3) The paper introduces and justifies four new datasets  that are shown to be more suitable for this task due to their low sparsity, feature-structure complementarity, and performance separability.


Weaknesses:

(1) Limited Methodological Novelty: The core idea of GNNmim is to concatenate a zero-imputed feature matrix with a missingness mask. This is a direct translation of the Missing Indicator Method (MIM), a technique that has been studied for decades in statistics (e.g., for linear regression) and has been applied in modern deep learning for tabular data. Its application to GNNs, while effective, is a straightforward engineering adaptation and lacks the theoretical or algorithmic innovation expected of a top conference paper.

(2) Unfair and Potentially Misleading Experimental Comparison: This is a major weakness. The paper states that GNNmim uses a GNN encoder tuned via grid search, selecting from GCN, GIN and GraphSAGE. However, the paper does not clarify if the competitor methods were also given the same flexibility to have their core GNN backbone optimized. Many of these competitors are built upon a standard GNN architecture. If they were evaluated using a fixed, potentially suboptimal GNN backbone (e.g., only a GCN) while GNNmim was allowed to pick the best one, the comparison is fundamentally unfair and the reported gains for GNNmim are inflated. This oversight undermines the validity of one of the paper's main claims.

(3) Marginal Performance Gains in Key Settings: On the newly proposed and more challenging datasets (AIR, ELECTRIC, TADPOLE) under the standard i.i.d. missingness regime (R1), GNNmim's performance is often comparable to, and sometimes even slightly worse than, simpler methods like GNNmi, GNNzero and FP. For instance, in Tables 12-16, GNNzero are highly competitive. The most significant gains for GNNmim appear primarily in the shift setting (R2, Table 42). This suggests that for the most common evaluation scenario (i.i.d. missingness), the proposed method does not offer a clear advantage, which significantly tempers the paper's overall message.

(4) Disconnect Between Critique and Solution: The paper provides a strong critique of datasets and evaluation, but the proposed solution (GNNmim) does not directly address the core issue it raises about sparsity. GNNmim is a method to handle missing data, and its evaluation on the new datasets is sound. However, the deep theoretical analysis about why sparse datasets are problematic does not lead to a new method that, for example, is theoretically justified to handle missingness differently on sparse vs. dense features. It feels like two separate contributions (a critique and a simple method) are being packaged together, with the critique serving to highlight the method's performance on new data, but without a unifying theoretical or methodological link.

(5) Scalability and Generalizability Concerns: The paper dismisses large-scale datasets by arguing they are unsuitable due to sparsity or embedding-based features. While this point is valid for the study of missingness, it severely limits the demonstrated applicability of GNNmim. It leaves open the question of whether GNNmim would be effective or even usable on truly large graphs where scalability is a primary concern. The runtime analysis in Table 6 is only on the small synthetic datasets and does not alleviate this concern.

---

> ### Author Rebuttal · Authors · 2026-03-31
>
> We thank the Reviewer for the detailed feedback. We have addressed all concerns below.
>
> **Limited Methodological Novelty**
>
> We agree that GNNmim is simple and builds on the Missing Indicator Method. This is intentional. Our goal is not to introduce a novel architecture, but to provide a principled re-evaluation of how missingness is investigated (including dataset sparsity, more realistic MNAR settings, and distribution shifts). In such a more realistic framework, the adaptation of an existing base solution (GNNmim) is highly competitive. This shows that previously proposed complex methods were largely validated under flawed protocols. When evaluated properly, a simple assumption-free method can match or outperform them.
> From this perspective, the contribution is not “yet another method”, but rather a reframing of the problem itself: we provide a principled foundation for studying missingness in GNNs, and demonstrate that conclusions drawn under current protocols can be misleading. We believe this type of contribution is essential for a top conference: clarifying problem formulations and establishing reliable evaluation standards are necessary steps for meaningful progress in the field. We will make this positioning more explicit in the final version
>
>
>
> **Fairness of Experimental Comparison**
>
> All competitor methods were evaluated using their official implementations and recommended hyperparameter settings. GNNmi, GNNmedian, and GNNzero received the same backbone grid search as GNNmim, as they share the same backbone-agnostic design. For GCNmf, GSPN, FairAC, and GOODIE the backbone is integral to the architecture and cannot be modified without altering the method itself. For FP and PCFI, where the backbone is separable from the imputation strategy, we ran an additional experiment applying the same validation-based grid search (GCN, GraphSAGE, GAT, GIN) used for GNNmim. GraphSAGE is preferred on AIR and ELECTRIC. Updating the results, bold entries change only on AIR in some settings, while SYNTHETIC, ELECTRIC, and TADPOLE are unaffected. Even with this correction, GNNmim remains competitive across all datasets and mechanisms. We will update the paper accordingly
>
>
> **Marginal Performance Gains**
>
> On AIR, ELECTRIC, and TADPOLE, we do not expect GNNmim to always outperform all baselines under i.i.d. missingness. Rather, our goal is to achieve a clear advantage in MNAR settings, where modeling missingness becomes crucial. This is exactly what we observe: as shown in Tables 27–41 (AUC results), GNNmim ranks in the top-2 across all 8 MNAR settings. More broadly, across these datasets and all settings, GNNmim ranks in the top-3 in 19 out of 20 cases, confirming its overall robustness.
> Regarding Tables 12–16, we believe this concern is based on a misunderstanding. These tables correspond to Citeseer, a dataset we explicitly argue is unsuitable for evaluating missingness. Consistently, GNNmim is not included in these tables. On such datasets, features are extremely sparse (>99% zeros), so most “missing” entries are in fact zeros. In this setting, GNNzero performs well simply because it imputes zeros, i.e., it recovers the true value. This does not reflect true robustness, but rather an artifact of the evaluation protocol, precisely the issue our work aims to highlight
>
>
> **Disconnect Between Critique and Solution**
>
> We thank the reviewer for acknowledging that the evaluation on our proposed datasets is sound. We respectfully disagree that the critique and the method are disconnected. Our central claim is that if the evaluation setup is flawed, the priority is to correct it, not to propose more complex models. This is precisely what we do: we provide a principled foundation for studying missingness in GNNs (suitable datasets, realistic missingness protocols and distribution shift), and demonstrate that conclusions drawn under current protocols can be misleading. Within this corrected setting, GNNmim is intentionally a simple baseline, not a novel architecture. Its role is to demonstrate that, once the evaluation is fixed, existing complex methods do not provide the expected advantages, and a minimal, assumption-free approach can perform as well or better.
> Therefore, the theoretical analysis, the new evaluation framework, and GNNmim are tightly connected: the analysis motivates the new evaluation, and GNNmim serves as a clear reference point to reveal its implications
>
>
>
>
> **Scalability and Generalizability Concern**
>
> Please refer to our response to Reviewer iL2S (W1) for a detailed answer, including new experiments on large-scale datasets.
>
> **Limitation**
>
> Extending the analysis to other tasks (e.g., link prediction or graph classification) is an interesting future direction. Our focus on node classification is intentional, as it is the primary setting studied in the existing GNN literature on missing features, which we aim to critically reassess. We will explicitly mention this in the conclusions of the final version.

---

> > ### Author Rebuttal · Reviewer_cbFq · 2026-04-05
> >
> > Thanks for the response. I remain the concerns on Potentially Misleading Experimental Comparison and Marginal Performance Gains in Key Settings, and will maintain my score.

---

> > > ### Author Response · Authors · 2026-04-05
> > >
> > > We thank the reviewer for the acknowledgment.
> > >
> > > On **Potentially Misleading Experimental Comparison**
> > >
> > > We would like to clarify that we followed the reviewer’s request by re-running all methods with a separable backbone using the same grid search as GNNmim; the results remain consistent with our original claims. We would appreciate further clarification on what specifically remains unconvincing, so we could better address it.
> > >
> > > On **Marginal Performance Gains**
> > >
> > > We would like to clarify that Tables 12–16 do not include GNNmim (as explicitly discussed in the manuscript), and therefore not directly relevant to this concern. If the reviewer is instead referring to other datasets, we reiterate that we do not claim GNNmim outperforms all competitors under all i.i.d. missingness settings. Rather, our claim is about robustness across settings, particularly MNAR. As shown in Tables 27–41, GNNmim ranks in the top-2 across all 8 MNAR settings, and in the top-3 in 19 out of 20 cases overall. We believe these results directly address the concern.
> > >
> > > Sincerely,
> > > The Authors

---

### Official Review · Reviewer_iL2S · 2026-03-11

**Soundness:** 3
**Presentation:** 3
**Significance:** 3
**Originality:** 4
**Overall Recommendation:** 4
**Confidence:** 3

**Summary:**

This paper argues that existing evaluations of GNNs under missing node features are fundamentally flawed due to two artifacts: (a) standard benchmarks (Cora, CiteSeer, PubMed) use extremely sparse BoW/TF-IDF features where additional missingness has negligible informational impact, and (b) evaluation protocols rely almost exclusively on simplistic uniform MCAR mechanisms. The paper provides an information-theoretic analysis (Theorems 2.2 and 3.2) formalizing why high sparsity dampens information loss from missingness. To address these gaps, the authors introduce one synthetic and three real-world datasets (Air, Electric, Tadpole) with dense, low-dimensional, semantically meaningful features, along with more realistic missingness protocols: label-dependent MCAR (LD-MCAR), feature-dependent MNAR (FD-MNAR), class-dependent MNAR (CD-MNAR), and train-test distribution shifts. They propose GNNmim, a simple baseline that concatenates a binary missingness indicator mask to zero-filled features, and show it is competitive with or outperforms specialized GNN architectures across all datasets and missingness regimes.

**Compliance With Llm Reviewing Policy:**

Affirmed.

**Final Justification:**

The authors have adequately addressed my concerns, and I believe my current rating reflects the quality of the paper.

**Key Questions For Authors:**

1. How does GNNmim perform relative to a standard GNN with MICE/iterative imputation as preprocessing?
2. Can you provide per-node-degree breakdowns of performance under missingness?

**Limitations:**

Yes

**Strengths And Weaknesses:**

Strengths：
1. The formalization of feature-MAR and label-MAR (Definition 2.1), the ignorability result (Theorem 2.2), and the information-theoretic bound (Theorem 3.2) provide a principled foundation. The connection between feature sparsity and the binary entropy function h_2 in the lower bound is elegant and makes the argument quantitatively precise.
2. Moving beyond U-MCAR to LD-MCAR, FD-MNAR, and CD-MNAR is a meaningful step toward realism. The CD-MNAR mechanism is particularly creative. The train-test distribution shift regime (R2) captures a genuinely important practical scenario that prior work ignores entirely.
3. The proposed model adds zero parameters beyond the standard GNN backbone. This design choice is well-motivated by the theoretical analysis. The fact that this trivial baseline is competitive with complex architectures powerfully reinforces the paper's thesis about evaluation artifacts.
4. The paper tests 10 models across 4 datasets, 5 missingness mechanisms, 11 missingness rates, and 2 evaluation regimes. The AUC heatmaps (Figure 2) provide an effective visual summary.

Weaknesses：
1. The benchmark's small scale limits confidence in generalizability.
2. Looking at the AUC heatmaps and detailed tables, GNNmim frequently performs within standard deviations of simpler baselines and sometimes underperforms specialized methods.
3. The authors treat missingness as operating purely at the feature level, but in graph learning, message passing propagates information from neighbors. How does node degree affect robustness?
4. Theorem 3.2 provides a bound on information loss under U-MCAR for binary features. However, the assumption may not hold for the real-world datasets.

---

> ### Author Rebuttal · Authors · 2026-03-31
>
> We thank the Reviewer for the careful reading and constructive feedback. We have addressed all concerns below. Where figures or tables are needed, we provide anonymous links as per the submission guidelines; however, the textual description is self-contained.
>
> **Generalizability on bigger datasets**
>
> To show that our approach generalizes to large-scale datasets, we adapted our MIM
> approach to RelBench (Fey et al., 2023), a benchmark for relational deep learning
> where relational tables are converted into graphs, featuring large-scale, real-world
> datasets with naturally occurring missingness. The datasets range from 97K to 41M
> rows and from 21 to 140 columns, with missingness rates between 0.07% and 23.9%.
> The base model (RDL) internally uses mean imputation; we obtain RDLmim by adapting the MIM strategy to this setting. Full results are available [ https://ibb.co/Jw8cjFTs ].
>
> On datasets with substantial missingness (rel-trial: 23.9%, rel-event: 12.2%),
> RDLmim improves over the mean-imputation baseline. On rel-arxiv
> (0.07% missingness), results are comparable between RDL and RDLmim, as expected, since MIM provides the largest benefit when missingness is substantial and informative.
>
> Regarding runtime, RDLmim introduces no meaningful overhead across all tasks:
> differences are within standard deviation in all cases, and on study-adverse and
> rel-f1 RDLmim is marginally faster than the base RDL.
>
>
>
>
> **GNNmim results limitation**
>
> We agree that GNNmim does not always strictly outperform specialized methods. Our goal is not to propose a SOTA method under a fixed missingness assumption, but rather a model that is robust and competitive across diverse missingness mechanisms. In real-world scenarios, the missingness mechanism is typically unknown, and methods tailored to specific assumptions (e.g., MAR) may degrade when these are violated. GNNmim is assumption-free, making it applicable across MCAR, MAR, and MNAR settings.
> To quantify robustness, we consider Tables 22–41 (4 datasets × 5 missingness mechanisms) and compute a conservative ranking by adjusting performance using (mean − std) and aggregating via AUC. Under this evaluation, GNNmim ranks in the top-3 in 19 out of 20 settings, and within the top-2 in all 8 MNAR settings, where missingness is informative and MAR-based methods may fail.
> Overall, GNNmim provides consistent performance across heterogeneous conditions, which we believe is more aligned with practical deployment. We acknowledge that this framing could be made more explicit in the paper, and we will clarify in the final version that GNNmim serves as a simple yet robust baseline within a corrected evaluation framework, rather than a SOTA method.
>
> **Not realistic assumption**
>
> We would like to clarify that the role of Theorem 3.2 is not to model realistic real-world data, but rather to expose a limitation of current evaluation practices, which do rely on these assumptions. The assumption of binary features under U-MCAR is intentionally aligned with widely used benchmarks such as cora, citeseer and pubmed, where node features are sparse and binary (bag-of-words). Our goal is precisely to analyze this prevalent, but unrealistic, setting.
> Under these assumptions, Theorem 3.2 shows that when features are highly sparse, the information loss induced by missingness is inherently limited unless the missingness rate is extremely high. This formally explains why models appear artificially robust on these datasets.
> Therefore, this assumption is a deliberate choice that allows us to demonstrate that current evaluation protocols, which are built on this assumption, are not suitable for assessing robustness to missing features.
>
>
> **Comparison with standard imputation**
>
> We ran this comparison across all our proposed datasets under CD-MNAR (results will be added to the paper). GNNmim is consistently better than MICE+GNN across all datasets and missingness levels, with particularly large gaps on ELECTRIC (e.g., F1 0.850 vs 0.561 at μ=0.5) and SYNTHETIC (0.725 vs 0.621 at μ=0.5). This confirms that explicitly modeling the missingness indicator outperforms iterative imputation as preprocessing even under the challenging class-dependent MNAR mechanism. Full results are shown here: [ https://ibb.co/996rx2xQ ].
>
>
> **Node degree vs robustness**
>
> We conducted a per-degree breakdown across all four datasets under U-MCAR, splitting test nodes into low-, mid-, and high-degree tertiles (figure: [ https://ibb.co/RTW2fMTG ]). On Synthetic, all groups degrade similarly with no advantage for high-degree nodes. On Air and Electric, high-degree nodes are slightly more robust at high μ.

---

> > ### Author Rebuttal · Reviewer_iL2S · 2026-04-01
> >
> > The authors have adequately addressed my concerns, and I believe my current rating reflects the quality of the paper.

---

> > > ### Author Response · Authors · 2026-04-02
> > >
> > > Thank you for your positive feedback; we are glad that we were able to address all your concerns. In addition, we have further strengthened the paper by:
> > >
> > > (i) extending the evaluation to different distribution shifts,
> > > (ii) adding comparisons with non-graph baselines,
> > > (iii) providing empirical evidence that also missingness on embeddings is not a meaningful evaluation setting,
> > > (iv) clarifying our core contribution: establishing reliable evaluation standards, which are key to meaningful progress in the field,
> > > (v) including large-scale experiments.
> > >
> > > If you feel these improvements further reinforce the contribution, we would greatly appreciate your support in the final evaluation.

---

### Decision · Program_Chairs · 2026-04-30

**Decision:**

Accept (regular)

**Comment:**

This paper studies node classification with GNNs under missing features. It argues that widely used benchmarks with highly sparse bag-of-words features are fundamentally unsuitable for evaluating missingness, and introduces both more realistic missingness mechanisms (including MNAR variants and distribution shifts) and new low-sparsity datasets. It also proposes a simple baseline, GNNmim, which augments zero-imputed features with a missingness indicator mask. The reviewers agree that the critique of current evaluation practices, the improved experimental protocols, and the introduction of new datasets are important contributions, supported by theoretical motivation and extensive empirical evaluation.

Nevertheless, several concerns have been raised about the overall impact of the paper. The proposed methodology has limited novelty, being a straightforward adaptation of the Missing Indicator Method, and its empirical gains are often modest outside of distribution shift settings. In their response, the authors have partially addressed this concern with additional experiments. Another limitation was the lack of non-graph baselines. Although in their response the authors have included MLP+MIM and XGBoost, I believe these are not very representative, and the list of baselines could further be enhanced. I am also wondering why a Variational Graph Autoencoder was not used for the graph-based baselines. Overall, the paper makes an interesting contribution, even if some aspects could be further enhanced. I would suggest that the authors consider the reviewers' comments in the revised version of the paper.